# Altered succinylation of mitochondrial proteins, APP and tau in Alzheimer's disease

Yun Yang[1,2,3], Victor Tapias [2], Diana Acosta [4], Hui Xu [2,3], Huanlian Chen[2,3], Ruchika Bhawal [5], Elizabeth T. Anderson[5], Elena Ivanova[6,19], Hening Lin [7,8], Botir T. Sagdullaev[9,10,19], Jianer Chen [1], William L. Klein [11], Kirsten L. Viola [11], Sam Gandy [12,13,14,15], Vahram Haroutunian [15,16,17,18], M. Flint Beal[2], David Eliezer [4], Sheng Zhang [5] & Gary E. Gibson [2,3✉]

Abnormalities in brain glucose metabolism and accumulation of abnormal protein deposits called plaques and tangles are neuropathological hallmarks of Alzheimer's disease (AD), but their relationship to disease pathogenesis and to each other remains unclear. Here we show that succinylation, a metabolism-associated post-translational protein modification (PTM), provides a potential link between abnormal metabolism and AD pathology. We quantified the lysine succinylomes and proteomes from brains of individuals with AD, and healthy controls. In AD, succinylation of multiple mitochondrial proteins declined, and succinylation of small number of cytosolic proteins increased. The largest increases occurred at critical sites of amyloid precursor protein (APP) and microtubule-associated tau. We show that in vitro, succinylation of APP disrupted its normal proteolytic processing thereby promoting Aβ accumulation and plaque formation and that succinylation of tau promoted its aggregation to tangles and impaired microtubule assembly. In transgenic mouse models of AD, elevated succinylation associated with soluble and insoluble APP derivatives and tau. These findings indicate that a metabolism-linked PTM may be associated with AD.

[1] Integrated Medicine Research Center for Neurological Rehabilitation, College of Medicine, Jiaxing University, 314001 Jiaxing, China. [2] Feil Family Brain and Mind Research Institute, Weill Cornell Medicine, New York, NY 10065, USA. [3] Burke Neurological Institute, White Plains, NY 10605, USA. [4] Department of Biochemistry, Weill Cornell Medicine, New York, NY 10065, USA. [5] Proteomics and Metabolomics Facility, Institute of Biotechnology, Cornell University, Ithaca, NY 14853, USA. [6] Imaging Core, Burke Neurological Institute, White Plains, NY 10605, USA. [7] Department of Chemistry and Chemical Biology, Cornell University, Ithaca, NY 14853, USA. [8] Howard Hughes Medical Institute, Department of Chemistry and Chemical Biology, Cornell University, Ithaca, NY 14853, USA. [9] Ophthalmology and Neuroscience, Weill Cornell Medicine, New York, NY 10065, USA. [10] Laboratory for Visual Plasticity and Repair, Burke Neurological Institute, White Plains, NY 10605, USA. [11] Department of Neurobiology, Northwestern University, Evanston, IL 60208, USA. [12] Department of Neurology and Mount Sinai Center for Cognitive Health and NFL Neurological Center, Icahn School of Medicine at Mount Sinai, New York, NY 10029, USA. [13] Research and Development Service and Division of Neurology, James J Peters VA Medical Center, 130 West Kingsbridge Rd, Bronx, NY 10468, USA. [14] James J Peters Veterans Medical Center, Bronx, NY 10468, USA. [15] Department of Psychiatry Icahn School of Medicine at Mount Sinai, New York, NY 10029, USA. [16] Department of Neuroscience, Icahn School of Medicine at Mount Sinai, New York, NY 10029, USA. [17] JJ Peters VA Medical Center MIRECC, Bronx, NY 10468, USA. [18] Mount Sinai NIH Neurobiobank, New York, NY 10029, USA. [19] Present address: Regeneron Pharmaceuticals, Inc., Tarrytown, NY 10591, USA. ✉email: ggibson@med.cornell.edu

Misfolded deposits of the amyloid beta peptide (Aβ)[1,2] and the microtubule-associated protein tau (MAPT)[3] are pivotal pathological features in Alzheimer's disease (AD), wherein reduced brain regional glucose metabolism and synaptic density are correlated with the development of clinical cognitive dysfunction[4]. Preclinical research studies show that reduced glucose metabolism exacerbates learning and memory deficits concurrent with the accumulation of Aβ oligomers and plaques[5] and misfolded hyperphosphorylated tau[6,7]. However, the interrelationship(s) linking these keys but apparently disparate pathological processes remain unknown. While identification of proamyloidogenic and/or immune-inflammatory genetic factors has played a prominent role in advancing our understanding of AD, more recent formulations have expanded the scope of molecular underpinnings of the disease[8,9]. Sims and colleagues coined the term "multiplex hypothesis of AD" to highlight the increasingly recognized shortcomings of the "amyloid hypothesis of AD"[9].

Post-translational modifications (PTMs) of proteins provide an efficient and rapid biological regulatory mechanism that links metabolism to protein and cell functions. PTMs contribute to the functional diversity of proteomes without the formation of new proteins or a change in their abundance by covalent addition of functional groups that can alter protein charge, structure, and their interactions. Protein PTMs play a central role in the pathology of neurological diseases. The function of tau can be altered via its phosphorylation[10], acetylation[11], methylation[12], and O-GlcNAcylation[13]. Protein succinylation of lysine residues is a relatively understudied PTM and changes the net charge from positive to negative. The interactions of lysine succinylation and acetylation play important roles in metabolic pathways[14]. However, succinylation is poorly studied in the nervous system; our previous work demonstrated that lysine succinylation functionally modifies enzymes of energy metabolism[15].

There is an increasing interest in defining the precise metabolic pathways involved in the pathogenesis of AD[9,16–19]. A significant correlation between reduced brain regional glucose metabolism and decreased α-ketoglutarate dehydrogenase complex (KGDHC)[20,21] has been described in AD. Inhibition of KGDHC activity leads to a widespread reduction in regional brain post-translational lysine succinylation, for which the succinyl donor is presumably succinyl-CoA, both in yeast[22] and cultured neurons[15,23,24]. Studies of organisms deficient in NAD$^+$-dependent desuccinylase sirtuin 5 (SIRT5)[25] provide evidence of the regulatory importance of succinylation in metabolic processes[26–30]. However, the role of succinylation in metabolic pathways of the human nervous system or in neurodegenerative diseases is unknown. Our research examines the human brain succinylome and its changes in AD. The results suggest that succinylation may link AD-related metabolic deficits to structural, functional, and pathological alterations involving APP and tau.

## Results

**Human brain succinylome and proteome.** Analysis of two cohorts each consisting of brain tissues from five controls and five AD patients (information provided in Supplementary Data 1) was performed to maximize the precision and reproducibility of the succinylome (Fig. 1a, b) and the proteome (Fig. 1c, d) determinations. When the two independent cohorts were taken together, 1908 succinylated peptides from 314 unique proteins were identified across a total sample size of 20 brains (Fig. 1b). Over 90% of succinylated peptides being identified were equivalent to 0 miss-cleavage and 10% contained 1 miss-cleavage site in our data, which is consistent with what we observed in our regular global proteomics. The parallel global proteomic analysis detected 4678

proteins (Fig. 1d). Nearly all the succinylated proteins identified during the study were found in the global proteomes of the same samples (Fig. 1e).

To understand the role of succinylation in cell function, subcellular localization analysis of 314 succinylated proteins in 20 human brains was performed (Fig. 2a and Supplementary Data 2). Succinylated proteins were mapped to multiple subcellular compartments. Interestingly, ~73% (229/314) of the succinylated proteins were mitochondrial (Fig. 2a). The pyruvate dehydrogenase complex E1 component subunit alpha (PDHA1), which links glycolysis to the TCA cycle, was significantly succinylated. The eight enzymes of the TCA cycle located within the mitochondrial matrix and their multiple subunits were also extensively succinylated. Furthermore, succinylation of proteins was detected in the cytosol (95 proteins, ~30%) and nucleus (73 proteins, ~23%) (Fig. 2b). Mitochondrial distribution of succinylated proteins has also been observed in the mouse liver[29,30].

The mole fraction of succinylation sites per protein varied from 1 to 23 (Fig. 2c and Supplementary Data 2). Our assessment found 40% (125/314) of proteins with one succinylated site, 20% (60/314) showing two sites, and the remaining 40% (127/314) having three or more succinylated sites. The bulk of mitochondrial proteins (~89%) contain more than two succinylated lysines. In addition, the most extensively succinylated proteins with over ten distinct succinylated sites/peptides were all mitochondrial proteins; 61% (14/21) of these were exclusively mitochondrial proteins, including isocitrate dehydrogenase (IDH2), fumarate hydratase (FH), and malate dehydrogenase (MDH2) (Supplementary Data 2). Overall, these succinylated proteins typically appeared in metabolism-associated processes and were linked to multiple disease pathways in the KEGG enrichment analysis (Supplementary Fig. 1 and Supplementary Data 3).

Since no specific motifs for lysine succinylation in human tissues have been reported, a succinylation motif analysis of all 1908 succinylated peptides using Motif-X was performed. A total of five conserved motifs were identified (Fig. 2d) in which nonpolar, aliphatic residues including alanine, valine and isoleucine surround the acceptor succinylated lysines. The succinylated lysine site analysis indicated a strong bias for alanine residues, which is consistent with motifs identified in tomato[27]. IceLogo heat maps assessed the preference of each residue in the position of a 15 amino-acid-long sequence context (Fig. 2e). Isoleucine was detected downstream of lysine-succinylation sites, while alanine and lysine (two of the most conserved amino-acid residues) were found upstream. Valine residues occurred both upstream and downstream. Tryptophan, proline or serine residues had the lowest probability to occur in succinylated peptides.

**Succinylome and proteome changes in AD brains.** Completion of the human brain succinylome and global proteome analyses allowed direct comparison between brains from controls and AD patients. Without enrichment of succinylated peptide in global proteome data, the number of succinylated peptides identified is 0.13% total peptides for cohort 1 and 0.28% for cohort 2. The notable difference in ratio of succinylated peptides over total peptides between the two cohorts of global proteome datasets is not surprising, since succinylation has relatively low occupancy level. Therefore, there will be an anticipated variation between two cohorts' datasets for detection of those low abundance succinylated peptides under global and complex quantitative proteomics analysis. This assessment also indicates that the enrichment is important for identifying the succinylated peptides in large cohorts. After enrichment, we found that the average enrichment of succinylated peptides was found to be 33.9% in two

**Fig. 1 Global analysis of protein lysine succinylation and proteomic profiles in human brains. a** A schematic diagram of the workflow for investigation of human brain lysine succinylome by label-free quantitation (See methods section). **b** After quantitative data screening and mining, the combined results from 20 brain samples in two batches revealed 932 common succinylated peptides quantified from 259 proteins (Supplementary Data 4). **c** A schematic diagram of the workflow for quantitative proteomics of human brain by Tandem mass tags (TMT) labeling analysis (See methods section). **d** After quantitative data screening and mining, the combined results from 20 brain samples in two batches revealed 4442 common proteins in both AD and controls (Supplementary Data 5). Eighty-one proteins showed significant alterations between samples patients with AD and controls. **e** The overlap between succinylomes and proteomes. Nearly all of the succinylated proteins were also identified in its global proteomic analysis. Source data are provided as a Source Data file.

cohorts while 0.2% of succinylated peptide was identified in global proteome without enrichment. Of 1908 succinylated peptides identified in two independent analyses, 932 succinylated peptides were quantifiable (Fig. 1a). A volcano plot revealed that succinylation of 434 unique peptides declined with AD while the abundance of 498 unique succinylated peptides was increased (Fig. 3a and Supplementary Data 4). Succinylation of 29 peptides from 20 distinct proteins differed significantly (two-tailed Student's $t$-test, $p < 0.05$) between AD and control subjects (Fig. 3a, b). Ten succinylated peptides were increased while succinylation

of 19 peptides declined in AD. Proteomic analysis of 20 samples in two cohorts (Fig. 1c) showed that of the 4678 identified proteins, 4442 common proteins were quantifiable in both AD and controls (Fig. 1d and Supplementary Data 5 and Supplementary Fig. 2a, b). A comparison of the succinylome with the proteome demonstrated minimal AD-related changes in protein levels of those succinylated proteins, and therefore the succinylation variations are most likely independent from the changes of the abundance of each corresponding protein (Fig. 3c). The proteomic analysis showed that 81 proteins changed significantly

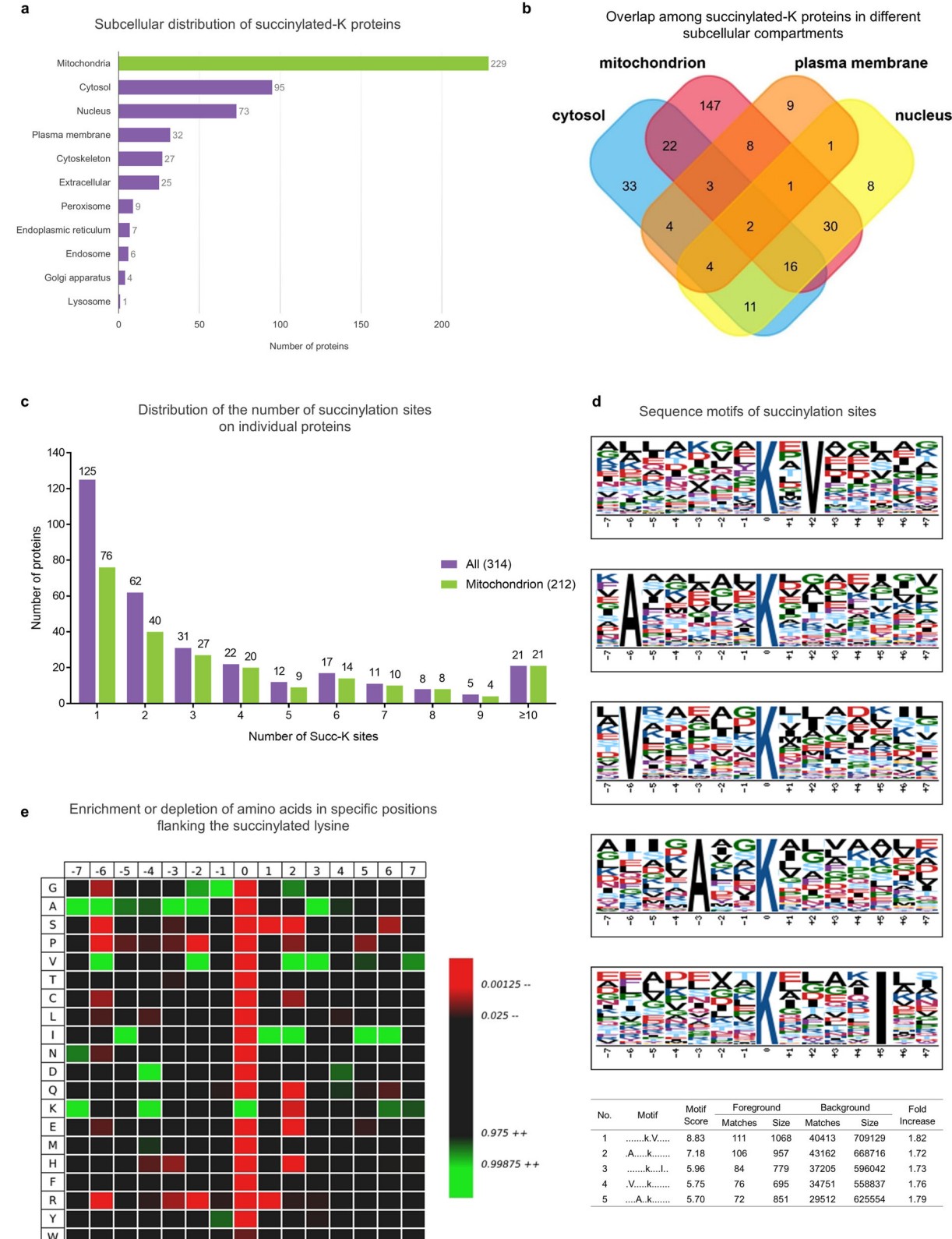

(two-tailed Student's *t*-test, $p < 0.05$ and $|\log_2 FC| > 0.25$). Eight proteins decreased in brains from AD patients while 73 protein levels were increased (Supplementary Fig. 2a). In a recent large-scale proteomic scan, the protein abundance of PDHA, PDHB, and DLD were all decreased in AD, which is consistent with our finding, representing a decreased abundance of proteins in impaired mitochondrial states[17]. Thus, changes in protein levels

and succinylation may be important in AD. Relatively small fold-changes between control and AD brain samples were probably due to a well-known ratio compression caused by the co-isolation of isobaric-labeled background ions in MS2-based TMT quantitative proteomics.

The majority of the peptides (16/19) with AD-related decreases in succinylation were mitochondrial, and more than half of them

**Fig. 2 Subcellular distribution of lysine-succinylation proteins in human brains. a** Subcellular distribution of succinylated-K proteins identified by Cytoscape and stringAPP software. The majority of succinylated-K proteins are mitochondrial. **b** Overlap of succinylated-K proteins located in the mitochondrion, nucleus, cytosol, and plasma membrane. The details of the subcellular distribution of individual proteins are shown in Supplementary Data 2. **c** The extent of succinylation of individual proteins and their enrichment in mitochondria. Distribution of the number of succinylation sites per protein in all of the succinylated proteins (purple bars) or succinylated mitochondrial proteins (green bars) as classified by Cytoscape and stringAPP. **d** The succinylation sites were analyzed for seven amino acids up- and downstream of the lysine residue using Motif-X. The height of each letter corresponds to the frequency of that amino-acid residue in that position. The central blue K refers to the succinylated lysine. **e** Heatmap of the 15 amino-acid compositions of the succinylated site showing the frequency of the different amino acids in specific positions flanking the succinylated lysine. The different colors of blocks represent the preference of each residue in the position of a 15 amino-acid-long sequence context (green indicates greater possibility, while red refers to less possibility). Source data are provided as a Source Data file.

showed exclusive localization within mitochondria (Supplementary Data 6). A novel association of the ATP5H/KCTD2 locus with AD has been reported[31]. Moreover, ATP-synthase activity declines in the AD brain[32]. In line with these findings, we identified the maximal AD-associated decrease ($-1.33$ in $\log_2 FC$) was localized to ATP-synthase subunit d (ATP5H), with two additional peptides from ATP5H down at $-0.52$ and $-0.49$ in $\log_2 FC$. In addition, two peptides from the ATP-synthase subunit b (ATP5F1) were also reduced ($\log_2 FC$ at $-0.47$ and $-0.32$) in the brain of AD patients. Succinylation of three lysine residues ($Lys^{77}$, $Lys^{244}$, and $Lys^{344}$) of PDHA1 were significantly diminished in AD cases (Fig. 3a, b).

The highest AD-related increases in succinylation were in non-mitochondrial proteins (Fig. 3a, b). Succinylation of four peptides from brain cytosolic and/or extracellular brain hemoglobin subunits alpha and beta increased by 1.91 (0.978 in $\log_2 FC$) to 2.18 fold (1.127 in $\log_2 FC$) in AD subjects. Strikingly, two extramitochondrial peptides with the largest AD-associated increases in succinylation were from two proteins critical to AD pathology: APP and tau. Both proteins were succinylated at critical sites in nine out of ten AD brain samples, whereas succinylation at those sites in APP or tau was not detectable in control brains (Fig. 3b, c).

**Subcellular responses of succinylation to impaired mitochondrial function.** Subcellular succinylation in response to perturbed mitochondrial function was determined by compromising the function of mitochondria in HEK293T cells through mild inhibition of complex I. Mitochondrial dysfunction causes a reduction in succinylation in whole-cell lysates and mitochondrial fractions (Fig. 4a), consistent with previous findings in N2a cells[23]. However, alterations in mitochondrial function increased succinylation of 30–70 kDa proteins in the non-mitochondrial fractions. We previously demonstrated that mitochondrial dysfunction can alter mitochondrial/cytosolic protein signaling[33]. Here we extend this line of investigation by showing that mitochondrial dysfunction resulted in a release of mitochondrial proteins including all subunits of pyruvate dehydrogenase complex (PDHC) and KGDHC (Fig. 4b, c). This was not due to disruption of the mitochondrial integrity because cytochrome c oxidase subunit 4 isoform 1 (COX-IV), an integral membrane protein in mitochondria, did not increase in the cytosol fraction. Confocal microscopy further confirmed that rotenone caused a redistribution of mitochondrial proteins without mitochondrial lysis, as mitochondria were clearly outlined by COX-IV immunolabeling. Exogenous administration of rotenone increased the amount of the cytosolic E2k component of KGDHC (DLST) outside of mitochondria defined by COX-IV (Fig. 4d). Thus, impaired mitochondrial function induced a metabolic disturbance leading to an increased leakage of mitochondrial proteins into the cytosol, including DLST. Consistent with its identity as a succinyltransferase[34] and succinyl-CoA generator[35], DLST

was associated with elevated succinylation in non-mitochondrial fractions.

**Functional significance of succinylation of APP.** AD-associated succinylation of APP occurred at a critical site (K612) in nine of ten brains from AD patients but not in brains from age-matched subjects with no dementia (Fig. 5a, b), and the following experiments demonstrated it to be pathologically important. Immunofluorescence staining with antibodies to pan-lysine-succinylation and to Aβ oligomers (NU-4)[36] or Aβ plaques (β-Amyloid, D3D2N) showed an early increase in lysine succinylation that appeared to parallel oligomer accumulation (Fig. 5c and Supplementary Fig. 3a) in the hippocampus of a transgenic mouse model of AD (Tg19959 mice), which carries the human APP with two pathogenic familial AD mutations. However, the immunoreactivity of lysine succinylation was significantly decreased in 10-month-old wild-type and transgenic mice relative to 4-month-old mice, which results in a reduced colocalization between lysine succinylation and Aβ plaque accumulation (Fig. 5d and Supplementary Fig. 3b). This could result either from a decrease in lysine succinylations or from their sequestration into a context (e.g., perhaps in the form of Aβ plaques) that prevents Succi-K antibody from access to possibly buried succinylation sites. These findings suggest that APP succinylation might be involved in early Aβ aggregation events in vivo, while its role and mechanism in later events leading to subsequent plaque development remain to be further explored.

The generation of Aβ is a highly regulated process by the secretases. β-secretase initiates the amyloidogenic pathway, while α-secretase is part of the non-amyloidogenic pathway, bisecting the Aβ domain and thereby inhibiting the formation of Aβ. In subsequent experiments, we tested the relationship between succinylation and APP processing by the secretase enzymes. K612-L613 is the APP α-secretase scissile bond, and missense mutation at K612N produces early-onset AD[37]. Furthermore, global proteomics showed an increase in β-secretase (BACE1) abundance of 31% in 5 AD brains compared to 5 controls, while no changes occurred for either α-secretase or the sirtuin (SIRT) family (Supplementary Fig. 2c). Seyfried et al.[38] quantified a total of 2745 proteins in two regions (dorsolateral prefrontal cortex (FC, Brodmann Area 9) and precuneus (PC, Brodmann Area 7). The number that overlapped was about 2332 proteins (85.3%) compared with our data (4442 proteins from 10 controls and 10 AD, Brodmann area 44/45). The four disintegrin metalloproteinase (ADAM) family members identified in that paper were also identified in our proteome. The protein level of ADAM 10, 22, 23 neither changed in that paper nor our data, while ADAM11 showed a similar decrease in the two cases; SIRT2 and SIRT5 levels did not vary[38]. Further, protein levels of SIRTs do not necessarily reflect activity, which are often regulated by substrates and post-translational modifications.

Thus, succinylation of APP at K612 in AD may promote Aβ production by inhibiting cleavage by α-secretase. To test

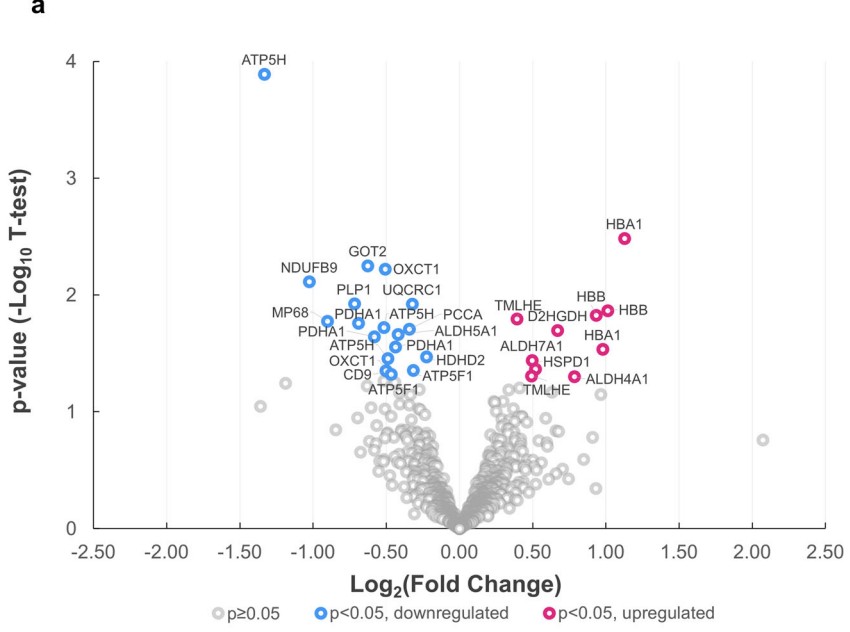

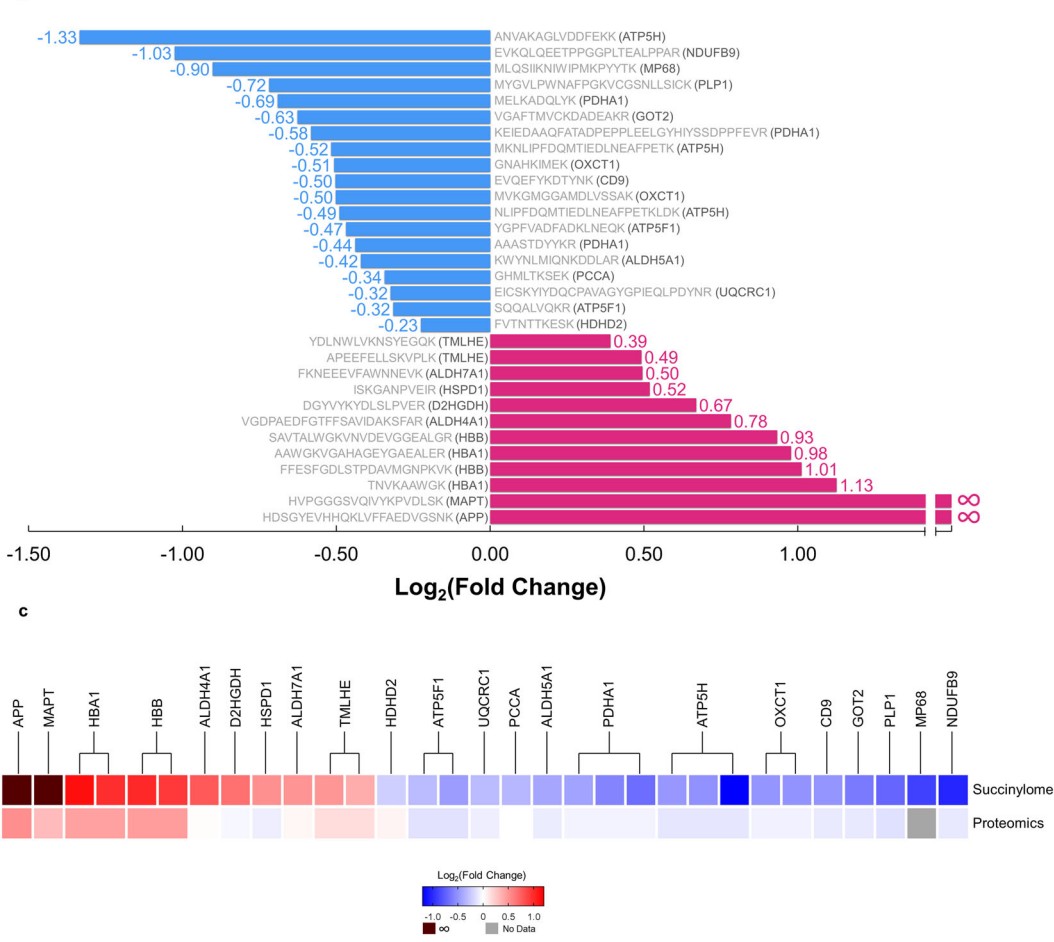

this hypothesis, synthetic peptides comprised of residues 6–29 in $A\beta_{42}$ (numbering with respect to the N-terminal aspartase of $A\beta_{42}$), which span the α-secretase cleavage site, with or without succinylation at K16 (corresponding to K612 in APP695), were assayed for α-secretase cleavage susceptibility. Recombinant human disintegrin metalloproteinase 10 (rhADAM10), the main basal α-secretase acting in the non-amyloidogenic processing of APP, cleaved the native (control) peptide (substrate) with 84% efficiency, whereas no cleavage of its succinylated counterpart was detectable following a 24 h incubation (Fig. 5e). Measurement of the two fragments that are produced by α-secretase activity confirmed a strong inhibition of catalysis at the α-secretase site (Supplementary Fig. 3c–g).

**Fig. 3 Comparison of the succinylome of brains from ten controls and ten patients with AD reveal many specific differences ($p < 0.05$, two-sided Student's $t$-test). a** Volcano plot of 932 brain protein peptide succinylation in controls and AD patients. The signal detection result shows the magnitude ($\log_2$Fold Change, $x$-axis) and significance ($-\log_{10}$ $p$-value, $y$-axis) for brain succinylation changes associated with AD. Each spot represents a specific succinylated peptide. Blue symbols to the left of zero indicate succinylated peptides that are decreased significantly while red symbols to the right of zero indicate succinylated peptides that are upregulated significantly in AD brains ($p < 0.05$, two-sided Student's $t$-test). **b** Peptides with significant differences in succinylation between control and AD brains. Decreases (blue bars) or increases (red bars) from the control succinylome are depicted as relative fold change. The sequence of the peptide and the name of the gene to which the peptides belong is noted for each bar. **c** Comparison of the AD-related changes in global proteome and succinylome. The succinylated peptides from the succinylome were clustered based on their proteins. For each protein, its relative fold change in succinylome and global proteome of AD cases versus controls is shown. Source data are provided as a Source Data file.

Residue K16 (K612 in APP) is critical for both aggregation and toxicity of $A\beta_{42}$[2,39]. $A\beta$ oligomers are widely regarded as the most toxic and pathogenic form of $A\beta$[40]. To assess whether succinylation can directly alter $A\beta$ oligomerization, aggregation of succinylated and non-succinylated $A\beta_{42}$ was determined by anti-$A\beta$ oligomer antibody NU-2[36] and electron microscopy (EM). After 24 and 48 h incubations, succinylated $A\beta$ appeared to undergo more robust oligomerization (Fig. 5f). Moreover, EM microscopy clearly revealed elevated levels of oligomeric, protofibrillar, and fibrillar $A\beta$[41] in the succinylation reaction mixture at t = 24 or 48 h (Fig. 5g). These data suggest that succinylation of K612 of APP may contribute to $A\beta$ oligomerization. Taken together, our findings suggest that succinylation of K678 might lead to early-onset and/or enhanced generation, oligomerization, and accumulation of $A\beta$, consistent with the effects of known pathogenic mutations at this site[37,42].

**Functional significance of succinylation of tau.** Tau has two important nucleating sequences that initiate the aggregation process: paired helical filament 6 (PHF6, residues 306–311) and PHF6* (residues 275–280) (Fig. 6a)[43,44]. PHF6* is located at the beginning of the second repeat (R2) and is only present in four-repeat tau isoforms, while PHF6 is located at the beginning of the third repeat (R3) and is present in all tau isoforms. Post-translational modifications within these two hexapeptide regions can alter protein function and may provide a critical link to pathological hallmarks of tauopathies. Acetylation of K280 of PHF6* in tau is a well-characterized modification that affects tau function[3], and has become a prognostic factor and a new potential therapeutic target for treating tauopathies. Until now, very little evidence of post-translational modifications within the PHF6 region have been uncovered. Tau succinylation on K311 within the PHF6 hexapeptide $^{306}$VQIVYK$^{311}$ was detected in nine of ten AD brain samples but was undetectable in all control samples (Fig. 6b). Previous studies indicate the removal of residue K311 in PHF6 abrogates fibril formation[45], but the structural and functional implications of K311 succinylation are unknown. Thus, exploring the influence of tau succinylation on K311 may be important as we seek to develop a comprehensive understanding of its biological importance.

To characterize tau succinylation in a transgenic mouse model of tauopathy, we used high-resolution confocal laser scanning microscopy to compare the presence of lysine succinylation with that of tau oligomers (T22)[46] and phospho-tau (AT8) in the hippocampus of wild-type and TgP301S mice (Fig. 6c, d and Supplementary Fig. 4a, b). At 4 months of age, the fluorescence signal of succinyl lysine was significantly increased in parallel with the oligomeric tau T22 (green) and phospho-tau AT8 (green). Similar to what we observed found in $A\beta$ deposits, 10-month-old wild-type and transgenic tau mice displayed a significant reduction in the levels of succinyl lysine in comparison to 4-month-old mice, thereby leading to an attenuated colocalization between succinylation epitopes and tauopathy epitopes.

The heparin-induced thioflavin S (ThS) tau aggregation assay was used to test the influence of tau succinylation at K311 on the ability of PHF6 to self-aggregate. PHF6* and K280-acetylated PHF6* (A-PHF6*) were also used as controls in parallel assays (Supplementary Fig. 4c). Surprisingly, at peptide concentration of 10 μM in the presence of 2.5 μM heparin, neither PHF6* nor A-PHF6* fibrillized during an 80 min incubation period. Although PHF6* is an initiation site for tau aggregation, its potency is much lower than that of PHF6[47], possibly explaining the observed lack of aggregation under these conditions. In contrast, PHF6 and K311-succinylated PHF6 (S-PHF6) fibrillized by 20 and 80 min, respectively (Fig. 7a). The aggregation of PHF6 was remarkably accelerated by the K311 succinylation. A substantial enhancement of PHF6-induced aggregation occurred even with a mixture containing 90% PHF6 and only 10% S-PHF6, suggesting that succinylated tau can promote aggregation of unmodified protein (Fig. 7a). Longer incubation time (24 h) with PHF6, S-PHF6, and a 90%/10% mixture was visualized by EM (Fig. 7b–d). All the reactions exhibited fibrils with a typical paired helical filament appearance. However, the succinylated peptide formed abundant short filaments, a feature of brain-derived Alzheimer PHFs[48–50], while unmodified PHF6 filaments are longer and sparser, morphologies more typical of recombinant tau peptide fibers (Fig. 7e, f). Thus, both ThS and EM results support an important role of succinylation in promoting pathological tau aggregation.

To understand the role of succinylation in tau function, tubulin polymerization was assessed using the tau K19 peptide, a 99-residue 3-repeat tau microtubule-binding domain (MBD) fragment (MQ244-E372), and succinylated K19 (Supplementary Fig. 4d–f). Native tau K19 promoted tubulin assembly as determined by increased light scattering at 350 nm, as previously reported[3,51]. Nevertheless, succinyl-CoA treated K19, which is succinylated at multiple lysine residues including Lys311, showed a complete suppression of tubulin assembly activity (Fig. 7g). These findings suggest that succinylation of tau leads to a loss of normal tau function in regulating microtubule dynamics.

Nuclear magnetic resonance (NMR) spectroscopy was used to investigate whether succinylation-induced loss of tau microtubule assembly resulted from a loss of tau-tubulin interactions. The binding of the tau MBD fragment K19 to a construct composed of two tubulin heterodimers stabilized by a stathmin-like domain (T2R), was monitored as described[52]. In the presence of T2R, a number of NMR HSQC resonances show a decreased intensity compared to corresponding resonances of matched samples of K19 in the absence of T2R (Fig. 7h). This reduced resonance intensity indicates an interaction between the corresponding K19 residue and the much larger T2R complex. The most highly attenuated resonances (intensity ratios < 0.2) within the MBD corresponded to residues ranging from positions 308–323, located in R2 of the MBD and included most of the PHF6 sequence. Succinylation of $^{15}$N-labeled K19 (Supplementary Fig. 4g–i) largely abrogated this interaction as observed by the increased intensity ratios across all residues compared to unmodified K19

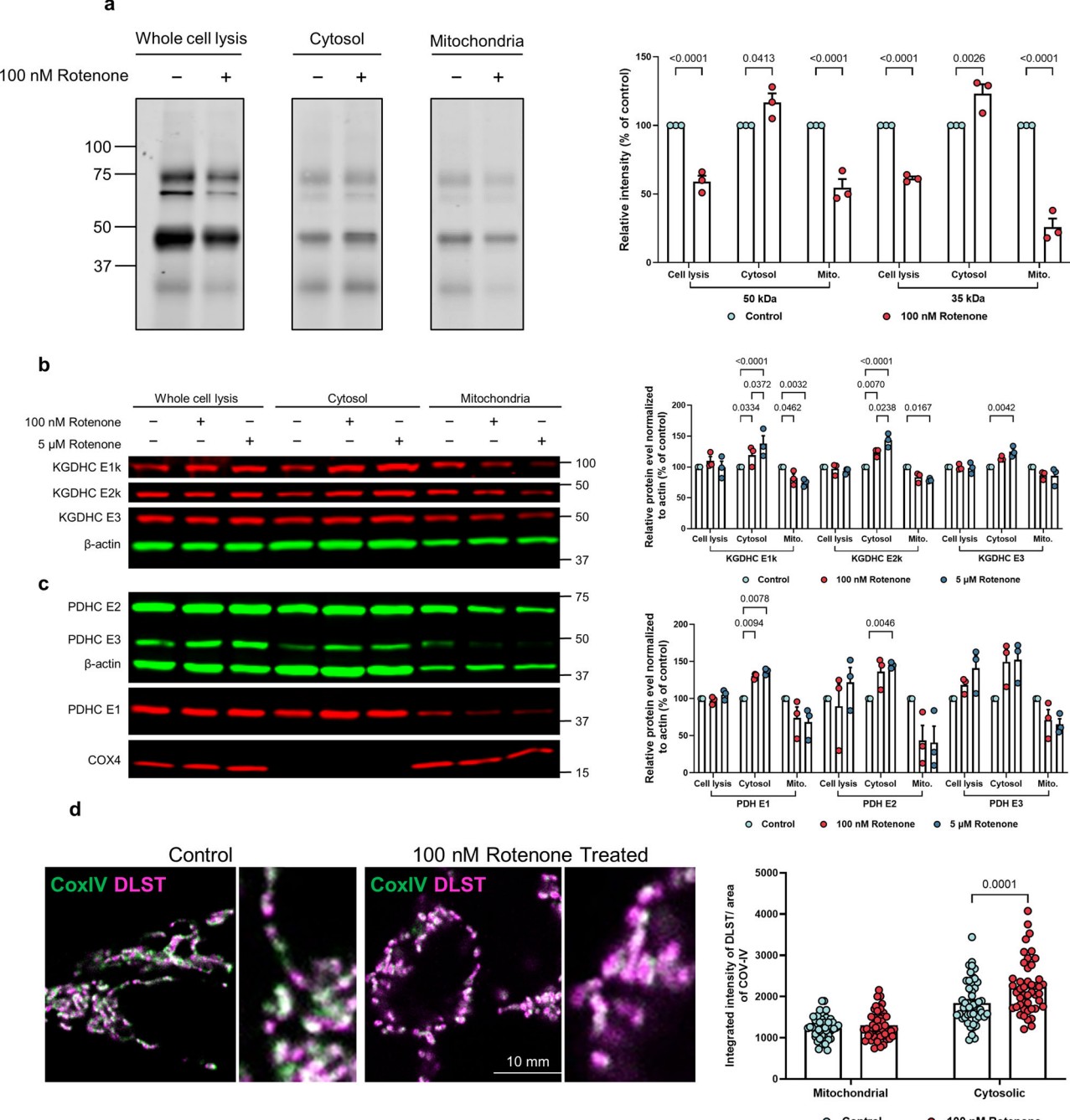

**Fig. 4 Impairing mitochondrial function altered succinylation and protein distribution in the whole cell as well as in the mitochondria and non-mitochondrial fractions. a** The effects of rotenone (100 nM/20 min) on succinylation in HEK cells. After separation, mitochondrial and non-mitochondrial fractions were immune-precipitated with anti-succinyl lysine antibody and separated by SDS-PAGE followed by western blotting. The data from three different replicate experiments were expressed as the mean with error bars from standard error of the mean (SEM) ($n = 3$ independent experiments, two-way ANOVA followed by Bonferroni's multiple comparisons test). **b** The effects of rotenone (100 nM or 5 μM/20 min) on the distribution of KGDHC protein between mitochondria and non-mitochondrial fractions. The data from three different replicate experiments were expressed as the mean with error bars from SEM ($n = 3$ independent experiments, two-way ANOVA followed by Tukey's multiple comparisons test). **c** The effects of rotenone (100 nM, 5 μM/20 min) on the distribution of PDHC protein between mitochondria and non-mitochondrial fractions. The data from three different replicate experiments were expressed as the mean with error bars from SEM ($n = 3$ independent experiments, two-way ANOVA followed by Tukey's multiple comparisons test). **d** Rotenone induces release of DLST into cytoplasm. In the control conditions, DLST (magenta) was concentrated inside mitochondria defined by COX-IV labeling (green). After 1 h of 100 nM Rotenone treatment, additional DLST labeling was found in the cytoplasm. Inserts on the right are magnified regions. Magenta: DLST; Green: COX-IV; Error bars represent SEM deviation from the mean ($n = 98$ fields from 19 dishes, two-way ANOVA followed by Bonferroni's multiple comparisons test). Source data are provided as a Source Data file.

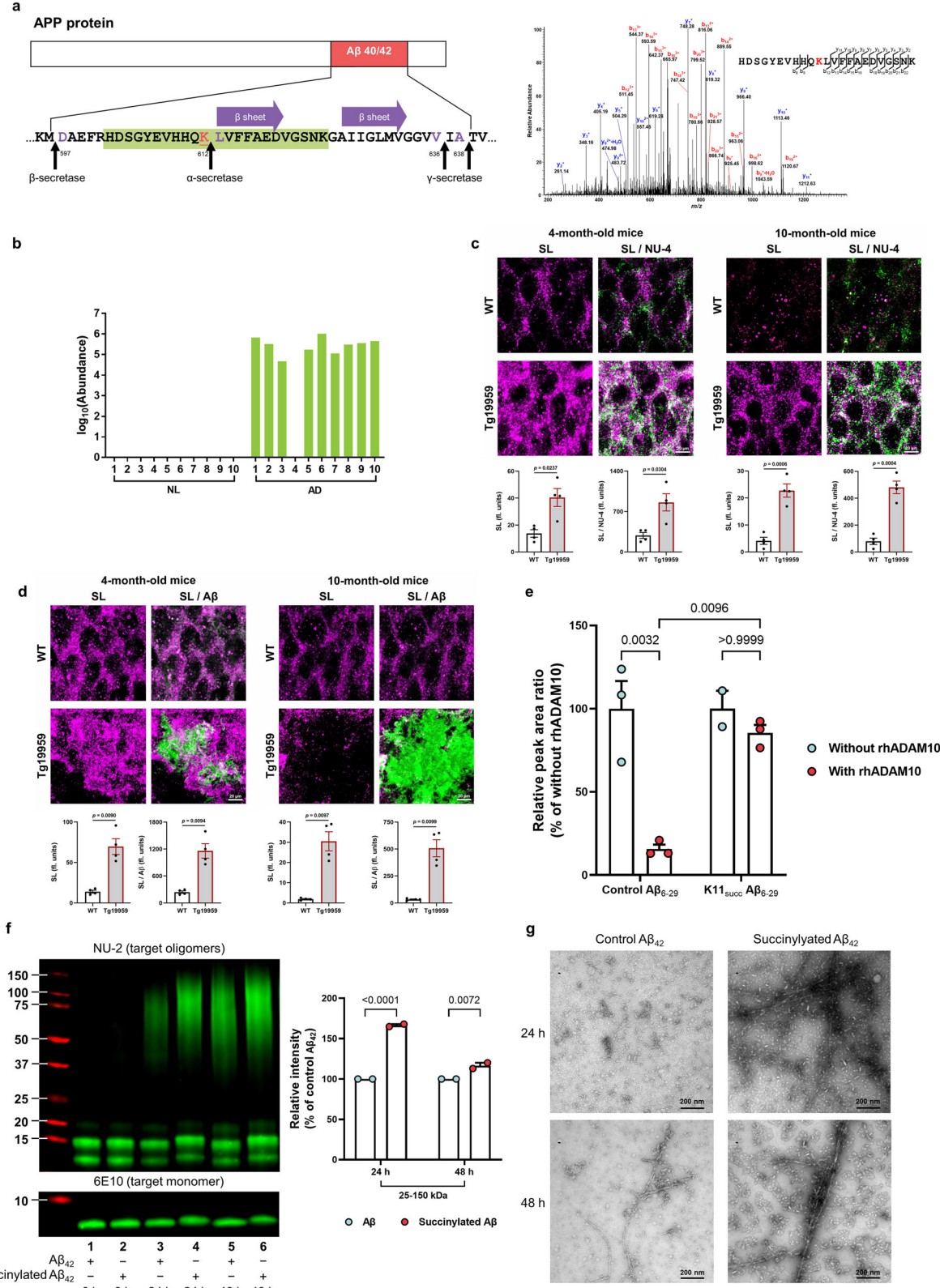

(Fig. 7i). These results indicate that succinylation of K19 negatively modulates the interaction with the T2R tubulin tetramer.

To establish whether succinylation of K311 specifically was sufficient to decrease tau-tubulin interactions, [1]H STD NMR was employed to analyze the tubulin interactions of a tau peptide (residues 296–321) that was previously shown to comprise a high

affinity microtubule-binding motif within tau[53–55]. STD signals were observed for unmodified tau peptide (296–321) in the presence of tubulin (Fig. 7j), as previously reported[55], indicative of binding. Succinylation of residue K311 within the tau peptide (296–321) resulted in a dramatic loss of STD signals (Fig. 7k), indicating that K311 succinylation results in a significantly decreased binding affinity of this microtubule-binding tau peptide

**Fig. 5 Succinylation occurs uniquely on APP from AD patients, in early stages of plaque formation in mouse models and disrupts APP processing.**
**a** Location and identity of succinylation K612 near the Aβ region. Residues are numbered according to APP695 sequence. Purple amino acids refer to α- or β- or γ- cleavage sites. The red underlined lysine refers to succinylated K612. Purple arrow represents the two central strands of the β-sheet (Leu613-Asp619 and Ala626-Val632). Green highlights the peptide identified in the MS. MS2 spectrum of m/z 686.5744[4+] leads to confident identification of a succinylated peptide from APP protein with K612 succinylation site being highlighted in red text. **b** Abundance of succinylation K612 found in brains from 10 controls and 10 AD patients. Data transformed by $\log_{10}$ (abundance) for normalization purposes and to facilitate presentation. **c** Confocal microscope analysis of the colocalization of succinylation (magenta) and amyloid oligomers (green) in the hippocampal region of 4 and 10-month-old Tg19959 or wild-type (WT) mice ($n = 4$ per each group, two-tailed Student's $t$-tests). **d** Brain sections were stained against Aβ plaques (green) and succinyl lysine (magenta). Quantitative analysis of the colocalization of succinylation and plaque pathology in the hippocampus of 4 and 10-month-old Tg19959 or WT mice ($n = 4$ per each group, two-tailed Student's $t$-tests). **e** Succinylation blocks α-cleavage. Peptides were incubated for 24 h with or without rhADAM10. Peak area ratio values were calculated and are shown relative to corresponding controls without rhADAM10. Each sample was run in triplicate and data were expressed as the mean with SEM ($n = 3$ biologically independent samples, two-way ANOVA followed by Bonferroni's multiple comparisons test; except for one sample from the group of succinylated peptide without rhADAM10 was damaged). **f** Western blot analysis of succinylated and control Aβ42 from aggregation assay showed that the succinylation generates more oligomerized Aβ even after a long incubation. The data were expressed as the mean with SEM ($n = 2$ biologically independent samples, two-way ANOVA followed by Bonferroni's multiple comparisons test). **g** Two timepoints from aggregation assay were analyzed by negative-staining electron microscopy. This experiment was performed once. Source data are provided as a Source Data file.

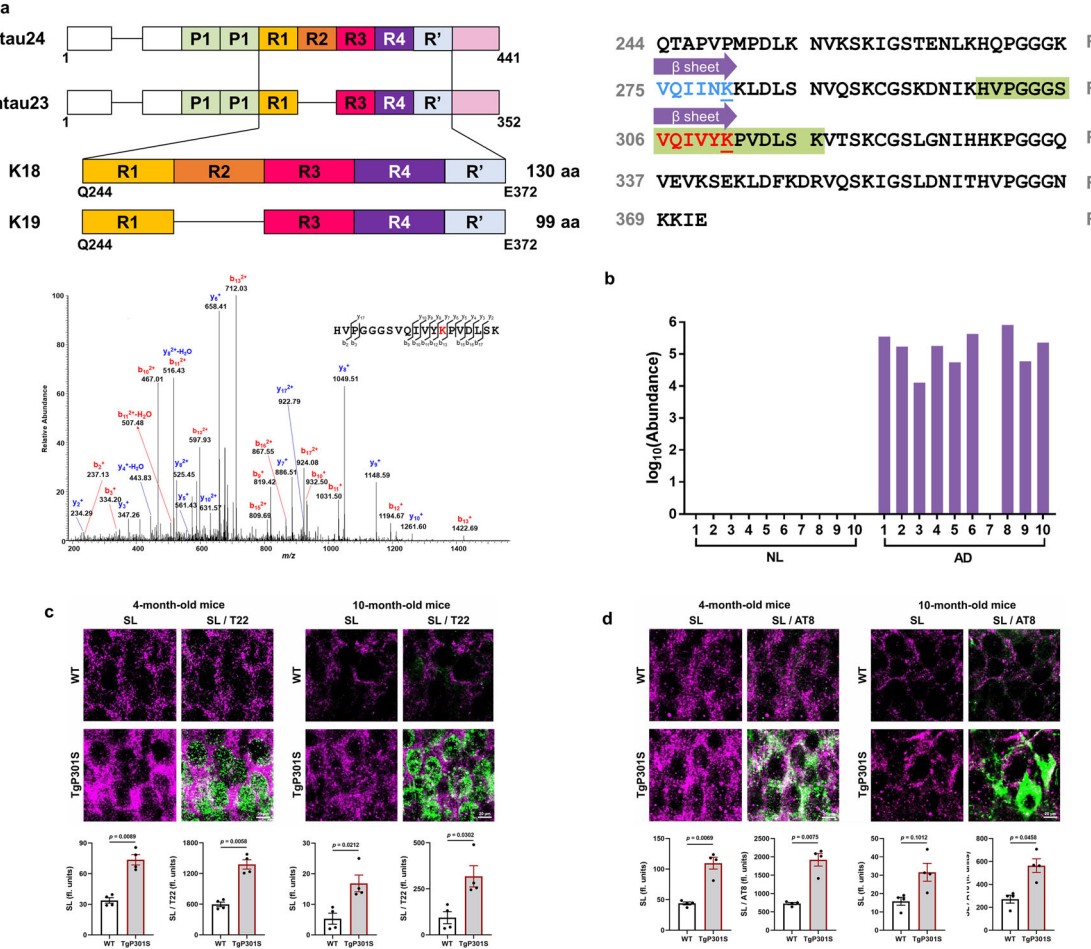

**Fig. 6 The unique succinylation of K311 on tau in brains from patients with AD. a** Domain structure of tau and the location of succinylation K311. The diagram shows the domain structure of htau23 and 24, which contain three and four repeats, respectively. The constructs K18 and K19 comprise four repeats and three repeats, respectively. Residues are numbered according to tau441 sequence. Purple arrow represents the two central strands of the β-sheet (PHF6*: Val275-Lys280, highlighted in blue, the blue underlined lysine refers to acetylated K280; PHF6: Val306-Lys311, highlighted in red, the red underlined lysine refers to succinylated K311). Green highlights the peptide identified by MS. MS2 spectrum of m/z 694.0407[3+] leads to confident identification of a succinylated peptide from tau protein with K311 succinylation site being highlighted in red text. **b** Abundance of succinylation K311 found in brains from ten controls and ten patients with AD. Data transformed by $\log_{10}$ (abundance) for normalization purposes and to facilitate presentation. **c** High-resolution images acquired using confocal laser microscopy display the colocalization of succinylation(magenta) and tau oligomers(green) in the hippocampus of 4-month-old and 10-month-old Tg19959 or WT mice ($n = 4$ per each group, two-tailed Student's $t$-tests). **d** Fluorescence micrographs obtained from the hippocampus of 4-month-old and 10-month-old Tg19959 or WT mice show the colocalization between succinylation(magenta) and NFTs (green) ($n = 4$ per each group, two-tailed Student's $t$-tests). Source data are provided as a Source Data file.

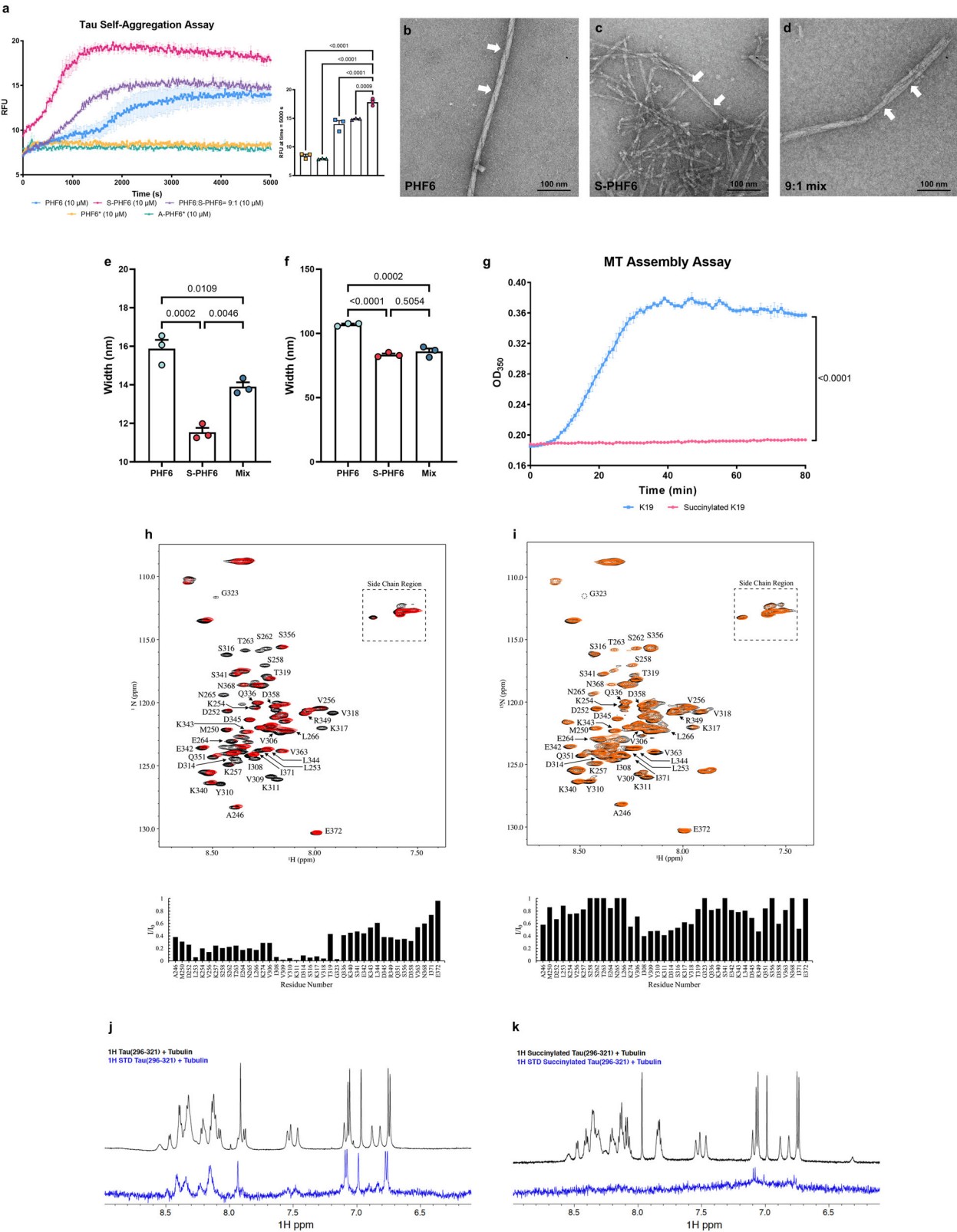

for tubulin. The recently reported structure of tau repeat 2 bound to microtubules shows that K280, the R2 equivalent of K311, lies along the microtubule surface[54]. K280/K311 have their positively charged amino group in close proximity to residue E415 of α-tubulin (Supplementary Fig. 4j). Therefore, it seems possible that

succinylation at K311 might result in an electrostatic clash between the negatively charged succinyl group and the glutamate residue. A decreased affinity of K311-succinylated tau for tubulin and/or microtubules could contribute to the progression of tau pathology in AD.

**Fig. 7 The succinylation of K311 on tau promotes AD like features in tau pathology. a** Succinylation promotes self-aggregation of tau. Tau peptides (10 μM) were in presence of 2.5 μM heparin: PHF6 (blue squares), S-PHF6 (magenta circles), PHF6:S-PHF6 = 9:1 (purple triangle), PHF6* (yellow square), A-PHF6* (lake green triangle). Error bars represent SEM deviation from the mean. All statistical analysis was implemented at time = 5000 s ($n = 3$, one-way ANOVA followed by Tukey's multiple comparisons test). Experiments repeated three times with similar results. **b–d** Negative stain electron microscopy of in vitro polymerized PHFs after 24 h incubation. **b** 50 μM PHF6; **c** 50 μM S-PHF6; **d** 50 μM mixture (PHF6:S-PHF6 = 9:1). White arrows denote paired helical filaments. $N = 3$ independent biological replicates. Scale bar is 100 nm. **e, f** The width and height of the fiber helix found in polymerized PHFs after 24 h incubation in vitro. Error bars represent SEM deviation from the mean ($n = 3$ different fields per group, one-way ANOVA followed by Tukey's multiple comparisons test). **g** Inhibition of assembly reaction of K19 and microtubules by succinylation of K19. Incubations (30 min) were with 30 μM succinylated K19 (magenta circles) or non-succinylated K19 (blue Squares). Error bars represent SEM deviation from the mean. All statistical analysis was implemented at time = 80 min ($n = 3$, two-way ANOVA followed by Bonferroni's multiple comparisons test). Experiments repeated three times with similar results. **h, i** Succinylation of K19 weakens its interactions with T2R. [1]H,[15]N HSQC spectra were recorded for unmodified and succinylated K19 in the absence (black) and in the presence (red for unmodified K19, orange for succinylated K19) of T2R. Unmodified or succinylated [15]N K19 spectra (assignments for well-resolved residues as indicated) exhibit intensity loss for multiple residues including Ile[308], Val[309], Tyr[310], Lys[311] in the presence of T2R. **j, k** Succinylation of K311 weakens the interactions of tau peptide (296–321) with tubulin. Comparison of 1D [1]H spectra (black) and saturation transfer difference NMR spectra (blue) of unmodified tau peptide (296–321) or K311-succinylated tau peptide (296–321) in the presence of 20 μM tubulin. Source data are provided as a Source Data file.

## Discussion

Our study provides a system level view of the human brain succinylome as a marker of metabolic activity, particularly in mitochondria, and reveals a dramatic alteration of succinylation in AD brain. Our results raise the possibility that succinylation may provide a link between the signature metabolic reductions and proteinopathies in AD. We show that changes in protein succinylation, as a molecular signal, correlate with altered cerebral metabolic function in AD as the disease progresses. Other PTMs, such as ubiquitination, acetylation, and phosphorylation, have been recently shown to affect Aβ clearance[56,57] and tauopathy[56–59], thereby contributing to progression of the severities of those proteinopathies. It is not perhaps surprising that succinylation would have such an effect since it increases the size of the lysine side chain considerably and could lead to steric hindrance of intermolecular interactions. Moreover, as noted, succinylation reversing the net charge of the lysyl side chain (Supplementary Fig. 4j).

The mechanisms and control of both nonenzymatic and enzymatic succinylation by cellular succinyltransferases and desuccinylases are unknown. Our data clearly demonstrate that impairing mitochondrial function decreases mitochondrial succinylation and promotes succinylation of specific non-mitochondrial proteins by altering the distribution of succinyltransferases from the mitochondria to cytosol. A precedent for this concept is provided by results showing that translocation of the DLST subunit of KGDHC to the nucleus increases histone succinylation[35]. Rotenone induces translocation of PDHC from mitochondria to other compartments[60]. The decline in succinylation of mitochondrial proteins appears to be due, at least in part, to a failure in maintaining succinylation levels. This raises the possibility that activators of desuccinylases (e.g., sirtuins) and/or interventions that elevate levels of NAD[+] should be considered. The large increase in succinylation in 4-month-old Tg19959 mice in which abnormal mitochondrial function in AD promotes the release of KGDHC and subsequent increases succinylation is consistent with our hypothesis that succinylation status may represent a therapeutic opportunity.

We showed that transgenic mouse strains of either tauopathy or amyloidosis phenotype, exhibit widespread increases in lysine succinylation at 4 months of age, which is not exclusive to tau and APP but parallels the early appearance of these proteinopathies. This suggests that each transgene is altering common processes (e.g., mitochondria/metabolism) in addition to tau or APP processing. Metabolism is altered even in embryonic cultures of mouse models of AD[61]. The data in Fig. 4 demonstrate that disrupted mitochondrial function increases succinyl transferase in the cytosol. Indeed, the widespread succinylation in both models

provides further evidence of that possibility. Interestingly, a pharmacological increase in vitamin B1 (a key vitamin in metabolism) significantly reduces Aβ burden[62] and tauopathy[63] in mice and also showed encouraging results in AD patients[64], suggesting these fundamental processes are critical even in mice genetically engineered to create the pathologies. A more precise interpretation requires knowing which proteins are succinylated since the human brain succinylome probably involves hundreds of succinylated proteins.

Decreases in succinylation status at 10 months are observed in the brains of mouse models of both amyloidosis and tauopathy, suggesting that this reduction is not an artefact of staining protocols or efficiencies. The changes in succinylation status that we observed between wild-type and transgenic mice and between 4- and 10-month-old mice were consistent across comparisons with two different Aβ antibodies, with two different tau antibodies, and also between the different transgenic mice, provide further evidence that pathology-related changes in succcinylation status are unlikely to be attributable to interindividual variability in tissue processing and/or staining.

While proteins in addition to tau or APP are succinylated, APP and tau succinylation status increase in brains from AD patients, which suggests that increased tau and APP succinylation may play a role in the development of AD pathology. Intriguingly, lysine-succinylation levels decrease in 10-month-old mice over 4-month mice, while both amyloid aggregation and tauopathy continued to increase. This change is not likely to be a technical artifact. All sections were stained at the same time under the same conditions (solutions, washing, temperature, antibody preparation, etc.) and analyzed under identical experimental settings. In addition, perusal of the first column of panel in the Supplementary Figs. 3 and 4 show that the immunoreactivity of MAP2 (a neuronal marker, cyan) does not change during aging, either in WT or Aβ/tau transgenic mice. The decline in succinylation may reflect either desuccinylation processes, or sequestration of succinylated sites away from labeling antibodies. Notably, both K16 in Aβ and K311 in tau are buried in the structured core of their respective aggregated forms[65,66]. The decrease in the association between succinylation and pathology at 10 months may be due to results at least in part incorporation of succinylated sites inside aggregated species, preventing detection by immunohistochemistry. However, based on the current data, it is not possible to rule out alternative explanations, including potential changes in metabolism leading to desuccinylation reactions that may be related, or unrelated, to the progression of pathology and disease. Importantly, some precedent is provided by reports in which tau acetylation at residue K280 also peaks and decays during the

course of tangle formation and cell death, leading to the suggestion that this epitope is either masked in PHFs or else is subjected to deacetylation in later stages of aggregate maturation[67]. An adequate explanation requires a complete accounting of which proteins are involved (i.e., a complete mouse brain succinylome at multiple ages) and knowing which proteins are incorporated into deposits in both humans and mouse models. The mitochondrial succinylome in human brain tissues was significantly reduced in AD while succinylation of APP and tau was increased. Despite these remaining questions, our results suggest that the modification of metabolism in disease may lead to critical succinyl-mediated modifications of extramitochondrial proteins including APP and tau leading to aggregation and deposition. Preventing APP and tau succinylation and/or increasing mitochondrial succinylation may provide novel therapeutic targets for the prevention and/or treatment of AD and associated pathologies[24].

Lysines are highly modified residues. Understanding the relationship(s) of succinylation to the other PTM(s) is critical to a complete understanding of its role in AD pathology. A direct comparison is difficult because each PTM requires a different enrichment strategy. Lysine311 of tau has been reported to be subjected to ubiquitination, dimethylation, and acetylation in transgenic mice[68]. Our experiments raise the question of whether precise changes in mitochondria are required to alter modification of specific proteins. Succinylation appears directly linked to KGDHC and mitochondria. Whether other post-translational modifications are also linked to mitochondrial dysfunction remains to be determined.

Overall, our data report the human brain succinylome and its implications for mitochondrial function as well as for the molecular pathogenesis of amyloidosis and tauopathy, two of the cardinal proteinopathies of AD. We provide a rich resource for functional analyses of lysine succinylation and for the dissection of metabolic networks in AD. The current studies also lay the foundation for future investigation into the crosstalk between different PTMs, including acetylation, phosphorylation, ubiquitination and succinylation associated with AD and tau pathology. The discovery that succinylation links mitochondrial dysfunction to amyloidosis and tauopathy may provide new molecular diagnostics as well as potential targets for therapies. Since both succinylated Aβ and tau are closely associated with disease state, future investigations may reveal additional succinylated proteins that are associated with AD or other neurodegenerative diseases.

## Methods

All experiments were approved by the Animal Care and Use Committee of Weill Cornell Medicine. All human brains used were obtained from Neurobiobank and Mount Sinai School of Medicine. All brain tissues were procured, stored, and distributed according to applicable state and federal guidelines and regulations involving consent, protection of donor anonymity. Consistent with institutional policies and procedures as well as requirements of New York State Department of Health for Tissue Bank Operations (License number NA083), written consent for postmortem brain donation for diagnosis and research use was obtained from the legal next of kin of each donor. The deidentified specimens used in this research were exempt from Institutional Review Boards (IRB) review.

**Human brain tissue samples**. All brain tissues from Broca's area (BM-44/45, frontal lobe) were from the NIH Neurobiobank and Mount Sinai School of Medicine. All patient information including diagnosis, clinical dementia rating (CDR), age, sex, postmortem interval (PMI) disease status, and neuropathological diagnostic criteria (including mean plaques, and Braak staging) are detailed in Supplementary Data 1. Neuropathological evaluation of amyloid plaque distribution was based on the Consortium to Establish a Registry for Alzheimer's disease (CERAD) criteria, while the extent of spread of neurofibrillary tangle pathology was performed according to the Braak staging system.

**Quantitative proteomics and succinylome**. The experimental strategy and workflow are present in Fig. 1. The analysis of two independent batches of brains

tested for replicability. In each batch, samples from five patients and five controls were analyzed in parallel for TMT-based comparative proteomics and label-free quantitation of their succinylome after enrichment of succinylated peptides. After a complete analysis of the first batch, the second batch was analyzed. The mass spectrometry proteomics data have been deposited to the ProteomeXchange[69,70] Consortium via the PRIDE[71] partner repository with the dataset identifier PXD015124.

**Protein extraction, digestion, and TMT10-plex labeling**. These procedures followed the PTMScan Succinyl-Lysine Motif [Succ-K] kit protocol (Cat # 13764 Cell Signaling Technology, Inc., Danvers, MA, USA). Brain tissue powders were denatured in 20 mM HEPES pH = 8.0, 9 M Urea, 1 mM Sodium orthavanadate, 2.5 mM sodium pyrophosphate, 1 mM β-glycerophate, and then homogenized with a Dounce homogenizer. After centrifugation at $20,000 \times g$ for 15 min at room temperature (r.t.), the supernatant was into a new tube. The protein concentration for each sample was determined by BCA assay using BSA as the standard.

Further processing of the proteins was then performed according to TMT Mass Tagging Kits (Thermo Fisher Scientific, Waltham, MA, USA) and Reagents protocol (http://www.piercenet.com/instructions/2162073.pdf) with a slight modification[72,73]. A total of 50 µg protein of each sample was reduced with 10 mM DTT for 1 h at 34 °C, alkylated with 50 mM iodoacetamide for 30 min in the dark and then quenched with 38 mM dithiothreitol (DTT). Each sample diluted with 50 mM tetraethylammonium bromide (TEAB) to a final concentration of 1 M Urea. Each sample was digested with 5 µg trypsin (1:10 w/w) for 18 h at 35 °C. Samples were then dried down in speed vac and reconstituted to a final volume of 100 µL in 50 mM TEAB prior to labeling. The Tandem Mass Tag™ (TMT™) 10-plex labels (dried powder) were reconstituted with 50 µL of anhydrous acetonitrile prior to labeling and added with 1: 2 ratio to each of the tryptic digest samples for labeling over 1 hour at r.t.. The peptides from the 10 samples (5 controls and 5 AD cases) were mixed each tag respectively with 126-tag, 127N-tag, 127C-tag, 128N-tag, 128C-tag, 129N-tag, 129C-tag, 130N-tag, 130C-tag, and 131-tag. The order of labeling each of the 10 samples by TMT10-plex was randomized. Same labeling as above was also conducted for the second sets of additional 10 samples. After checking label incorporation using Orbitrap Fusion (Thermo Fisher Scientific, San Jose, CA, USA) by mixing 5 µL aliquots from each sample and desalting with SCX ziptip (Millipore, Billerica, MA), the 10 digested samples were pooled together. The pooled peptides were evaporated to 200 µL and subjected to cleanup by solid phase extraction (SPE) on Sep-Pak Cartridges (Waters, Milford, MA). The eluted tryptic peptides were evaporated to dryness, and ready for the first dimensional LC fractionation via a high pH reverse-phase chromatography as described below.

**High pH reverse-phase (hpRP) fractionation**. The hpRP chromatography was carried out using a Dionex UltiMate 3000 HPLC system with the built-in micro fraction collection option in its autosampler and UV detection (Thermo Fisher Scientific, Sunnyvale, CA, USA) as reported previously[72,73]. Specifically, the TMT10-plex tagged tryptic peptides were reconstituted in buffer A (20 mM ammonium formate pH = 9.5 in water), and loaded onto an XTerra MS C18 column (3.5 µm, 2.1 × 150 mm) from Waters (Waters Corporation, Milford, MA, USA) with 20 mM ammonium formate ($NH_4FA$), pH = 9.5 as buffer A and 80% ACN/20% 20 mM $NH_4FA$ as buffer B. The LC was performed using a gradient from 10 to 45% of buffer B in 30 min at a flow rate 200 µL/min. Forty-eight fractions were collected at 1 min intervals and pooled into a total of 10 fractions based on the UV absorbance at 214 nm and with multiple fraction concatenation strategy[72]. Each of the 10 fractions was dried and reconstituted in 125 µL of 2% ACN/0.5% FA for nanoLC-MS/MS analysis.

**Protein digestion and enrichment of succinylated peptides by Anti-succiK antibody beads**. For global succinylome analysis, 1 mg of proteins of each sample was reduced with 10 mM DTT for 1 h at 34 °C, alkylated with 50 mM iodoacetamide for 1 h in the dark and then quenched by additional of 38 mM DTT. Each sample was diluted with 50 mM TEAB to a final concentration of 1 M Urea. Each sample was digested with 55 µg trypsin (1:18 w/w) for 18 h at 35 °C. The digests were cleaned up with Bond Elute C18 100 mg/mL cartridge (Agilent) and peptides were eluted with 50% ACN-0.1% TFA and dried down by a SpeedVacuum. Subsequent enrichment for succinylated peptides was conducted using a PTMScan® Succinyl-Lysine Motif [Succ-K] kit (Cat # 13764, Cell Signaling Technology, Inc., Danvers, MA, USA) following the vendor's recommended procedures. Specifically, the peptides from each sample were reconstituted in 350 µL immunoaffinity purification (IAP) buffer and transferred to the vial containing 20 µL equilibrated Succ-K motif antibody beads, incubated on a vortex mixer at 4 °C for 2 h. After centrifugation at $2000 \times g$ for 30 s, the beads were washed twice with 250 µL IAP buffer and three times by water. Finally, the enriched peptides were eluted three times with 55 µL of 0.15% TFA. The eluted fractions were pooled together, dried down and reconstituted in 22 µL of 0.5% formic acid (FA) for subsequent label-free quantitative analysis by nanoscale LC-MS/MS.

**Nanoscale reverse-phase chromatography and tandem MS (nanoLC-MS/MS)**. The nanoLC-MS/MS analysis was carried out using an Orbitrap Fusion (Thermo Fisher Scientific, San Jose, CA, USA) mass spectrometer equipped with a nanospray

Flex Ion Source using high energy collision dissociation (HCD) similar to previous reports[74,75] and coupled with the UltiMate 3000 RSLCnano (Dionex, Sunnyvale, CA, USA). Each reconstituted fraction (4 µL = 0.8 µg for global proteomics fractions and 20 µL for enriched SuccK samples) was injected onto a PepMap C18 RP nano trap column (3 µm, 100 µm × 20 mm, Dionex) with nanoViper Fittings at 20 µL/min flow rate for online desalting and then separated on a PepMap C18 RP nano column (3 µm, 75 µm × 25 cm), and eluted in a 120 min gradient of 5–35% acetonitrile (ACN) in 0.1% formic acid at 300 nL/min., followed by a 8 min ramping to 95% ACN-0.1% FA and a 9 min hold at 95% ACN-0.1% FA. The column was re-equilibrated with 2% ACN-0.1% FA for 25 min prior to the next run. The Orbitrap Fusion is operated in positive ion mode with nanospray voltage set at 1.6 kV and source temperature at 275 °C. External calibration for FT, IT and quadrupole mass analyzers was performed. The instrument was operated in data-dependent acquisition (DDA) mode using FT mass analyzer for one survey MS scan for selecting precursor ions followed by 3 second "Top Speed" data-dependent HCD-MS/MS scans in Orbitrap analyzer for precursor peptides with 2–7 charged ions above a threshold ion count of 10,000 with normalized collision energy of 38.5%. MS survey scans at a resolving power of 120,000 (fwhm at $m/z$ 200), for the mass range of $m/z$ 400–1600 with AGC = 3e5 and Max IT = 50 ms, and MS/MS scans at 60,000 resolution with AGC = 1e5, Max IT = 120 ms and with Q isolation window (m/z) at 1.6 for the mass range $m/z$ 105–2000 in TMT10-plex fractions.

For label-free SuccK peptides analysis, one MS survey scan was followed by 3 second "Top Speed" data-dependent CID ion trap MS/MS scans with normalized collision energy of 30%. Dynamic exclusion parameters were set at 1 within 45 s exclusion duration with ±10 $ppm$ exclusion mass width. Ten samples for both control and AD cases were analyzed in Orbitrap in random order for data acquisition. All data are acquired under Xcalibur 3.0 operation software and Orbitrap Fusion Tune 2.0 (Thermo Fisher Scientific, Waltham, MA, USA).

**Data processing, protein identification, and data analysis.** All MS and MS/MS raw spectra from each set of TMT10-plex experiments were processed and searched using the Sequest HT search engine within the Proteome Discoverer 2.2 (PD 2.2, Thermo Fisher Scientific, Waltham, MA, USA). The *Homo sapiens* NCBI UniprotKB.fasta database containing 20,153 entries downloaded on October 17, 2016 were used for database searches. The default search settings used for 10-plex TMT quantitative processing and protein identification in PD 2.2 searching software were: two mis-cleavage for full trypsin with fixed carbamidomethyl modification of cysteine, fixed 10-plex TMT modifications on lysine and N-terminal amines and variable modifcations of methionine oxidation, deamidation on asparagines/glutamine residues and protein N-terminal acetylation. The peptide mass tolerance and fragment mass tolerance values were 10 ppm and 0.02 Da, respectively. Identified peptides were filtered for maximum 1% FDR using the Percolator algorithm in PD 2.2 along with additional peptide confidence set to high. The TMT10-plex quantification method within Proteome Discoverer 2.2 software was used to calculate the reporter ion abundances that were corrected for the isotopic impurities. Both unique and razor peptides were used for quantitation. Signal-to-noise (S/N) values were used to represent the reporter ion abundance with a co-isolation threshold of 50% and an average reporter S/N threshold of 10 and above required for quantitation spectra to be used. The S/N values of peptides, which were summed from the S/N values of the PSMs, were summed to represent the abundance of the proteins. For relative ratio between the two groups, normalization on total peptide amount for each sample was applied.

For label-free quantitative data analysis of succinylated peptides, fragment ion tolerance 0.5 Da was used for the ion trap analyzer and an additional succinylation on Lys residue with mass shift (+100.0160 Da) was specified as variable modifications. In addition, methionine oxidation (+15.995 Da), acetylation (+42.011) on N-terminal proteins and deamidation (+0.984 Da) on asparagines/glutamine were also setup as variable modifications. For each relative ratio of succinylated peptides/sites, no normalization was applied. The search result including ratio, *p*-value, succinylated peptide abundance for each sample was output to Microsoft Excel software (MicroSoft Office 365) for further data analysis. The threshold of succinylation localization score was set at PSM grouper node of consensus workflow in Proteome Discoverer (PD) 2.2 database searching software, and succinylation localization scores for each succinylated peptide must be greater than 75 and it lies in between 75–100. In combination with threshold score (≥75) for succinylation localization, falsely localized succinylated peptides were further filtered out under peptide validator node in consensus workflow where q values and PEPs are validated for available PSMs and assigned the PSMs confidences based on the user defined target FDRs in percolator node (Target FDR-0.01). In addition, data analysis for all identified succinylated peptides indicated that the average number of missed cleavages for each succinylated peptides = 1 (Supplementary Data 7). This is not surprising, as lysine succinylation would prohibit trypsin from cut the modification site creating one missed cleavage. As a result, we found that over 90% of succinylated peptides being identified were equivalent to 0 miss-cleavage and 10% contained 1 miss-cleavage site in our data, which is consistent with what we observed in our regular global proteomics.

**α-cleavage assay.** Synthetic C- and S-Aβ$_{6-29}$ peptides were purchased from GenScript USA Inc. (Piscataway, NJ, USA). Recombinant human ADAM 10 protein (rhADAM10) was purchased from R&D Systems, Inc. (Minneapolis, MN,

USA). The C-Aβ$_{6-29}$ and S-Aβ$_{6-29}$ peptides were dissolved in DMSO to reach a final concentration of 1 mM, respectively. rhADAM10 was diluted with assay buffer (25 mM Tris, 2 µM ZnCl$_2$, 0.005% (w/v) Brij-35, pH = 9.0) to reach a final concentration of 0.1 µg/µL. C-Aβ$_{6-29}$ and S-Aβ$_{6-29}$ peptide solution (10 µL) was added to 170 µL assay buffer together with 20 µL rhADAM10 or assay buffer, briefly mixed and then incubated at 37 °C for different time intervals with continuous shaking. Reactions were stopped and saved at −80 °C.

The assay reactions were conducted in triplicate at each timepoints for C-Aβ$_{6-29}$ and S-Aβ$_{6-29}$ peptides. The reaction samples were cleaned up by SPE as described previously. The eluted peptides with 50% ACN-0.1% FA were dried down and reconstituted in 100 µL of 0.1% FA with 2% acetonitrile and spiked with 0.4 pmol/µL Angiotensin I peptide (used as an Internal Standard). The samples were analyzed by LC-MS.

The resulting samples were subject to quantitative analysis for C-Aβ$_{6-29}$ peptide, S-Aβ$_{6-29}$ peptide and their respective cleaved product peptides (fragment 1, fragment 2 and fragment 3) by LC-MS/MS using an Exion LC coupled with the X500B Q-TOF mass spectrometer (SCIEX, Framingham, MA, USA). Each of 24 samples (6 µL) were injected onto a Luna C$_{18}$ column (150 × 2.1 mm i.d., 3 µm; Phenomenex) and eluted with 5–40% B (solvent B = 99.9% ACN-0.1% FA; solvent A = 0.1% FA) at 200 µL/min in a 6-min gradient, followed by ramp up to 70% B in 1.5 min and 95% B in 0.5 min. The gradient at 95% B was holding for 2 min prior back to 5% B in 0.5 min for subsequent 5.5 min of column equilibration. The X500B instrument was calibrated using the integrated calibrant delivery system (CDS). The source temperature (350 °C), spray voltage (4.5 kV) and gas conditions (with 30 arbitrary units for gas 1, gas 2 and curtain gas) were initially optimized and obtained using a positive TOF-MS survey scan by tee-in with infusion of 5 pmol/µL Angiotensin I. Quantitative data were acquired under MRM-HR mode optimized for the C-Aβ$_{6-29}$ and S-Aβ$_{6-29}$ peptides using a Guided MRM-HR module. The detected MRM transitions used for quantitation with corresponding precursor ion dependent parameters were given in Supplementary Fig. 3f. For quantation of each peptide, the 2 most highest intensity $m/z$ precursors ions with having 2–4 charges and 2 singly charged product ions (whose $m/z$ masses are larger than their precursor m/z) generated for each precursor $m/z$ were selected for MRM-HR data acquisition of all 24 samples using Sciex OS 1.3 software. The final MRM-HR method includes a TOF-MS scan with $m/z$ 300 to 1400 with 0.15 sec accumulation time followed by 12 targeted TOF-MS/MS scans for 5 peptides plus an IS peptide with 0.1 sec accumulation time and unit Q1 resolution for each scan with start/stop mass = 100/1400 Da.

Quantitative data were processed using Analytics module in Sciex OS 1.3 software for automatically integrating peaks for each precursor in TOF-MS scan and MS/MS scan. One of the MRM-HR transition data with highest intensity signal was used for quantitation of each peptide. A peak area ratio for each peptide against IS peptide was calculated for all 24 samples, and used for determining relative quantitation of both C-Aβ$_{6-29}$ and S-Aβ$_{6-29}$ peptides and their cleavage cleaved peptides for each timepoints against the time point of 24 h without rhADAM10 (Fig. 5e and Supplementary Fig. 3f, g).

**Aβ$_{42}$ aggregation assay.** The Beta Amyloid (1-42) Aggregation Kit (Cat # A-1170-2) was from rPeptide (Watkinsville, GA, USA). Rabbit pan-specific anti-succinyllysine (Cat # PTM-401) antibody and anti-β-Amyloid 1-16 Antibody 6E10 were purchased from PTM Biolab, Inc (Chicago, IL, USA) and BioLegend, Inc. (San Diego, CA, USA), respectively. To remove the preformed aggregates, the peptides were pretreated by the method[76] with minor modifications. Aβ$_{42}$ peptides were monomerized with a 2 h pre-treatment with 1, 1, 1, 3, 3, 3-Hexafluoro-2-propanol (HFIP) at 25 °C, and the solvent was evaporated. This process was repeated three times to remove any preformed aggregates. Samples were stored at −20 °C. Right before the experiments, 1 mg Aβ$_{42}$ was dissolved in 100 mM NaOH (1.1 mL), sonicated for 5 min and the solution was filtered through a 0.22 µm filter. The solution was diluted with Tris-buffered saline (TBS) (100 µM final concentration).

Aβ$_{42}$ (final concentration of 80 µM) was mixed with Thioflavin T (ThT, final concentration 24 µM) and succinyl-CoA (final concentration 3 mM) or ddH$_2$O in the TBS. Aliquots (100 µL) were immediately added to each well of a 96-well plate. The plate was maintained at a temperature of 37 °C for 24–48 h. Samples were taken during the time course of peptide aggregation to perform the experiments described below.

**Aβ$_{42}$ aggregates negative stain electron microscopy.** Eighty µM Aβ$_{42}$ peptide aggregated in vitro in the presence of succinyl-CoA (final concentration 3 mM) or ddH$_2$O for 24–48 h, which was prepared as mentioned in the Aβ$_{42}$ aggregation assay. Samples (5 µL) were placed on 400-mesh formvar-carbon coated copper grid and allowed to settle to 2 min. Excess fluid was removed, and the grids were negatively stained with one drop of 1.5% uranyl acetate solution and left for 2 min. Then the grids were picked up and excess negative stain was wicked off, and the grids were allowed to air dry. The samples were viewed using a JEM-1400 TEM (JEOL, Ltd, Peabody, MA, USA), operated at 100 kV and imaged on a Veleta 2 K × 2 K CCD camera (EMSIS GmbH, Munster, Germany).

**Western blot analysis of SDS-PAGE**. Aβ$_{42}$ aggregate samples were diluted with Tricine SDS Sample Buffer (Cat # LC1676, Thermo Fisher Scientific, Waltham, MA, USA) and separated on a 10–20% Tris-Tricine gel using Tricine SDS Running Buffer (Cat # LC1675, Thermo Fisher Scientific, Waltham, MA, USA). The separated bands were transferred onto a nitrocellulose membrane and detected with the mouse monoclonal anti-Aβ oligomer specific antibody NU-2 (1:4000; Klein's lab) and the mouse monoclonal anti-β-Amyloid antibody 6E10 (1:1000; Cat # 803001, BioLegend, San Diego, CA, USA). The membrane was probed with s680RD Goat anti-Rabbit IgG Secondary Antibody (1:10,000; Cat # 926-68071, LI-COR Biosciences, Lincoln, NE, USA) and 800CW Goat anti-Mouse IgG Secondary Antibody (1:10,000; Cat # 926-32210, LI-COR Biosciences, Lincoln, NE, USA). The protein bands were quantified by the Image Studio Lite software (version 5.2, LI-COR Biosciences, Lincoln, NE, USA). Molecular weights were estimated using a pre-stained protein ladder from Bio-Rad (Hercules, CA, U.S.A.).

**Tau peptide self-aggregation assay**. Synthetic peptides (Supplementary Fig. 4c) were purchased from GL Biochem (Shanghai, China). In order to avoid pre-aggregation, all of these peptides were pretreated with HFIP for the aggregation assays as detailed[77,78]. The synthetic lyophilized peptides were monomerized in HFIP for 10 min. HFIP was removed by evaporation, and the peptides were dissolved in ddH$_2$O and sonicated for 10 min. Aggregation was induced by incubating a final concentration of 10 μM peptides and 100 μM Thioflavin S at 25 °C in 20 mM MOPS, 0.15 M NaCl, pH = 7.2. Heparin (2.5 μM) was added immediately prior to the readings in order to initiate the aggregation[43,44]. The data were collected in triplicate at 20 s intervals using a kinetic assay mode with a Gemini EM microplate fluorescence reader (Molecular Devices, USA). The excitation and emission wavelengths were 440 and 490 nm, respectively.

**Tau peptide negative stain electron microscopy**. Fifty μM synthetic peptides reassembled in vitro in the presence of the polynomic cofactor heparin (12.5 μM) for 24 h, which was prepared as mentioned in the self-aggregation assay. Samples (5 μL) were placed for 1 min on 400-mesh copper grids covered with carbon-stabilized Formvar film. Excess fluid was removed, and the grids were negatively stained with 4 successive drops of 1.5% uranyl acetate solution, blotting excess stain between drops. After the final drop and blotting, the grids were allowed to air dry. The samples were viewed using a JEM-1400 TEM (JEOL, Ltd, Peabody, MA), operated at 100 kV and imaged on a Veleta 2 K × 2 K CCD camera (EMSIS GmbH, Munster, Germany).

ImageJ (version 1.52a) was used for the quantification of the width and height of the fiber helix. All photographed examples were measured in three cases, and the results averaged. All statistical analysis and visualization were implemented in Graphpad Prism 8 (GraphPad Software, San Diego, CA, USA).

**Expression and purification of recombinant tau K19**. Recombinant K19 protein was expressed in E. coli BL21/DE3 cells (Novagen, San Diego, CA, USA) transfected with plasmids for the tau fragment K19 under the control of a T7 promoter, as previously described[79]. Briefly, to produce $^{15}$N-labeled protein, cells were grown in a minimal medium containing $^{15}$N-labeled ammonium sulfate as the sole source of nitrogen. Over-expression was induced with 0.5 mM IPTG at mid-log growth phase at 37 °C. 3 h after induction, cells were collected by low speed centrifugation and lysed by sonication in a solution containing 3 mM Urea, 1 mM EDTA, 1 mM DTT, 10 mM Tris, and 1 mM PMSF followed by ultracentrifugation at 193,000 × g for 1 h in a Beckman ultracentrifuge using a Ti 50.2 rotor. The supernatant was dialyzed against 25 mM Tris, 20 mM NaCl, 1 mM EDTA, and 1 mM DTT before being purified by cation exchange chromatography, eluting with an NaCl gradient. Fractions containing tau K19 were pooled and dialyzed against 5% acetic acid before further purification by reverse-phase high-performance liquid chromatography on a C$_4$ column eluted with an acetonitrile gradient with 1% trifluoroacetic acid. Purified protein was dialyzed against dH$_2$O before being lyophilized and stored at −20 °C. Purity was confirmed by SDS-page.

**In vitro succinylation of purified K19 and $^{15}$N-labeled K19**. Purified K19 (a final concentration 28 μM) was mixed with 80 mM PIPES, pH 6.8, 2 mM MgCl$_2$, 1 mM GTP, and 0.5 mM EGTA. The reaction was initiated by the addition of Succinyl-CoA (final concentration of 1.5 mM) or the equivalent amount of buffer for control. Samples were incubated for 30 min at 27 °C after addition of Succinyl-CoA. The samples were concentrated by ultra-centrifugal filters for Succinyl-CoA removal, then diluted to 75 μL, and were ready for immediate use. 10 μL samples were stored at −80 °C, and then digested by trypsin. After sample cleanup by SPE, the samples were analyzed by nanoLC-MS/MS analysis and database search using PD 2.2 as described above.

Purified $^{15}$N-labeled K19 (a final concentration 99 μM) was mixed with Succinyl-CoA (a final concentration 2.88 mM) or the equivalent amount of buffer for control in the same reaction buffer for 30 min incubation at 27 °C. The samples were concentrated by ultra-centrifugal filters for Succinyl-CoA removal and stored on ice until use. Ten microliters samples were stored at −80 °C, and then digested by trypsin followed by LC-MS/MS analysis and database search using PD 2.2 as described above.

**RB3-SLD expression and purification**. Stathmin-like RB3 domain (RB3$_{SLD}$) containing two point mutations (C14A, F20W, stathmin numbering) was expressed in E. coli BL21/DE3 cells transfected with a PET-3d plasmid (a kind gift from Benoit Gigant) as previously described[52]. Briefly, over-expression was induced with 0.5 mM IPTG at mid-log growth phase at 37 °C. 3 h after induction, cells were collected by low speed centrifugation and lysed by sonication in a solution containing 20 mM Tris-HCl, 1 mM EGTA, 1 mM DTT at pH = 8.0, followed by ultracentrifugation for 15 min at 20,000 × g in a Beckman ultracentrifuge using a Ti 50.2 rotor. The supernatant was subjected to thermal denaturation (80 °C for 15 min then 10 min on ice) and was centrifuged again, followed by nucleic acid precipitation using 20 mM spermine-HCL at pH = 7.0 for 30 min at 4 °C with gentle agitation, and additional centrifugation (1 h at 100,000 × g). The supernatant was dialyzed against 20 mM Tris-HCl, 1 mM EGTA at pH 8.0 before further purification by anion exchange, eluting with an NaCl gradient. Purity was confirmed by SDS-PAGE before concentrating the purified protein to ca. 420 μM and buffer exchanging into 50 mM Phosphate, 0.1 M NaCl at pH = 7.0 using a gel filtration column. Final protein solution was flash-frozen in liquid nitrogen and stored at −80 °C until use. SDS-PAGE analysis was used to estimate the concentration of final protein stocks.

**T2R preparation**. Stathmin-like RB3 domain (RB3) binds to heterodimeric tubulin with a 1:2 stoichiometry, forming a longitudinal dimer of tubulin dimers. For $^1$H,$^{15}$N HSQC NMR experiments, $^{15}$N K19, succinylated or unmodified, was dissolved to a final concentration of 48 μM in Tris-d$_{11}$ buffer (25 mM Tris-d$_{11}$, 25 mM NaCl, 2.5 mM EDTA, and 1.5 mM DTT with 10% D$_2$O at pH 6.7) followed by addition of 53 μM of RB3-SLD stock solution. This mixture was then used to take up 4 mg of lyophilized tubulin (Cytoskeleton Inc., CO, USA) for a final T2R concentration of 50 μM.

**Microtubule assembly assay**. Tubulin (32 μM; Cat # T240-B, Cytoskeleton, Inc., Denver, CO, USA) in 80 mM PIPES, pH = 6.8, 2 mM MgCl$_2$, 1 mM GTP, and 0.5 mM EGTA was incubated for 2 min at 37 °C. The polymerization was started by adding 20 μL concentrated succinylated or normal K19 (a final concentration of 60 μM). The data were collected at 350 nm (20 s intervals) using a kinetic assay mode, using a SpectraMax 250 plate reader (Molecular Devices, San Jose, CA, USA).

**NMR spectroscopy**. Samples for NMR $^1$H,$^{15}$N HSQC experiments of free K19 were prepared by resuspending lyophilized $^{15}$N-labeled K19 in Tris-d$_{11}$ buffer (25 mM Tris-d$_{11}$, 25 mM NaCl, 2.5 mM EDTA, and 1.5 mM DTT with 10% D$_2$O at pH = 6.7). Samples for $^1$H saturation transfer difference (STD) NMR were prepared by dissolving synthetic tau peptides (GenScript USA Inc, NJ, USA) in GPEM buffer (80 mM PIPES, 2 mM MgCl$_2$, and 0.5 mM EGTA, at pH = 6.9) to a final concentration of 1 mM. Peptide solutions were then used to take up lyophilized tubulin to a final concentration of 20 μM.

All NMR spectra were collected on a Bruker AVANCE 600-MHz spectrometer equipped with a cryogenic triple resonance probe analyzed using Topspin4.1.1 and NMRView 9.2.0. $^1$H,$^{15}$N HSQC spectra were collected at 20 °C with 1024 complex points in the $^1$H dimension, 256 complex point in the $^{15}$N dimension, spectral widths of 13 and 26 ppm in the proton and nitrogen dimensions, with the carrier positions on water in the $^1$H dimension. Assignments were based on previously published assignments of K19[79,80]. Intensity ratios (I/I$_0$) were calculated in which I$_0$ represents the resonance intensity of residues when the protein is in a free-state, and I represents the resonance intensity of residues when the protein is in the presence of T2R (bound-state). $^1$H STD data were collected at 10 °C, with 4096 scans, a saturation time of 3 seconds, and on/off-resonance frequencies set to −0.5 ppm and 60 ppm, respectively.

**Bioinformatics analysis and statistical analysis**. These two cohorts were combined by protein ID. Filtering base on the unique GI number or UniProtKB accession number, identical protein data in these two batch results were merged together and the final result set included only the proteins or succinylated peptides that were present in both batches.

**Subcellular localization analysis**. Subcellular localization of the identified candidates was determined using Cytoscape (version 3.6.1)[81] and stringAPP (version 1.4.0)[82] software. All the parameters were set to the default values, but only these highest compartment scores equal 5 as the high confidence localization were kept. The result was visualized in FunRich (version 3.1.3).

**Succinylated peptide sequence motif discovery and iceLogo heatmap**. To determine the sequence motif for succinylation, succinylated peptides were extracted cytowithin seven amino acids upstream and downstream of identified succinylation sites. The web-based Motif-X program (version 1.2 10.05.06) (http://motif-x.med.harvard.edu/)[83,84] was used to identify statistically significant motifs from the large post-translational modification peptide sequences. The motif width was chosen to be a length of 15, the occurrences number was set at 5, and the

significance threshold was set at 0.00001. Since all proteins were derived from human brain, the "IPI Human Proteome" was used as the background database.

Heatmap of 15 amino-acid compositions is used to create an overview of all possibilities in a 2D space compared to the central succinylated site, on the IceLogo tool[85]. The precompiled Swiss-Prot composition was chosen to be Homo sapiens and the start position was set at −7. Only significantly up and downregulated elements, according to the given $p$-value ($p$-value = 0.05), are colored in, respectively, a shade of green and red. The nonregulated elements are colored black.

**Gene ontology (GO) and KEGG pathway enrichment analysis.** Analysis and visualization of Gene Ontology terms associated to succinylated proteins was performed with ClueGO (version 2.5.1)[86]. The following parameters were used when running ClueGO: Min GO Level = 3; Max GO Level = 8; Minimum Number of Genes associated to GO term = 3; Minimum Percentage of Genes associated to GO term = 4. Enrichment $p$-values were based on a two-sided hypergeometric test and Bonferroni step-down method corrected for multiple testing correction. $P$-value cutoff of 0.01 and a minimum of five genes per ontology were used as filters prior to pruning the ontologies.

**Peptide and protein quantitation.** Perseus software[87] (version 1.6.0.7) was used for statistical analysis of the peptide and protein abundance data. In brief, quantitation was performed on these defined as quantifiable peptide and protein set including only those identified in both two batches and in a minimum of eight replicate in each group. The abundance ratio of AD and control peptides or proteins was defined as fold change (FC), and we used the logarithmic transformation of fold change (log₂FC) to represent the AD/control difference. Then the significant differences in succinylated peptide and protein levels were computed using two-tailed Student's $t$-test and significant succinylated peptide level changes were defined as $p$-value < 0.05. Significant protein level changes were defined as $p$-value < 0.05 and |log₂FC| > 0.25. The search result including ratio, $p$-value, succinylated peptide abundance for each sample was output to Microsoft Excel software (MicroSoft Office 365) for further data analysis. All the succinylated peptides and proteins with log₂FC and corresponding $p$-value are available in Supplementary Data 4 and 5.

**Hierarchical clustering.** Global proteomic proteins data (rows) were clustered using uncentered pearson correlation and samples (columns) were clustered using city block distance, with an average linkage clustering method by Cluster 3.0[88]. Clustering results were visualized using Java TreeView3.0 beta01 (https://bitbucket.org/TreeView3Dev/treeview3/).

**Cell culture.** HEK293T cells (Cat # CtRL-3216) was purchased from the American Type Culture Collection (ATCC, Manassas, VA, USA) and was cultured following ATCC protocol. Unless otherwise noted, all cell culture supplies and medium were from Thermo Fisher Scientific (Waltham, MA, USA).

**Immunocytochemistry, image acquisition, and image analysis.** HEK293T cells were cultured on the Poly-D-Lysine-coated Delta T dishes (25 μg/mL) at a seeding density of $5 \times 10^4$ cells /dish. The next day, cells were treated with 100 nM Rotenone in complete medium (DMEM + 10% FBS) for 60 min at 37 °C. After 60 min, the cells were fixed in 4% formaldehyde in PBS (Image-iT Fixative Solution, Thermo Fisher Scientific, Waltham, MA, USA) at R.T. for 15 min. After washing in PBS and pre-incubation (1% Triton-X100 in PBS for 10 min at R.T. and PBS + 2% BSA for 60 min at r.t.), they were labeled with a mixture of rabbit anti-DLST (1:600; Cat # 11954, Cell Signaling Technology, Inc. Danvers, MA, USA) and mouse anti-COX-IV (1:1,000; Cat # 11967, Cell Signaling Technology, Inc., Danvers, MA, USA) antibodies. The primary antibodies were diluted in PBS + 0.5% BSA and incubated overnight followed by a mixture of the secondary antibodies (Donkey anti-Mouse IgG (H + L) Highly Cross-Adsorbed Secondary Antibody, Alexa Fluor 488 (1:1,000; Cat # A21202, Thermo Fisher Scientific, Waltham, MA, USA); Donkey anti-Rabbit IgG (H + L) Highly Cross-Adsorbed Secondary Antibody, Alexa Fluor 568 (1:1,000; Cat # A10042, Thermo Fisher Scientific, Waltham, MA, USA)) for 1 h at r.t.. Confocal stacks were taken using a ×100 oil objective in 0.4 μm z-steps under inverted Nikon C1 confocal microscope. Image analysis was performed in Fiji software (Fiji, RRID:SCR_002285). From a confocal stack, all images (6–13 images) were taken for further analysis. Mitochondria were outlined using an automated thresholding of COX-IV-positive particles. Cytoplasm was defined as an area surrounding mitochondria. The created masks of mitochondria and cytoplasm were separately applied to the correspondent DLST images and integrated density of DLST was determined in mitochondria and cytoplasm, respectively. Integrated density of DLST was normalized to the area of mitochondria.

MEAN integrated intensity was calculated from all particles in mitochondrial or cytosolic region in each field. Error bars represent SEM deviation from the mean ($n = 98$ fields from 19 dishes from two experiments; mitochondrial (52 control; 46 treated); cytosol (52 control; 46 control); Tukey's multiple comparisons test).

**Preparation of the cytosolic and mitochondrial fractions.** HEK293T cells were cultured on the 10 cm dishes. After two days, cells (80% confluent) were treated with or without rotenone (100 nM or 5 μM) in balanced salt solution (140 mM NaCl, 5 mM KCl, 1.5 mM MgCl₂, 5 mM glucose, 10 mM HEPES, and 2.5 m M CaCl₂, pH = 7.4) for 20 min at 37 °C. Cells were then washed twice with cold phosphate-buffered saline (Cat # 14190250; Thermo Fisher Scientific, Waltham, MA, USA). The cytosolic and mitochondrial fractions were prepared as described previously[34].

**Immunoprecipitation and western blots.** Samples were prepared as described[34] with the following antibodies: Pan anti-succinyl lysine (1:200 for immunoprecipitation; 1:1000 for western blot; Cat # PTM-401, PTM Biolabs Inc., Chicago, IL, USA); Anti-Pyruvate dehydrogenase E2/E3bp antibody (1:1000; Cat # ab110333, Abcam, Cambridge, MA, USA); Pyruvate Dehydrogenase Antibody (1:500; Cat # 2784, Cell Signaling Technology, Inc., Danvers, MA, USA); Anti-beta Actin antibody (1:4000; Cat # ab8226, Abcam, Cambridge, MA, USA); COX-IV (3E11) Rabbit monoclonal antibody (1:3000; Cat # 4850, Cell Signaling Technology, Inc., Danvers, MA, USA); Anti-KGDHC E1k (1:1,000; Rockland, Limerick, PA, USA); Anti-KGDHC E2k (1:1,000; Rockland, Limerick, PA, USA); Anti-KGDHC E3 (Lipoamide Dehydrogenase Antibody) (1:2000; Cat # 200-4160 S, Rockland, Limerick, PA, USA); β-Actin (13E5) Rabbit antibody (1:1,000; Cat # 4970, Cell Signaling Technology, Inc., Danvers, MA, USA); s680RD Goat anti-Rabbit IgG Secondary Antibody (1:10,000; Cat # 926-68071, LI-COR Biosciences, Lincoln, NE, USA) and 800CW Goat anti-Mouse IgG Secondary Antibody (1:10,000; Cat # 926-32210, LI-COR Biosciences, Lincoln, NE, USA).

All the quantification of immunoblotting analyses was taken in Image Studio Lite (version 5.2, LI-COR Biosciences, Lincoln, NE, USA), and all statistical analysis and visualizations were implemented in Graphpad Prism 8 (GraphPad Software, San Diego, CA, USA).

**Animals.** All the experiments were carried out in four and 10-month-old transgenic mouse models of AD. Tg19959 mice (that overexpress a double mutant form of the human amyloid precursor protein) were obtained from Dr. George Carlson (McLaughlin Research Institute, Great Falls, MT, USA). Tg19959 mice were constructed by injecting FVB X 129S6 F1 embryos with a cosmid insert containing human APP₆₉₅ with 2 familial AD mutations (KM670/671NL and V717F), under the control of the hamster PrP promoter[89]. P301S (PS19, that overexpress the human tau gene harboring the P301S mutation) transgenic mice were purchased from The Jackson Laboratory (Bar Harbor, ME, USA). The transgenic mice used in this study express the human pathogenic mutation P301S of tau together with the longest human brain tau isoform (htau40) under control of the neuron-specific mThy1.2 promoter[90]. Animals were maintained under standard conditions of 12 h light/dark cycle, 22 ± 1 °C r.t., and 50–70% humidity. Subjects were given ad libitum access to food and water. All experiments were approved by the Animal Care and Use Committee of Weill Cornell Medicine.

**Immunofluorescence.** All brain tissue sections were stained at the same time under the same conditions (solutions, washing, temperature, etc.) and analyzed under identical experimental settings. Our results were expressed as the mean with SEM representative of the average of ~900–1000 pyramidal neurons comprised in 3–4 different brain sections per animal ($n = 4$ mice per each group). For fluorescence analysis, tissue was washed six times in PBS for 10 min each at r.t. After blocking in 10% normal donkey serum in PBS for 1 h, sections were incubated using primary antibodies (anti-MAP2 Antibody (Cat # AB15452, Millipore, Burlington, MA, USA); pan anti-succinyl lysine (Cat # PTM-401, PTM Biolabs Inc., Chicago, IL, USA); anti-mouse NU-4 (Klein's lab); anti-mouse β-amyloid (Cat #15126 S, Cell Signaling Technology, Inc., Danvers, MA, USA); anti-mouse T22 (Kayed's lab); anti-phospho-tau AT8 (Cat # MN1020, Thermo Fisher Scientific, Waltham, MA, USA)) in PBS with 1% normal donkey serum overnight at 4 °C. Next, hippocampal sections were rinsed three times in PBS for 10 min each and subsequently incubated with conjugated secondary antibodies (donkey anti-mouse IgG (H + L) Highly Cross-Adsorbed Secondary Antibody, Alexa Fluor 488 (1:500; Cat # A21202, Thermo Fisher Scientific, Waltham, MA, USA); Cy3 AffiniPure donkey anti-Rabbit IgG (H + L) (1:500; Cat # 711-165-152, Jackson ImmunoResearch, West Grove, PA, USA); 647 AffiniPure donkey anti-Chicken IgY (IgG) (H + L) (1:500; Cat # 703-005-155, Jackson ImmunoResearch, West Grove, PA, USA)) for 2 h. After three washes in PBS for 10 min, the samples were mounted directly onto plus-coated slides and coverslipped using gelvatol mounting media.

**Image analysis.** The immunoreactivity of succinyl lysine was assessed in three confocal images for the CA1 region and two images for the CA2 region. For quantitative analysis, confocal fluorescence micrographs were obtained with the ×40 objective lens. Representative images were captured using ×60 magnification at high resolution (100 μs). Values are mean ± SEM representative of the average of ~900–1000 MAP2 neurons or 60 Aβ plaques comprised in 3–4 different hippocampal sections per animal. The fluorescence intensity of succinyl lysine was normalized to the number of pyramidal neurons. Four mice per group. Data were expressed as the mean with SEM representative of the average of ~900–1000 MAP2 neurons or 60 Aβ plaque comprised in 3–4 different hippocampal CA1 sections per

animal ($n = 4$ per each group). The fluorescence intensity of succinyl lysine was normalized to the number of pyramidal neurons or the number of plaques. Two-tailed Student's $t$-test was performed.

**Statistics and reproducibility**. Data were presented as mean ± SEM unless indicated otherwise. Two-tailed Student's $t$-test was applied when comparing data from two groups. One-way ANOVA followed by Tukey's multiple comparisons test were performed for data with more than two groups. For studies with repeated measures, two-way ANOVA followed by Tukey's or Bonferroni's multiple comparisons test was performed. Statistical analyses were performed using GraphPad Prism 8 statistical software or Microsoft Excel software (MicroSoft Office 365). Sample sizes and statistical details are indicated in figure legends.

All cell culture experiments presented in the manuscript were repeated at least three times independently with similar results.

**Reporting summary**. Further information on research design is available in the Nature Research Reporting Summary linked to this article.

## Data availability

All data needed to evaluate the conclusions of this study are available in the manuscript, in the supplementary information files or from the data bases listed in the manuscript. The mass spectrometry proteomics data files including raw MS files, peak list files and search results files were deposited to PRIDE database by ProteomeXchange (PXD015124). Source data are provided with this paper.

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

## Acknowledgements

The studies were supported by: NIH-NIA grants P01AG014930 (G.E.G., M.F.B., and S.Z.) and R37AG019391 (D.E.); R01-EY026576 and R01-EY029796 (B.T.S.); NIH SIG 1S10 OD017992-01 (S.Z.); HHSN271201300031C (V.H.); AG18877 & AG22547 (W.L.K.); AG059319 and AG058469 (S.G.); AG005138 and AG066514 (to Mary Sano); R37AG019391 and RF1AG066493 (D.E.) and F31AG069416 (D.A.); S10OD016320 (William Clay Bracken, director of WCM NMR facility) Burke Neurological Institute, Weill Cornell Medicine; Zhejiang Provincial Natural Science Foundation of China LGF21H090018 (Y.Y.); Integrated Medicine Research Center for Neurological Rehabilitation, College of Medicine, Jiaxing University, Jiaxing, China (Dean J. Chen). We thank L. Cohen-Gould, MS, director of the Microscopy and Image Analysis Core Facility (Weill Cornell Medicine) for the EM and C. Bracken, PhD, director of the NMR Facility (Weill Cornell Medicine) for help with NMR experiments. We thank E. Ivanova and Structural and Functional Imaging Core at the Burke Neurological Institute for the technical assistance. We are grateful to the NIH Neurobiobank for providing the carefully characterized human brains. We thank Dr. R. Kayed (Department Neurology, University of Texas Medical Branch) for kindly providing the T22 antibody for tau aggregates. The mass spectrometry proteomics data have been deposited to the ProteomeXchange Consortium via the PRIDE partner repository with the dataset identifier PXD015124. For public access by webpage: http://www.ebi.ac.uk/pride/archive/projects/PXD015124 and for FTP download: ftp://ftp.pride.ebi.ac.uk/pride/data/archive/2021/12/PXD015124.

## Author contributions

Y.Y. and G.G. conceived the research program and designed the experiments. V.H. contributed to the patient consent, collection of samples. V.H., X.H., and E.T.A. processed the brain samples. R.B. and S.Z. performed nanoLC-MS/MS analysis. Y.Y. performed data analyses. Y.Y. and X.H. performed biochemical experiments. Y.Y., X.H., and H.C. performed the cell experiments. X.H., H.C., E.I., and B.T.S. performed immunofluorescence on the rotenone treated cells and analyzed the data. D.A. and D.E. designed and performed NMR analysis, processed and interpreted the data and prepared figures. V.T participated in the design and conceptualization of the animal study, analyzed the data, and prepared the figures. M.F.B. participated in the design and conceptualization of the animal study. H.L. participated in the experimental design and write-up. S.G., W.K., and K.V. provided antibodies and contributed to the design. Y.Y. and G.G. wrote and edited the manuscript. J.C. contributed to the additional experiments in revision of the manuscript. All authors discussed the results, and Y.Y., G.G., S.Z., D.E., S.G., V.H., V.T., B.T.S, H.L., and E.I. contributed to the writing. All authors read and approved the manuscript.

## Competing interests

The authors declare no competing interests.

## Additional information

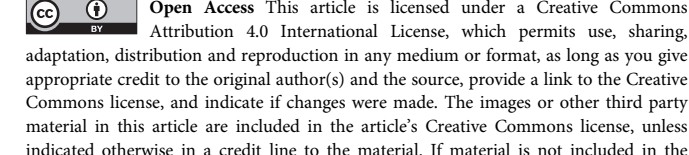

