## [Peer Review File · Nature Communications]

Altered succinylation links abnormal metabolism to APP and tau in Alzheimer's DiseaseREVIEWER COMMENTS

Reviewer #1 (Remarks to the Author):

Yang et al. describes a link between succinylation and amyloid and tau pathology in Alzheimer's disease (AD). The authors start by identifying differentially-changed succinyl-containing peptides by isobaric tandem mass tagging in two separate cohorts containing control and AD brain tissues (n= 5 cases each). They briefly describe characteristics of identified, succinylated proteins by in silico analyses. By stressing HEK293 cells with rotenone, the authors observe the translocation of KGDHC proteins to the cytosol and the loss of succinylation in mitochondrial proteins, highlighting a possible mechanism of metabolic change in AD. Authors also identify A β and Tau succinylation exclusively in AD brain. APP and Tau transgenic mouse models were also used to assess lysine succinylation on amyloid and tau pathology. The authors show increased rates of fibrillization in purified, succinylated A β and Tau peptides. Importantly for Tau, they show that the K311Succ site identified in AD brains shows a compromised ability to bind tubulin, similar to the effect of hyperphosphorylation and acetylation previously observed. Overall this is an interesting report that highlights the role of a new PTM in modifying Tau and amyloid structure. However, despite the many intriguing results, there remain many limitations of the data presented. This would include an improved vetting of the proteomic data, starting with sharing of peptide data from Proteome Discoverer to aid in the review of this manuscript. As succinylation in the realm of AD has been understudied, the levels of succinylation, especially on tau, should be compared to acetylation and ubiquitination. There were no comparisons of the spectra of A β or MAPT peptides identified in brain to synthetic standards. Also, the imaging has many labeling errors and does not strictly agree with the interpretations of the authors. An independent validation of the A β and Tau succinylation by some other means (WB, immunogold microscopy, etc.) is also warranted.

Major Comments:

- In what manner was the subcellular localization determined for Fig.2a/Supplementary Table 2? Gene ontology?
- Was the sample arrangement per each TMT batch ordered 5 CTL in a row then 5 AD in a row?
- Given the low number of succinylated peptides corroborated by both cohorts, please share a supplementary table that include the peptide data following database search (including Xcorr, PEP, etc.)
 - o What are the succinylation localization scores for each succinylated peptide?
 - o What was the succinylation localization score threshold that you set your Proteome Discoverer to?
 - o What methods were used to filter out falsely-localized succinylated peptides?
- ♣ Please provide the succinylation mass shift (+100.0160) used in the database searches as well as masses for the other PTMs.
 - o What was the average number of MS2 ions identified in succinylated peptides compared with non-succinylated peptides? The % enrichment
 - o What were the number of succinylated peptides identified in the total proteome dataset without enrichment?
 - o What was the average number of missed cleavages for each succinylated peptide?
- Please provide spectra for the A β HDSGYEVHHQKLVFFAEDVGSNK succinylated peptide from brain and compare this to the synthetic standard (preferably heavy labeled). The authors do a nice job characterizing the in vitro succinylation of amyloid and Tau by MRM. Can the authors show side by side the MS/MS or MRM profiles of these standards compared to the peptides identified in the discovery proteomics screen?
- It is surprising that the VQIVYK Tau hexapeptide was identified with the C-terminal lysine (residue 311) succinylated (Fig. 6b). It has been previously reported that trypsin/LysC cleavage is blocked by succinylation (https://link.springer.com/protocol/10.1007/978-1-60327-259-9_53). The authors need to confirm the MS/MS spectrum in AD with standards or use a targeted MS approach to map the more likely modified VQIVYK(succ)PVDSLK peptide in AD brain that they identified in their in vitro assays.
- In most published proteomic studies mitochondria proteins are decreased in AD within the total proteome. How do the authors reconcile this with the succinylome? Are differences in protein abundance rather than site specific changes due to the PTMs driving the decrease in AD. Can the

authors provide or estimate any changes in stoichiometry measurements?

- It is not clear why the pan-lysine succinyl antibody labeling so drastically attenuated in WT 10 month old mice? And why there such an increase in succinylation in Tg19959 mice? Authors need to resolve.
- Are the genotype and time point labels switched in Figure 5d? The images do not agree with the quantitation Also, why is there so little co-localization with A β plaque and the succinyl-lysine antibodies? Authors need to explain.
- In Figure 6c the authors label the images with a mouse model (Tg19959) that does not match the mouse model described in the text (TgP301S).
- Again, there is little co-localization with aggregated Tau antibody IF labeling and pan-lysine succinylation. One would expect extensive co-localization if succinylation labels the core of PHFs, where this PTM maps.
- The authors state on Page 13, Lines 309-310 to Page 14, Lines 311-312: “a weak signal for succinylated tau occurred in 10-month-old TgP301S” ... “indicating a desuccinylation process may exist in the final states of tau deposition”. Also, with no evidence, the authors offer a phosphorylation-succinylation switch as a possible reason.
- Authors state that Both APP and Tau were highly succinylated at critical sites in nine out of ten AD brain samples, but no succinylation of APP or tau was detectable in any control brains. Just because these sites were detected only in AD does not necessarily mean they are highly succinylated. What about lysine acetylation or ubiquitination? Are these sites on tau modified by these PTMs? Did the authors search their data for these PTMs at the same sites? Lysines are highly modified residues and one would expect succinylation to be a relatively minor pool of modified Tau in the AD brain. The authors should measure the pools of these PTMs (succinylation, ubiquitination, acetylation) to assess the relative frequency of succinylation vis a vis acetylation and ubiquitination.
- The weak signals on Tau contradict the human data of increased succinylated Tau. How do the authors reconcile these findings? Could other PTMs on these sites (ubiquitin, acetylation) also occur or do mice have increased SIRT5 (or other SIRT/desuccinylase enzymes) abundances? The global human proteomes analyzed showed no changes in SIRT5 as the authors note in the manuscript in Extended Data Fig. 2c. Could this differ in the mice?
- The authors should visit larger proteomic datasets for more accurate steady-state protein level quantification in AD cohorts that can fully describe ADAM/SIRT family levels in AD as compared with controls.
- Can the authors isolate AD brain PHF tau aggregates then perform immunogold labeling with the pan lysine succinylation antibody? Also for Abeta plaques? Then with a non-AD tauopathy control (ex: Corticobasal degeneration with prominent K311 Acetylation?) Describing that your in vitro tau fibrils look like AD brain fibrils doesn't stand up. This would be an independent validation of your mass spectrometry experiments
- The authors should include PHF6-K311Acetyl as a positive control to compare the succinylated peptide for the Tau Self-aggregation assay.

Minor Comments:

- Fig. 2A: change “succinlyated” to “succinylated”
- Fig. 2B: change “succinlated” to “succinylated”
- Change APP770 to APP695 (as well as accompanying notations), as APP695 is the major neuronal species (Fig. 5)
- Specify the mass shift specific to lysine succinylation in the methods
- There are some very minor fold changes between the control and AD brain sample proteins in Extended Data Figure 2. Is this due to MS/MS spectral interference and compression. Can the authors discuss.
- Page 11, Line 252-253: The Tg19959 line contains three mutations:
 - o Swedish: K670N/M671L
 - o Indiana: V717F
- Page 9, Lines 205-206. What was the correlation between change in succinylation and change in protein abundance? Please illustrate/state
- Has acetylation been observed at A β K16?
- The authors should provide High-resolution images for Extended Data Figure 3a-b?

- Please provide a representative image for cleaved A β 6-29 fragment precursor ion peaks in Extended Data Figure 3. The plot in Extended Figure 3f-g is confusing
- What is “Percentage Change (%)” quantified in Figure 5f?
- Page 13, Line 300: effects of what?
- Page 13, Line 302: “IF staining to compare the presence or absence of succinylation with that of tau” – What does this mean
- Page 13, Line 304: “but in 4-month-old TgP301S mice”.
 - o What occurred in 4-month-old TgP301S mice?
- Figure 6d: “d” is covered up by panel c
- Please label Fig.6f-h with the peptide at hand (f=PHF6, g=S-PHF6, h=9:1 mix)
- Page 17, Line 394: “bot amyloidosis”?
- Page 8 Line 179: “Since no specific motifs for lysine succinylation in human cells have been reported” – They have been reported in HeLa cells in Weinert et al., Cell Reports (2013) <http://dx.doi.org/10.1016/j.celrep.2013.07.024>. Since they have reported previously, the authors should compare Motifs in AD brain to theirs, which upon first glance, don’t exactly match.
- Page 16, Line 375: fix spelling errors

Reviewer #2 (Remarks to the Author):

The manuscript “Succinylation Links Metabolic Reductions to Amyloid and Tau Pathology” compares succinylation in AD versus controls and surprisingly identifies AB and tau as targets that succinylated exclusively in AD. As a potential rationale for why these proteins may be succinylated in AD, the paper shows that mitochondrial dysfunction in cells leads to escape of proteins from the mitochondria that may be functioning in the succinylation of pathological AB and tau. In addition, the authors perform a series of biochemical experiments which suggest a way that succinylation may facilitate pathological AB and tau.

In the end, I think they can say that this paper demonstrates that a new protein modification is found on APP and tau that may correlate with AD status. In addition, some nice biochemical experiments raise a potential way in which this modification can potentially influence the aggregation of these proteins. There is no evidence that it is actually functionally doing so in vivo, nor is there any evidence that it correlates with progression in humans, and the correlation in mice is the weakest part of the paper. It remains possible that the modification is simply a consequence of mitochondrial dysfunction in AD patients and the biochemistry not actually relevant to what is functionally occurring in vivo—perhaps because the succinylation moiety is rather large. Nevertheless, even in this case, succinylation could serve as a marker, so it is still potentially relevant. Also, there is sufficient data to warrant following up the work. Thus, overall the findings are interesting. However, the level of over-interpretation and over-blown claims are reckless and unwarranted, so the text needs major revisions. There are also some experimental concerns.

Major comments:

The introduction is really short for an unknown topic and the Nature Communication format. A lot more needs to be added for the reader to understand succinylation, mitochondrial dysfunction, and AD pathology.

In Figure 3C, demonstrating that the change in succinylation is not just due to changes in protein levels is a critical point. The correlation that is shown, though weak, is a bit troubling. There is a brand new paper (Johnson et al, Nature Medicine 2020) focusing on proteomics in neurodegenerative diseases. I think it would be important to compare the changes in succinylated proteins to the changes in proteins presented in an independent paper, such as this one to make this point more convincing.

Figure 4A, without a control for protein loading, the overall change in levels of succinylation are

meaningless. This is somewhat mitigated by the B-actin control for the individual proteins in B. However, is the B-actin from 4B also being used to normalize 4C? The B-actin should be shown on the same blot in C. Also, there probably should be controls showing that the fractions have been sorted intact. In Figure 4D, it looks like there is less colocalization in the Rotenone treatment. Are the images reversed? In either case, the resolution of the images is too low to comment on the localization. Also, why was only 100nm Rotenone shown? What about 5uM? Overall, I probably buy the interpretation, but the data could be cleaned up.

In Figure 5C and 5D, it would be nice to have an unaffected staining control to show that the change succinylation is specific. More importantly, in WT there is a dramatic decrease in succinylation between 4 months and 10 months. Why is this? Could this be due to mitochondrial changes in normal aging. This should probably be commented on in the discussion. In the Tg mouse, there a lot more succinylation. However, there is also a decrease in succinylation from 4 to 10 months that is similar to WT, despite the fact that AB is definitely increasing in the Tg mice over this time period. Thus the two do not seem to be particularly well correlated. Also, is the level of succinylation increase in the Tg mice prior to appearance of AB? This should be checked because they already observe a dramatic increase in succinylation at 4 months when AB is first forming. Thus, it is possible that succinylation is changing in the Tg model well before this. Overall, though it is clear that succinylation is responding to the Tg, it is not at all clear that it correlates with the build up of AB. The exact same thing is true for tau in Figure 6. The fact that the same phenomenon is true in both AB and Tau models is perhaps even more disconcerting, because it suggests that the phenomenon is not specific. That is to say, the overall increase in succinylation seems to occur regardless of the pathological insult, which does not cause mitochondrial dysfunction exactly the same way in the two models, and the timing of that pathological insult, which is not exactly the same in the two mouse models. This should at least be discussed. In particular, it would be nice to know how the changes in succinylation relate to the changes in mitochondrial dysfunction in the two Tg models in their hands. Overall, the data supports a change in succinylation that occurs in the two pathological models, but does not support the conclusion that the change in succinylation truly correlates with pathology. At a minimum, this should be clearly noted in the results and discussion.

There is a slight concern that the peptide used for the tubulin polymerization assay is succinylated throughout, when they only detected succinylation at K311 in AD. Nevertheless, the loss of Tau polymerization function is impressive. This should be more clearly stated and used to qualify the interpretation. This is particularly true because the succinylation moiety is quite large. The authors should definitely discuss how such a large modification could affect proteins and the biochemical assays that they perform on them. The decrease in Tau-tubulin interactions is more convincing, particularly because they also performed this assay with Tau only succinylated at K311.

Overall, it is possible that succinylation is simply a consequence of mitochondrial dysfunction, and not necessarily functional in AD. To mitigate this, I think it might be good to provide some additional negative controls if possible. ie do AB and tau accumulate any other post-translational modifications that might just be due to disruption of mitochondria or the abnormal appearance of the pathological versions of these proteins in the cytosol. Or is it specific to succinylation? Is there any way to truly rule out that the observations are simply due to mitochondrial dysfunction and not necessarily functional?

There is a lot of wild speculation:

Example: The last sentence of the abstract is wildly overexaggerated- While there is a possibility that succinylation could contribute to pathologies in AD, the data presented in the manuscript certainly are not by themselves enough to even raise the possibility that succinylation must be addressed therapeutically for meaningful clinical benefit

Example: The last sentence of the introduction is overstated and unnecessary

Example: line 311- this reflected a potential existence of succinylation-phosphorylation switch as in the case of acetylation- the paper provides absolutely no evidence for this. This could be speculated about in the discussion, but is completely inappropriate for the results section.

Example: line 283- taken together the accumulated data strongly suggest that succinylation of K678

might lead to an early-onset enhanced generation, oligomerization and plaque biogenesis, consistent with the effects of known genetic disease mutations at this site. While the data suggest a potential was for succinylation to affect AB cleavage, there is no functional evidence that it does so.

Example line 369: Notably, these results demonstrate for the first time that succinylation is the key link between the signature metabolic reductions and amyloid plaques and neurofibrillary tangles in AD.-

Again, although this is possible, there is absolutely no evidence for this in the manuscript

Example line 371: The current results reveal that varied in protein succinylation, as a molecular signal, correlates with altered cerebral metabolic function in AD as the disease progresses.- While there may be a small amount of evidence of this in the mice data (if you compare to previous analysis of the mice strains employed and you ignore the fact that the mouse data don't truly correlate), they do not show this on their own and they certainly did not examine succinylation across the progression of the disease in human cases, so this statement is not warranted.

Minor comments:

I really don't understand the math in figure 1, and there is no description of what the 29 proteins in B are? Are they differential between AD vs control, as in D? And what percentages are up vs down? An effort should be made to make the numbers more clear.

The antibody used in extended data figure 3B needs to be clearly shown in the figure.

In extended figure 3F and G, why is the full length protein so different when the production of the cleavage products remains the same? Perhaps I don't quite understand the assay, but this should be clarified.

The label of the graph in 5F needs to be clarified as percentage change from 0hrs.

Line 309 refers to succinylated tau. However, it should refer to succinylation in the Tau Tg mice. There is a big difference.

The Y-axis in Figure 4B says evel instead of level

The Tg19959 and P301S mouse models need to be defined for the reader.

Line 194 should say from rather than form

PDHA1 is mentioned with no context

Line 257: paralleled should be parallel

The logic in line 266-267 needs to be better spelled out for the reader with regard to the competition of the two enzymes. Likewise in line 271, ADAM10 needs to be introduced as a secretase for the reader.

The S and C labels in 5C need to be defined in the main figure.

Line 302: abeta should be absence?

Line 371: varied should be variation

Line 375: involvon should be involved in?

Line 394: bot should be both?

Line 385: The decline in succinylation of mitochondrial proteins suggests that activation of descuccinylases- The alternative, that there could be a failure to maintain succinylation levels, should

be mentioned.

The manuscript should also be edited for grammar.

David Katz
Emory University

Reviewer #3 (Remarks to the Author):

The authors have investigated the potential role of succinylation and Amyloid and Tau pathology using brain tissue from AD cases and controls. They analysed brain tissue cell lysate proteomes using 10 plex TMT. They also analyzed the same 10 controls and 10 AD patients' brain samples Succinylome using Cell Signaling Tech IP-MS kit and ran LCMS of the PTM enriched tryptic peptides. Comments:

- 1- In addition to bioinformatics analysis of succinylome IP-MS data it would be useful to analyze and show the biological significance of whole tissue lysate 10- plex TMT data as well. It would be useful to cover the global proteome analysis done which may be relevant to disease pathogenesis in addition to the succinylome targeted concept.
- 2- It is important to pinpoint sites of protein succinylation. Succinylome localization shown in Figure1a is not clear.
- 3- The authors considered impaired mitochondrial function resulted in succinylome localization to be pushed out of mitochondria to cytosol by leakage (Figure 4, Line 244-247), however, whole tissue lysate mass spec succinylome data suggested an overall decrease in AD. These findings need to be reconciled
- 4- The K687 site in the middle of APP is the interaction site of α alpha-secretase and the cleavage was inhibited when the K was succinylated in vitro (Figure 5e). However, it cannot be assumed that succinylated A β has more aggregation property since the comparison shown in Figure 5f did not show a significant difference. The AD patient succinylome mass spec identified succinylated K687 peptide (S Table 5). It is not clear whether the succinylome IP conditions favored solubilizing Abeta plaque and protofibril oligomers.
- 5- APOE4 mutation is a risk factor of AD. Does the proteomics data reveal mutation in the 20 patient brains analyzed?

RESPONSE TO THE REVIEWERS' COMMENTS

The reviewers' comments are in RED.

Our answers are inserted in blue immediately after each question. The sections from the manuscript showing the change with line numbers were inserted after our rationale.

The line numbers refer to the respective document (manuscript, methods etc of final documents). A copy showing all changes is included.

REVIEWER COMMENTS

Reviewer #1 (Remarks to the Author)

Yang et al. describes a link between succinylation and amyloid and tau pathology in Alzheimer's disease (AD). The authors start by identifying differentially changed succinyl-containing peptides by isobaric tandem mass tagging in two separate cohorts containing control and AD brain tissues (n= 5 cases each). They briefly describe characteristics of identified, succinylated proteins by in silico analyses. By stressing HEK293 cells with rotenone, the authors observe the translocation of KGDHC proteins to the cytosol and the loss of succinylation in mitochondrial proteins, highlighting a possible mechanism of metabolic change in AD. Authors also identify A β and Tau succinylation exclusively in AD brain. APP and Tau transgenic mouse models were also used to assess lysine succinylation on amyloid and tau pathology. The authors show increased rates of fibrillization in purified, succinylated A β and Tau peptides. Importantly for Tau, they show that the K311Succ site identified in AD brains shows a compromised ability to bind tubulin, similar to the effect of hyperphosphorylation and acetylation previously observed. Overall, this is an interesting report that highlights the role of a new PTM in modifying Tau and amyloid structure. However, despite the many intriguing results, there remain many limitations of the data presented.

This would include an improved vetting of the proteomic data, starting with sharing of peptide data from Proteome Discoverer to aid in the review of this manuscript.

As detailed in response to specific comments below (lines 157-167, 721-727) in this response), we improved the vetting of the proteomic data including making the data from Proteome Discoverer software available (see Supplementary Table 7).

As succinylation in the realm of AD has been understudied, the levels of

38 succinylation, especially on tau, should be compared to acetylation and ubiquitination.

The interaction of succinylation with other modifications is critical. The new
paragraph in the introduction places our findings in the context of the field. In the
current manuscript we focused on succinylation, because each of these post-
translational modifications require specific immuno-enrichment.

Manuscript lines 166-175

Post-translational modifications (PTMs) of proteins provide an efficient and rapid biological regulatory
mechanism that links metabolism to protein and cell functions. PTMs contribute to the functional
diversity of proteomes without the formation of new proteins or a change in their abundance by covalent
addition of functional groups that can alter protein charge, structure, and their interactions. Protein PTMs
play a central role in the pathology of neurological diseases. The function of tau can be altered via its
phosphorylation¹⁰, acetylation¹¹, methylation¹² and O-GlcNAcylation¹³. Protein succinylation of lysine
residues is a relatively novel PTM and changes the charge from positive to negative. The interactions of
lysine succinylation and acetylation play an important role in metabolic pathways¹⁴. However,
succinylation is poorly studied in the nervous system; our previous work demonstrated that lysine
succinylation functionally modifies enzymes of energy metabolism¹⁵.

There were no comparisons of the spectra of A β or MAPT peptides identified in brain
to synthetic standards.

These mass spec are in this response and Lines 225-245 in this response.

Also, the imaging has many labeling errors and does not strictly agree with the
interpretations of the authors.

Our apologies, poor coordination of co-authors led to many errors in the imaging
section. All of these have been corrected

Lines 368 - Lines 375

368 To characterize tau succinylation in a transgenic mouse model of tangle formation, we used
369 immunofluorescence staining to compare the presence of lysine succinylation within tau oligomers (T-22)
370⁴⁶ and phospho-tau (AT8) in the hippocampus from wild type and TgP301S mice. No phosphorylated tau
and few tau oligomers were present in the brain of wild type mice (Figure 6c, d and Extended Data
Figure 4a, b). The immunofluorescence signal of tau oligomers and phospho-tau was significantly
augmented in the hippocampal region of TgP301S mice⁴⁷. A parallel increase in lysine succinylation and
oligomeric tau T-22 (green) and phospho-tau (green) was observed in tau mice compared to wild type
animals (Figure 6c, d and Extended Data Figure 4a, b). Thus, tau succinylation associates with tau
aggregates in a transgenic mouse model of tauopathy.

Lines 304-312

AD-associated succinylation of APP occurred at a critical site (K612) in nine of ten brains from AD
patients but not in brains from age-matched subjects with no dementia (Figure 5a, b), and the following
experiments demonstrated it to be pathologically important. We observed an increase in the levels of
lysine succinylation and severity of amyloid burden in a transgenic mouse model of AD (Tg19959 mice),
which carries the human APP with two mutations (Figure 5c and Extended Data Figure 3a). Double
immunofluorescence staining with antibodies to pan-lysine-succinylation and to A β oligomers (NU-4)³⁶
or to A β plaque (β -Amyloid, D3D2N) revealed an early increase in lysine succinylation that appeared to
parallel oligomer formation and subsequent plaque deposition in the hippocampus. These findings suggest
that the APP succinylation might be involved in A β oligomerization and plaque formation *in vivo*.

An independent validation of the A β and Tau succinylation by some other means
(WB, immunogold microscopy, etc.) is also warranted.

The paper contains extensive evidence by mass spectrometry to support the
unequivocal direct interaction of succinylation with A β and tau in human brains by
mass spectrometry. While immunogold may strengthen the conclusions in mouse
brains, we think that this should be part a complete time course study of the mouse
pathology and mass spectrometry to assess the precise labelling. Thus, to do it
properly is beyond the scope of this manuscript.

Major Comments:

We would like to thank Reviewer 1 for the detailed and pertinent questions raised
about the mass spectrometry datasets of both succinylome and global proteome
studies. The authors agree that important information was missing in the
supplementary documents and that it should be presented to provide a better
understanding of the specific succinylation sites and peptides, their identity
confidence, their biological importance and parallel comparison with the global
proteome results. Please find below a point by point response to Reviewer 1's
comments on our manuscript.

• In what manner was the subcellular localization determined for
Fig.2a/Supplementary Table 2? Gene ontology?

Methods section line numbers 310-313

310 Subcellular localization of the identified candidates was determined using Cytoscape (version
311 3.6.1)¹⁶ and stringAPP (version 1.4.0)¹⁷ software. All the parameters were set to the default values, but
312 only these highest compartment scores equal 5 as the high confidence localization were kept. The result
313 was visualized in FunRich (version 3.1.3).

• Was the sample arrangement per each TMT batch ordered 5 CTL in a row then 5
AD in a row?

No, for global proteome per each TMT batch with 5 AD and 5 CTL, we randomized
the order of labeling each of the 10 samples by TMT10plex, we added this point in
the revised method section.

Methods line numbers 35-37

over 1 hour at r.t. The peptides from the 10 samples (5 controls and 5 AD cases) were mixed each tag
respectively with 126-tag, 127N-tag, 127C-tag, 128N-tag, 128C-tag, 129N-tag, 129C-tag, 130N-tag,
130C-tag and 131-tag. The order of labeling each of the 10 samples by TMT10plex was randomized.

• Given the low number of succinylated peptides corroborated by both cohorts,
please share a supplementary table that include the peptide data following database
search (including Xcorr, PEP, etc.)

The requested supplementary table that includes all the peptide data following
database search (including Xcorr, PEP, etc.) information was added as Supplementary
Table 7.

What are the succinylation localization scores for each succinylated peptide?

The threshold of succinylation localization score was set at PSM grouper node of
consensus workflow in Proteome Discoverer (PD) 2.2 database searching software,
and succinylation localization scores for each succinylated peptide must be greater
than 75 and it lies in between 75-100. ptmRS node was not used in processing
workflow for determination of localization scores for other PTMs because ptmRS
node was more designated for phosphorylation in PD 2.2. We added this point in the
revised method section.

Methods section lines 122-134

output to Microsoft Excel software for further data analysis. The threshold of succinylation localization
score was set at PSM grouper node of consensus workflow in Proteome Discoverer (PD) 2.2 database
searching software, and succinylation localization scores for each succinylated peptide must be greater
than 75 and it lies in between 75-100. In combination with threshold score (≥ 75) for succinylation
localization, falsely-localized succinylated peptides were further filtered out under peptide validator node
in consensus workflow where q values and PEPs are validated for available PSMs and assigned the PSMs
confidences based on the user defined target FDRs in percolator node (Target FDR-0.01). In addition,
data analysis for all identified succinylated peptides indicated that the average number of missed
cleavages for each succinylated peptides = 1 (Supplementary Table 7). This is not surprising, as lysine
succinylation would prohibit trypsin from cut the modification site creating one missed cleavage. As a
result, we found that over 90% of succinylated peptides being identified were equivalent to 0 miss-
cleavage and 10% contained 1 miss-cleavage site in our data, which is consistent with what we observed
in our regular global proteomics.

• What was the succinylation localization score threshold that you set your Proteome
Discoverer to?

As stated in the above response, threshold score for site probability was set to 75

at PSM grouper node of consensus workflow in PD 2.2, which implies
modifications (variable) with lower site probability than the specified threshold
will not be shown in the final list of succinylated peptides. We added this point in
the revised method section.

Methods section line numbers 122-135

output to Microsoft Excel software for further data analysis. The threshold of succinylation localization
score was set at PSM grouper node of consensus workflow in Proteome Discoverer (PD) 2.2 database
searching software, and succinylation localization scores for each succinylated peptide must be greater
than 75 and it lies in between 75-100. In combination with threshold score (≥ 75) for succinylation
localization, falsely-localized succinylated peptides were further filtered out under peptide validator node
in consensus workflow where q values and PEPs are validated for available PSMs and assigned the PSMs
confidences based on the user defined target FDRs in percolator node (Target FDR-0.01). In addition,
data analysis for all identified succinylated peptides indicated that the average number of missed
cleavages for each succinylated peptides = 1 (Supplementary Table 7). This is not surprising, as lysine
succinylation would prohibit trypsin from cut the modification site creating one missed cleavage. As a
result, we found that over 90% of succinylated peptides being identified were equivalent to 0 miss-
cleavage and 10% contained 1 miss-cleavage site in our data, which is consistent with what we observed
in our regular global proteomics.

• What methods were used to filter out falsely-localized succinylated peptides?

In combination with threshold score (≥ 75) for succinylation localization, falsely-
localized succinylated peptides were further filtered out under peptide validator node
in consensus workflow where q values and PEPs are validated for available PSMs and
assigned the PSMs confidences based on the user defined target FDRs in percolator
node (Target FDR – 0.01). We added this point in the revised method section at

Methods lines 122-127

output to Microsoft Excel software for further data analysis. The threshold of succinylation localization
score was set at PSM grouper node of consensus workflow in Proteome Discoverer (PD) 2.2 database
searching software, and succinylation localization scores for each succinylated peptide must be greater
than 75 and it lies in between 75-100. In combination with threshold score (≥ 75) for succinylation
localization, falsely-localized succinylated peptides were further filtered out under peptide validator node
in consensus workflow where q values and PEPs are validated for available PSMs and assigned the PSMs
confidences based on the user defined target FDRs in percolator node (Target FDR-0.01). In addition,

• Please provide the succinylation mass shift (+100.0160) used in the database
searches as well as masses for the other PTMs.

We used the following modifications with specific mass shift as variable

modifications:-1. Dynamic Modification: Oxidation / +15.995 Da (M)

151 -2. Dynamic Modification: Acetyl / +42.011 Da (K) and N-terminal of proteins.

152 -3. Dynamic Modification: Succinyl / +100.016 Da (K)

153 -4. Dynamic Modification: Deamidated / +0.984 Da (N, Q)

We have added all the information in the revised method section

Methods **section lines 116-120**

For label-free quantitative data analysis of succinylated peptides, fragment ion tolerance 0.5 Da was used
for the ion trap analyzer and an additional succinylation on Lys residue with mass shift (+100.0160 Da)
was specified as variable modifications. In addition, methionine oxidation (+15.995 Da), acetylation
(+42.011) on N-terminal proteins and deamidation (+0.984 Da) on asparagines/glutamine were also set up
as variable modifications. For each relative ratio of succinylated peptides/sites, no normalization was

• **What was the average number of MS2 ions identified in succinylated peptides**
**compared with non-succinylated peptides? The % enrichment**

**For 1st cohort of succinylome study, the average number of MS2 ions identified**
**in succinylated peptides compared with non succinylated peptides =**
**51779/150384. The % enrichment = 34.4 %.**

**For 2nd cohort of succinylome study, the average number of MS2 ions identified**
**in succinylated peptides compared with non succinylated peptides =**
**49901/149319. The % enrichment = 33.4 %.**

**Manuscript lines 243-246** were inserted.

identifying the succinylated peptides in large cohorts. After enrichment, we found that
the average enrichment of succinylated peptides was found to be 33.9% in two cohorts while 0.2% of
succinylated peptide was identified in global proteome without enrichment. Of 1,908 succinylated
peptides identified in two independent analyses, 932 succinylated peptides were quantifiable (**Figure 1a**).

• **What were the number of succinylated peptides identified in the total proteome**
**dataset without enrichment?**

For global proteome analysis in 1st cohort, the number of succinylated peptide
identified = 126 out of total 94,263 peptides (0.13%). For global proteome analysis in
2nd cohort, the number of succinylated peptide identified = 201 out of 71,367
(0.28%).

The notable difference in ratio of succinylated peptides over total peptides without
enrichment between the two cohorts of global proteome datasets is not surprising, as
we know that the succinylation has relatively low occupancy level. Therefore, there will
be an anticipated variation between two cohorts' datasets for detection of those low
abundance succinylated peptides under global and complex quantitative proteomics analysis.
This assessment also indicates that the enrichment is important for reliably identifying
the succinylated peptides in large cohorts.

**Manuscript line numbers 234-243**

**Succinylome and proteome changes in AD brains**

Completion of the human brain succinylome and global proteome analyses allowed direct comparison
between brains from controls and AD patients. Without enrichment of succinylated peptide in global
proteome data, the number of succinylated peptides identified is 0.13% total peptides for cohort 1 and
0.28% for cohort 2. The notable difference in ratio of succinylated peptides over total peptides between
the two cohorts of global proteome datasets is not surprising, as we know that the succinylation has
relatively low occupancy level. Therefore, there will be an anticipated variation between two cohorts'
datasets for detection of those low abundance succinylated peptides under global and complex
quantitative proteomics analysis. This assessment also indicates that the enrichment is important for
identifying the succinylated peptides in large cohorts. After enrichment, we found that
the average enrichment of succinylated peptides was found to be 33.9% in two cohorts while 0.2% of
succinylated peptide was identified in global proteome without enrichment. Of 1,908 succinylated

- What was the average number of missed cleavages for each succinylated peptide?

The average number of missed cleavages for each succinylated peptides = 1 (the
requested Supplementary Table 7 for succinylome data has the missed cleavage
information for each succinylated peptide). This is expected as lysine succinylation
will prohibit trypsin from cutting the modification site, creating one missed cleavage.
While small percentage (~7%) with 0 miss cleavage reflects the succinylated lysines
are located at either protein C-terminus or with the Pro residue in its carboxyl side.
Therefore, the miss cleavage ratios for the succinylated peptides we identified are
equivalent to 90% with 0 miss cleavage and 10% with 1 miss cleavage, similar to
what we observed in our regular global proteomics.

For 1st cohort:

0 missed cleavage = 175 succ peptides (7.5%)

1 missed cleavage = 1935 succ peptides (83.2%)

2 missed cleavage = 215 succ peptides (9.2%)

For 2nd cohort:

0 missed cleavage = 163 succ peptides (7.3%)

1 missed cleavage = 1849 succ peptides (82.7%)

2 missed cleavage = 224 succ peptides (10.0%)

See supplementary data 2 and a brief summary in Methods lines 129-134

data analysis for all identified succinylated peptides indicated that the average number of missed
cleavages for each succinylated peptides = 1 (Supplementary Table 7). This is not surprising, as lysine
succinylation would prohibit trypsin from cut the modification site creating one missed cleavage. As a
result, we found that over 90% of succinylated peptides being identified were equivalent to 0 miss-
cleavage and 10% contained 1 miss-cleavage site in our data, which is consistent with what we observed
in our regular global proteomics.

Please provide spectra for the A β HDSGYEVHHQKLVFFAEDVGSNK succinylated
peptide from brain and compare this to the synthetic standard (preferably heavy
labeled). The authors do a nice job characterizing the in vitro succinylation of amyloid

and Tau by MRM. Can the authors show side by side the MS/MS or MRM profiles of
these standards compared to the peptides identified in the discovery proteomics
screen?

We have added the figures. Please see Figures and tables lines 76-83 in the Figure 5a
and lines 304-306.

Fig.5a The MS/MS spectrum of the succinylated peptide from brain
(Succinylation lysine residue is highlighted in red text)

Figures and tables legends lines 76-83

**Figure 5.** Succinylation occurs uniquely on APP from AD patients, in early stages of plaque
formation in mouse models and disrupts APP processing.

**a.** Location and identity of succinylation K612 near the A β region. Residues are numbered
according to APP695 sequence. Purple amino acids refer to α - or β - or γ - cleavage sites. The red
underlined lysine refers to succinylated K612. Purple arrow represents the two central strands of
the β -sheet (Leu613-Asp619 and Ala626-Val632). Green highlights the peptide identified in the
MS. The MS spectra of the succinylated peptide from brain (Succinylation lysine residue is
highlighted in red text).

Manuscript lines 304-306

304 AD-associated succinylation of APP occurred at a critical site (K612) in nine of ten brains from AD
305 patients but not in brains from age-matched subjects with no dementia (**Figure 5a, b**), and the following
306 experiments demonstrated it to be pathologically important. We observed an increase in the levels of

**Extended Data Figure 3d.** The MS/MS spectrum of the synthetic standard peptide

(Succinylation lysine residue is highlighted in red text)

Extended data line 22-23

22 d. The MS/MS spectra of the synthetic succinylated A β ₆₋₂₉ peptide used in assay (Succinylation lysine residue is
23 highlighted in red text).

Extended Data Figure 3d

• It is surprising that the VQIVYK Tau hexapeptide was identified with the C-
terminal lysine (residue 311) succinylated (Fig. 6b). It has been previously reported
that trypsin/LysC cleavage is blocked by succinylation
(https://link.springer.com/protocol/10.1007/978-1-60327-259-9_53). The authors need
to confirm the MS/MS spectrum in AD with standards or use a targeted MS approach
to map the more likely modified VQIVYK(succ)PVDSLK peptide in AD brain that
they identified in their in vitro assays.

The highlighted sequence VQIVYK (named PHF6) highlighted in red in Figure 6a
(not Figure 6b) was intended to indicate beta-sheet structure only. The peptide
“HVPGGGSVQIVYKPVDSLK” highlighted in green was the one identified by MS
in brains. And the MS/MS spectra were added for this peptide in Figure 6a (Line 116-
125 in the Figure).

**Figure 6.** The unique succinylation of K311 on tau in brains from patients with AD promotes AD
 like features in tau pathology.

**a.** Domain structure of tau and the location of succinylation K311. The diagram shows the domain
 structure of httau23 and 24, which contain three and four repeats, respectively. The constructs K18
 and K19 comprise four repeats and three repeats, respectively. Residues are numbered according
 to tau441 sequence. Purple arrow represents the two central strands of the β-sheet (PHF6*:
 Val275-Lys280, highlighted in blue, the blue underlined lysine refers to acetylated K280; PHF6:
 Val306-Lys311, highlighted in red, the red underlined lysine refers to succinylated K311). Green
 highlights the peptide identified by MS. The MS spectra of the succinylated peptide from brain
 (Succinylation lysine residue is highlighted in red text).

 • In most published proteomic studies mitochondria proteins are decreased in AD
 within the total proteome. How do the authors reconcile this with the succinylome?
 changes in mitochondrial protein levels and succinylation are occurring, but the
 decline in protein abundance cannot account for the changes in succinylation
 peptides/sites. There are some succinylated proteins where succinylation
 peptide/site levels were increased or decreased in AD much more than the abundance
 of the corresponding proteins determined in global proteomic data. We have added the
 following to reconcile these findings.

**Manuscript lines 258-264**

patients while 73 protein levels were increased (**Extended Data Figure 2a**). In a recent large-scale
 proteomic scan, the protein abundance of PDHA, PDHB, and DLD were all decreased in AD, which is
 consistent with our finding, representing a decreased abundance of proteins in impaired mitochondrial

8

states¹⁷. Thus, changes in protein levels and succinylation may be important in AD. Relatively small fold
 changes found between control and AD brain samples, were probably due to a well-known ratio
 compression caused by the co-isolation of isobaric-labeled background ions in MS2-based TMT
 quantitative proteomics.

**Are differences in protein abundance rather than site specific changes due to the**
 **PTMs driving the decrease in AD. Can the authors provide or estimate any changes in**
 **stoichiometry measurements?**

The comparison of the AD-related changes from our proteomics and succinylomics
 indicates the changes in the succinylome are likely independent of protein changes
 (Fig. 3c). The heatmap shows the magnitude of variation in the succinylome/proteome
 as color in two dimensions. Each cell's color indicates the value of the fold change
 ($\text{Log}_2(\text{Fold Change})$). The variation in the succinylome is much larger ($|\text{Log}_2(\text{Fold}$
 $\text{Change})| > 0.3$) than the abundance changes of the same protein that happens in the
 proteome ($|\text{Log}_2(\text{Fold Change})| < 0.2$).

A total of 213 out of 229 succinylated mitochondria proteins was identified in the
 proteome. Only 37 succinylated mitochondria proteins were significantly changed
 ($p < 0.05$), in which 27 proteins (73%) were decreased. 959 quantifiable succinylated
 peptides were found in 208 succinylated mitochondria proteins. Only 21 succinylated
 peptides from mitochondria proteins were significantly changed ($p < 0.05$), in which 21
 succinylated peptides (71%) were decreased. Only 4 mitochondria proteins
 significantly change at protein level accompanied by a significant alteration of the
 succinylated peptide level (5 succinylated peptides), which are listed below.

GI Number	UniProtKB	Entry name	Succinylome		Proteome	
			Log ₂ FC	p-value	Log ₂ FC	P-value
129379	P10809	HSPD1	0.52	0.04294	-0.08	0.0255
21542295	Q9NVH6	TMLHE	0.39	0.01607	0.16	0.0054
			0.49	0.04928		
20455474	P24539	ATP5F1	-0.32	0.04433	-0.14	0.0182
			-0.47	0.04788		

3.08E+08	P00505	GOT2	-0.63	0.00561	-0.11	0.0126
-----------------	---------------	-------------	--------------	----------------	--------------	---------------

Manuscript line numbers 197-200

197 peptides being identified were equivalent to 0 miss-cleavage and 10% contained 1 miss-cleavage site in
 198 our data, which is consistent with what we observed in our regular global proteomics. The parallel global
 199 proteomic analysis detected 4,678 proteins (Figure 1d). Nearly all of the succinylated proteins identified
 200 during the study were found in the global proteome of the same samples (Figure 1e).

Manuscript line numbers 253-264

controls (Figure 1d and Extended Data Figure 2a, b). A comparison of the succinylome with the
 proteome demonstrated little AD-related changes in protein levels of those succinylated proteins, and
 therefore the succinylation variations are most likely independent from the changes of the corresponding
 protein abundance (Figure 3c). The proteomic analysis showed that 81 proteins changed significantly
 (two-tailed Student's *t*-test, $p < 0.05$ and $|\log_2FC| > 0.25$). Eight proteins decreased in brains from AD
 patients while 73 protein levels were increased (Extended Data Figure 2a). In a recent large-scale
 proteomic scan, the protein abundance of PDHA, PDHB, and DLD were all decreased in AD, which is
 consistent with our finding, representing a decreased abundance of proteins in impaired mitochondrial

8

261 states¹⁷. Thus, changes in protein levels and succinylation may be important in AD. Relatively small fold
 262 changes found between control and AD brain samples, were probably due to a well-known ratio
 263 compression caused by the co-isolation of isobaric-labeled background ions in MS2-based TMT
 264 quantitative proteomics.

It is not clear why the pan-lysine succinyl antibody labeling so drastically attenuated
 in WT 10 month old mice?

Our chemistry test tube experiments are clearly consistent with our hypothesis
 that succinylation can promotes plaques and tangles. The goal of the mouse studies is
 to show an association of succinylation in the brain to APP or tau. The data clearly
 show that in mice that are four months old. We do not know the precise relation of
 succinylated tau or APP to the final pathology (i.e., tangles and plaques). One can
 imagine scenarios where they promote formation but not be high in final product.

The four-month data clearly shows an association of succinylation to tau and
 amyloid. Interpretation of the ten-month data, which includes the maturation process,
 adds confusion not clarity. Thus, we have chosen to omit the ten-month data. These
 results enhance our enthusiasm for the current study, because they reveal new exciting
 areas to be developed. We changed the manuscript to reflect the changes and the
 interpretation.

Manuscript line numbers 308-312, Lines 372-376

which carries the human APP with two mutations (Figure 5c and Extended Data Figure 3a). Double
immunofluorescence staining with antibodies to pan-lysine-succinylation and to A β oligomers (NU-4)³⁶
or to A β plaque (β -Amyloid, D3D2N) revealed an early increase in lysine succinylation that appeared to
parallel oligomer formation and subsequent plaque deposition in the hippocampus. These findings suggest
that the APP succinylation might be involved in A β oligomerization and plaque formation *in vivo*.

**Figure 4a, b).** The immunofluorescence signal of tau oligomers and phospho-tau was significantly
augmented in the hippocampal region of TgP301S mice⁴⁷. A parallel increase in lysine succinylation and
oligomeric tau T-22 (green) and phospho-tau (green) was observed in tau mice compared to wild type
animals (Figure 6c, d and Extended Data Figure 4a, b). Thus, tau succinylation associates with tau
aggregates in a transgenic mouse model of tauopathy.

Nevertheless, we have chosen to add a speculative answer to the reviewer's concern, but this has not been added to the text because we omitted the 10 month values.

Succinylation is a post-translational modification and several factors can regulate the balance between succinylation and desuccinylation. Most of these are unknown in brain. Our findings show that KGDHC is a major succinyl transferase in neurons. Brain KGDHC is not altered with age out to 30 months (Freeman, Nielsen et al. 1987) suggesting that the age-related change is not a reduction in active succinylation. Whether aging may alter KGDHC migration to the cytosol has never been studied. The desuccinylases in brain remain unknown. A prominent paper by two of our co-authors have shown that sirtuin 5 (SIRT5) plays a central role in modulating heart metabolism and function (Sadhukhan, Liu et al. 2016). SIRT5 is localized in the mitochondria and shows a weak deacetylase activity but a potent desuccinylase activity on lysine residues both *in vitro* and *in vivo* (Park, J 2013 Mol. Cell; Du et al., 2011; Peng et al., 2011). The catalytic reaction involves the removal of a succinyl group from the lysine side chain of protein substrates, a process that consumes NAD⁺ as a co-substrate and generates nicotinamide (NAM) and 2'-O-succinyl-ADP-ribose (Rardin MJ 2013 Cell Metabol). SIRT5 KO mouse embryonic fibroblasts display an increase in lysine succinylation but not acetylation (Du et al. 2011 Science). We have used SIRT5 to desuccinylate enzymes such as the pyruvate dehydrogenase complex. We have also looked at succinylation in SIRT5 KO mice, which show a significant increase in succinylation levels in the liver while trivial changes were found in the brain. Liver succinylation, but not that in brain responds to fasting. Furthermore, the data from the AD samples suggest that different desuccinylases are likely important in the cytosol and mitochondria. We believe that our current results justify further studies on the regulation of succinylation in the brain during aging and in neurodegenerative diseases.

And why there such an increase in succinylation in Tg19959 mice? Authors need to resolve.

Please note that this also occurs in P301S mice. We think that the widespread increase

in succinylation shows that the transgenes are causing widespread changes in multiple
proteins including tau and APP. We know the pathological implications for tau and
APP but it is likely changes in other proteins are likely important. This is yet another
important area of research opened by the current results. We have added the following
sentences to the paper.

To the results section Manuscript lines 306-312

experiments demonstrated it to be pathologically important. We observed an increase in the levels of
lysine succinylation and severity of amyloid burden in a transgenic mouse model of AD (Tg19959 mice),
which carries the human APP with two mutations (Figure 5c and Extended Data Figure 3a). Double
immunofluorescence staining with antibodies to pan-lysine-succinylation and to A β oligomers (NU-4)³⁶
or to A β plaque (β -Amyloid, D3D2N) revealed an early increase in lysine succinylation that appeared to
parallel oligomer formation and subsequent plaque deposition in the hippocampus. These findings suggest
that the APP succinylation might be involved in A β oligomerization and plaque formation *in vivo*.

The P301S mice reveals a widespread increase in background succinylation.

Lines 372-376 in the manuscript.

Figure 4a, b). The immunofluorescence signal of tau oligomers and phospho-tau was significantly
augmented in the hippocampal region of TgP301S mice⁴⁷. A parallel increase in lysine succinylation and
oligomeric tau T-22 (green) and phospho-tau (green) was observed in tau mice compared to wild type
animals (Figure 6c, d and Extended Data Figure 4a, b). Thus, tau succinylation associates with tau
aggregates in a transgenic mouse model of tauopathy.

To the discussion section

The results reveal that both transgenic mice strains reveal widespread increases in
succinylation, which suggests that many proteins in addition to tau and APP are
altered. Determining whether this is an artifact of the transgene or a down- stream
consequence of the abnormal tau and APP remains to be determined.

The manuscript lines 439-447 now read

We show that transgenic mouse strains of either tauopathy or amyloidosis phenotype including plaques
exhibit widespread increases in lysine succinylation, which is not exclusive to tau and APP, but parallels
the formation of pathological species. While proteins in addition to tau or APP maybe altered, APP and
tau are only succinylated in brains from AD patients, which suggests that increased tau and APP
succinylation may play a role in the development of AD pathology. Thus, the modification of metabolism
in disease may lead to critical succinyl-mediated modifications of extra-mitochondrial proteins including
APP and tau leading to aggregation and deposition. Preventing APP and tau succinylation and/or
increasing mitochondrial succinylation may provide novel therapeutic targets for the prevention and/or
treatment of AD and tauopathies.

• Are the genotype and time point labels switched in Figure 5d? The images do not
agree with the quantitation

We apologize for the mistake. They are all correct in the revised version. This has
been amended in the revised version. See Figures 5c and 5d.

Also, why is there so little co-localization with A β plaque and the succinyl-lysine
antibodies? Authors need to explain.

We have now provided improved images at low and high magnification to show the
co-localization (white) better. We have also omitted the data from the 10 month-old
mice. We do not postulate that APP and tau are the only cytosolic proteins
succinylated. We used a pan succinylation antibody. As shown by the succinylomics
data, hundreds of proteins are succinylated. Succinylation is not evenly distributed
among all proteins as some contain a higher number of succinylated sites than others
despite containing a similar amount of total lysines, suggesting site-specificity of
succinylation. Nevertheless, we do see co-localization of succinylation with tau and
APP oligomers.

We added the following sentences to the manuscript lines 442-450.

We show that transgenic mouse strains of either tauopathy or amyloidosis phenotype including plaques
exhibit widespread increases in lysine succinylation, which is not exclusive to tau and APP, but parallels
the formation of pathological species. While proteins in addition to tau or APP maybe altered, APP and
tau are only succinylated in brains from AD patients, which suggests that increased tau and APP
succinylation may play a role in the development of AD pathology. Thus, the modification of metabolism |
in disease may lead to critical succinyl-mediated modifications of extra-mitochondrial proteins including
APP and tau leading to aggregation and deposition. Preventing APP and tau succinylation and/or
increasing mitochondrial succinylation may provide novel therapeutic targets for the prevention and/or
treatment.²⁴

Since a pan anti-succinylation antibody was used, many proteins are labelled and the immunostaining for
succinylation shows broad distribution. Thus, the staining would not be expected to specific to A β
plaques and NFTs. The results support that there is an association and co-localization of succinylation
with plaques and NFTs, which is particularly prominent at early stages. Further, succinylation is not
evenly distributed among all proteins, as some contain a higher number of succinylated sites than others
despite containing a similar amount of total lysines. Thus, we would not expect succinylation to be
limited to plaques and tangles. APP and tau were only succinylated in brains from AD patients. Thus, the

• In Figure 6c the authors label the images with a mouse model (Tg19959) that does
not match the mouse model described in the text (TgP301S).

We apologize for the mistake. We have now labeled the figures correctly.

Lines 128-134 in Figures and Tables.

c. High resolution images acquired using confocal laser microscopy display the co-localization of
succinylation and tau oligomers in the hippocampus of TgP301S and WT mice. A T22 antibody
(green) was used to stain tau oligomers while a pan-succinyl-lysine antibody (magenta) labeled

the levels 131 of succinylation. Results were expressed as the mean with SEM representative of
the average of ~900-1000 pyramidal neurons comprised in 3-4 different brain sections per
animal (n = 4 per each group). ****: $p < 0.0001$, two-way ANOVA followed by Tukey's multiple
comparisons test.

• Again, there is little co-localization with aggregated Tau antibody IF labeling and
pan-lysine succinylation. One would expect extensive co-localization if succinylation
labels the core of PHFs, where this PTM maps.

The white shows clear co-localization. Since we used a pan succinylation antibody,
many other succinylated proteins are present. Further, succinylation is not evenly
distributed among all proteins as some contain a higher number of succinylated sites
than others despite containing a similar amount of total lysines, Thus, we would not
expect succinylation to be limited to tangles.

Further, we do not know the relation of PTM to tangle maturation. Tau
acetylation has been studied for decades and detailed mechanisms are still unknown,
but acetylation has many consequences on brain function besides just tangle
formation (Tracy, Claiborn et al. 2019) Tau acetylation-induced pathogenesis may
involve regulation of toxic forms of the protein, such as hyperphosphorylated protein,
in which the consequences are site-specific. Tau acetylation can modulate tau toxicity
by altering the formation of cleaved-caspase tau fragments. Acetylation at some sites
is sufficient to drive synaptic and cognitive deficits without producing tau fragments.
Tau acetylation may also affect the formation of tau oligomers and aggregates.
Acetylation of tau lysines blocks those residues from being targeted for
ubiquitination, slowing the rate of protein turnover and leading to accumulation.
Whether acetylated tau propagates from cell-to-cell in the brain is unknown, and
understanding if this property underlies its toxicity is an area of active research
(Tracy, Claiborn et al. 2019). Since our paper is the first paper on succinylation of tau,
it is not surprising that we do not know how all the pieces fit together. This means this
paper is opening a whole new area of research.

We have added the following to the text on **Manuscript lines 442-450**

We show that transgenic mouse strains of either tauopathy or amyloidosis phenotype including plaques
exhibit widespread increases in lysine succinylation, which is not exclusive to tau and APP, but parallels
the formation of pathological species. While proteins in addition to tau or APP maybe altered, APP and
tau are only succinylated in brains from AD patients, which suggests that increased tau and APP
succinylation may play a role in the development of AD pathology. Thus, the modification of metabolism
in disease may lead to critical succinyl-mediated modifications of extra-mitochondrial proteins including
APP and tau leading to aggregation and deposition. Preventing APP and tau succinylation and/or
increasing mitochondrial succinylation may provide novel therapeutic targets for the prevention and/or
treatment.²⁴

This the first study of tau succinylation, and the precise relation to tangle formation is
unknown. Tau acetylation has been well-documented for over a decade, and its
precise role is still unknown. **Manuscript lines 461-470**

Overall, our data represent the first report of the human brain succinylome and its implications for
mitochondrial function as well as for the molecular pathogenesis of amyloidosis and tauopathy, two of the
cardinal features of AD. We provide a rich resource for functional analyses of lysine succinylation, and
facilitate the dissection of metabolic networks in AD. The current studies also lay the foundation for
future investigation into the crosstalk between different PTMs, including acetylation, phosphorylation,
ubiquitination and succinylation associated with AD and tau pathology. The discovery that succinylation
links mitochondrial dysfunction to amyloidosis and tauopathy may provide new molecular diagnostics as
well as potential targets for therapies. Since both succinylated A β and tau are closely associated with
disease state, future investigations may reveal additional succinylated proteins that are associated with
AD or other neurodegenerative diseases.

• The authors state on Page 13, Lines 309-310 to Page 14, Lines 311-312: “a weak
signal for succinylated tau occurred in 10-month-old TgP301S” ... “indicating a
desuccinylation process may exist in the final states of tau deposition”. Also, with no
evidence, the authors offer a phosphorylation-succinylation switch as a possible
reason.

The decline in succinyl lysine signal in 10-month-old mice has been discussed
elsewhere in this response above **(lines 309-352 and 327-352)**. The data has been
withdrawn from the manuscript.

As discussed in more detail **above (Lines 309-352 of this response)** in the response),
acetylation facilitates phosphorylation, one of the defining features of tangles. No
such studies exist for succinylation, and we are hoping our findings will encourage
these studies.

• Authors state that Both APP and Tau were highly succinylated at critical sites in
nine out of ten AD brain samples, but no succinylation of APP or tau was detectable in
any control brains. Just because these sites were detected only in AD does not
necessarily mean they are highly succinylated.

**We agree. We have deleted the words highly succinylated from the manuscript.**

**What about lysine acetylation or ubiquitination? Are these sites on tau modified by**
**these PTMs? Did the authors search their data for these PTMs at the same sites?**

**Lysines are highly modified residues and one would expect succinylation to be a**
**relatively minor pool of modified Tau in the AD brain. The authors should measure**
**the pools of these PTMs (succinylation, ubiquitination, acetylation) to assess the**
**relative frequency of succinylation vis a vis acetylation and ubiquitination.**

**Studies of PTM** require immuno-enrichment of for each modification. It is
difficult to identify the lysine acetylation or ubiquitination in the global proteome
analysis without enrichment as those modifications generally have relatively low
abundance. Hence, to study the succinylation effect in AD patients, it is required and
necessary for enrichment of succinylated peptides prior to nano LC-MS/MS analysis.

Consequently, we did not search these modifications on tau in their data sets. We
also did not search our datasets for other PTMs because we used the specific
enrichment strategy specifically for succinylated peptides. Hence, it would not be
possible to detect other modifications than succinylation even if they are present in
the original samples.

We have searched the uniprot database, the lysine311 has been identified with
ubiquitination, dimethylation, and acetylation in transgenic mice (Morris, Knudsen et
al. 2015), not human brains. So far, these modifications (ubiquitination,
dimethylation, and acetylation) of this site (K311) have not been reported on relevant
to the mechanism of Tau protein pathology.

In additional study from our group we enriched with an anti-acetylation antibody
to identify lysine acetylation modification and its change with AD. That is a whole
new paper is being reviewed now.

We added the following to the discussion **(lines 452-459 in the manuscript)**

Lysines are highly modified residues. Understanding the relationship of succinylation to the other PTM is
critical to a complete understanding of its role in AD pathology. A direct comparison is practically
difficult because each PTM requires a different enrichment strategy. Lysine311 has been associated with
ubiquitination, dimethylation, and acetylation in transgenic mice⁵⁹, but not human brains. Our
experiments raise the question of whether precise changes in mitochondria are required to alter
modification of specific proteins. Succinylation appears directly linked to KGDHC and mitochondria.
Whether other post-translational modifications are also linked to mitochondrial dysfunction remains to be
determined.

• **The weak signals on Tau contradict the human data of increased succinylated Tau.**
**How do the authors reconcile these findings? Could other PTMs on these sites**
**(ubiquitin, acetylation) also occur or do mice have increased SIRT5 (or other**
**SIRT/desuccinylase enzymes) abundances? The global human proteomes analyzed**
**showed no changes in SIRT5 as the authors note in the manuscript in Extended Data**
**Fig. 2c. Could this differ in the mice?**

**The succinyl lysine signal in plaques and tangles in mice has been discussed above**

(Lines 295-338 of this response). As discussed previously, the succinylases and
 desuccinylases in brain are not known. Tau is well-known to be phosphorylated and
 acetylated. We have an active program looking at these interactions. Studies of
 phosphorylation have shown the degree of phosphorylation is not necessarily related
 to the functional implications. Protein levels of SIRT5 do not necessarily reflect
 activity, which can be regulated by substrates and post-translational modification.
 Whether or not SIRT5 is the primary desuccinylase in brain is unknown.
 The following is in the manuscript lines 181-187 and manuscript lines 436-440

which the succinyl donor is presumably succinyl-CoA, both in yeast²¹ and cultured neurons^{15,23,24}. Studies
 of organisms deficient in NAD⁺-dependent desuccinylase sirtuin 5 (SIRT5)²⁵ provide evidence of the
 regulatory importance of succinylation in metabolic processes²⁶⁻³⁰. However, the role of succinylation in
 metabolic pathways of the human nervous system or in neurodegenerative diseases is unknown. Our
 report represents the first investigation of the human brain succinylome and its changes in AD. The
 results suggest that succinylation may link AD-related metabolic deficits to structural, functional, and
 pathological alterations involving APP and tau.

from mitochondria to other cellular compartments⁵⁸. The decline in succinylation of mitochondrial
 proteins, appears due to a failure in maintaining succinylation levels, and may suggest that activation of
 desuccinylases (e.g., Sirtuins) or general increases in NAD⁺ should be reconsidered. The large increase
 in succinylation in 4-month-old Tg19959 mice agrees with our hypothesis, in which abnormal
 mitochondrial function in AD promotes the release of KGDHC and subsequent increases succinylation.

• The authors should visit larger proteomic datasets for more accurate steady-state
 protein level quantification in AD cohorts that can fully describe ADAM/SIRT family
 levels in AD as compared with controls.

In the paper by (Seyfried, Dammer et al. 2017), a total of 2745 proteins in two regions
 (dorsolateral prefrontal cortex (FC, Brodmann Area 9) and precuneus (PC, Brodmann
 Area 7) were quantified. The number that overlapped was about 2332 proteins
 (85.3%) compared with our data (4442 proteins from 10 controls and 10 AD,
 Brodmann area 44/45).

The four ADAM family members identified in that paper were also identified in our
 proteome. The protein level of ADAM 10, 22, 23 did not change in that paper nor our
 data, while ADAM11 showed a similar decrease in the two cases. The paper only
 identified SIRT2 and SIRT5. However, the SIRT family levels did not vary in the two
 cases. Furthermore, protein levels of SIRT5 do not necessarily reflect activity, which
 are often regulated by substrates and post-translational modifications.

Symbol	Unique ID	Frontal Cortex (FC)			Precuneus (PC)		
		p value	Tukey's	log ₂ (FC)	p value	Tukey's	log ₂ (FC)
		ANOVA	AD-CTL	FC (AD-CT)	ANOVA	AD-CTL	PC (AD-CT)
ADAM10	O14672	0.051817	0.623137	0.234946	0.613604	0.691346	0.147285
ADAM23	E7EWD3	0.054739	0.194228	0.1514	0.242464	0.230508	0.198556

ADAM22	Q9P0K1	0.598328	0.589506	0.065565	0.098855	0.110043	0.142058
ADAM11	B4DKD2	0.016906	0.043947	-0.27101	0.000878	0.035282	-0.24924
SIRT2	Q8IXJ6	0.168041	0.868595	0.116332	0.722619	0.782255	0.088465
SIRT5	Q9NXA8	0.295005	0.294575	-0.16512	0.413197	0.465075	0.136716

The following has added to the manuscript lines 318-330

bond, and missense mutation at K612N produces early onset AD³⁷. Furthermore, global proteomics
showed an increase in β -secretase (BACE1) abundance of 31% in AD brains compared to controls
(Supplementary Data Table 6), while no changes occurred for either α -secretase or the sirtuins (SIRT)
family (Extended Data Figure 2c). Seyfried et al., quantified a total of 2,745 proteins in two regions
(dorsolateral prefrontal cortex (FC, Brodmann Area 9) and precuneus (PC, Brodmann Area 7)
were quantified. The number that overlapped was about 2,332 proteins (85.3%) compared with
our data (4442 proteins from 10 controls and 10 AD, Brodmann area 44/45).

The four ADAM family members identified in that paper were also identified in our proteome.
The protein level of ADAM 10, 22, 23 did not change in that paper nor our data, while ADAM11
showed a similar decrease in the two cases. SIRT2 and SIRT5 levels did not vary³⁸. Further,
protein levels of SIRTs do not necessarily reflect activity, which are often regulated by substrates
and post-translational modifications.

- Can the authors isolate AD brain PHF tau aggregates then perform immunogold labeling with the pan lysine succinylation antibody? Also for Abeta plaques? Then with a non-AD tauopathy control (ex: Corticobasal degeneration with prominent K311 Acetylation?)

It would be interesting do measure succinylation by mass spec within the plaques and tangles by mass spec and compare with the tau and APP not in the plaques and tangles. Such experiments are beyond the scope of the current manuscript. We feel that are mass spec studies already show succinylation definitively. We even know the precise site of succinylation.

Describing that your in vitro tau fibrils look like AD brain fibrils doesn't stand up. This would be an independent validation of your mass spectrometry experiments

We agree that we overstated the conclusion. We changed the text to indicate that we increased aggregation. The formation PHF is complicated process that cannot be mimicked in a one protein system.

Please see lines 388-393 in the manuscript.

protein (Figure 6e). Longer incubation time (24 h) with PHF6, S-PHF6, and a 90%/10% mixture was
visualized by EM (Figure 6f-h). All the reactions exhibited fibrils with a typical paired helical filament
appearance. However, the succinylated peptide formed abundant, short filaments, a feature of brain-
derived Alzheimer PHFs⁴⁹⁻⁵¹, while unmodified PHF6 filaments are longer and sparser, morphologies
more typical of recombinant tau peptide fibers (Figure 6i and 6j). Thus, both ThS and EM results support
an important role of succinylation in promoting pathological tau aggregation.

We observed each individual fiber in normal PHF6 showed ~15 nm in width

and ~107 nm of crossover repeat, while a width of ~12 nm and periodically
appearing twists every ~86 nm are found in the succinylated PHF6 and mixture.

Morphologically speaking, the normal PHF6 fibers are much longer than
these succinylated ones. Compared with the reported electron micrographs of
PHFs from AD brain or assembled from recombinant tau peptides
(10.1021/bi0357006), the normal PHF6 is just like these recombinant tau peptides
with morphology, while the succinylated one is much like brain-derived
Alzheimer PHFs characteristics, short and in a mess. However, the mixture
seems much more like the normal PHF6 but it does have some small parts mixed
among the main fibers. See the following text in the manuscript lines 384-393.

In contrast, PHF6 and K311-succinylated PHF6 (S-PHF6) fibrillated by 20 min and 80 min, respectively
(Figure 6e). The aggregation of PHF6 was remarkably accelerated by the K311 succinylation. A
substantial enhancement of PHF6-induced aggregation occurred even with a mixture containing 90%
PHF6 and only 10% S-PHF6, suggesting that succinylated tau can promote aggregation of unmodified
protein (Figure 6e). Longer incubation time (24 h) with PHF6, S-PHF6, and a 90%/10% mixture was
visualized by EM (Figure 6f-h). All the reactions exhibited fibrils with a typical paired helical filament
appearance. However, the succinylated peptide formed abundant, short filaments, a feature of brain-
derived Alzheimer PHFs⁴⁹⁻⁵¹, while unmodified PHF6 filaments are longer and sparser, morphologies
more typical of recombinant tau peptide fibers (Figure 6i and 6j). Thus, both ThS and EM results support
an important role of succinylation in promoting pathological tau aggregation.

The authors should include PHF6-K311Acetyl as a positive control to compare the
succinylated peptide for the Tau Self-aggregation assay.

We purchased the requested peptides from GenScript. At peptide concentration of 1
μM in the presence of 2 nM heparin, neither PHF6 nor A-PHF6 fibrillated during a
90-min incubation period. Since there is no data to support/report that K311
acetylation can promote tau self-aggregation. We think the unmodified PHF6 can
serve as an adequate positive control instead of A-PHF6, as it is well documented to
aggregate under these conditions in the main test. Thus, we did not add these
experiments to the text.

Fig. Tau peptides concentrations were 1 μ M in presence of 2 nM heparin: PHF6 (●),
S-PHF6 (■), A-PHF6 (▲). Experiments were performed in quadruplicate and
repeated three times with similar results. All values in the present graph were
expressed as mean \pm SEM.

**Minor Comments:**

- • Fig. 2A: change “succinlyated” to “succinylated”

We modified the manuscript. Please see line 18 in the Figures and Tables.

18 a. Subcellular distribution of succinylated-K proteins identified by ~~Cytoscape~~ and ~~stringAPP~~

- • Fig. 2B: change “succinlated” to “succinylated”

We modified the manuscript. Please see line 20 in the Figures and Tables.

20 b. Overlap of succinylated-K proteins located in the mitochondrion, nucleus, cytosol and plasma

- • Change APP770 to APP695 (as well as accompanying notations), as APP695 is the
major neuronal species (Fig. 5)

These have all been corrected in the manuscript, Figure 5 and extended data figures.

See lines 78 and 79 in the figures and tables.

a. Location and identity of succinylation K612 near the A β region. Residues are numbered

according to APP695 sequence. Purple amino acids refer to α - or β - or γ - cleavage sites. The red

- • Specify the mass shift specific to lysine succinylation in the methods

We add the following to the methods lines 91-93

For label-free SuccK peptides analysis, one MS survey scan was followed by 3 second “Top
Speed” data-dependent CID ion trap MS/MS scans with normalized collision energy of 30%. Dynamic
exclusion parameters were set at 1 within 45 s exclusion duration with ± 10 ppm exclusion mass width.

- • There are some very minor fold changes between the control and AD brain sample
proteins in Extended Data Figure 2.

Is this due to MS/MS spectral interference and compression. Can the authors discuss.

Yes, that is correct. This is a common issue for TMT labeled quantitative proteomics based on
MS/MS fragmentation. It is well known that the co-isolation of near isobaric-labeled
background ions causes spectral interference resulting in ratio distortion or ratio compression.

The main text was modified lines 261-264

261 states¹⁷. Thus, changes in protein levels and succinylation may be important in AD. Relatively small fold
262 changes found between control and AD brain samples, were probably due to a well-known ratio
263 compression caused by the co-isolation of isobaric-labeled background ions in MS2-based TMT
264 quantitative proteomics.

• Page 11, Line 252-253: The Tg19959 line contains three mutations:

Please see the following in the methods lines 398-403

All the experiments were carried out in four and ten-month-old transgenic mouse models of AD.
Tg19959 mice (that overexpress a double mutant form of the human amyloid precursor protein) were
obtained from Dr. George Carlson (McLaughlin Research Institute, Great Falls, MT, USA). Tg19959
mice were constructed by injecting FVB X 129S6 F1 embryos with a cosmid insert containing
human APP₆₉₅ with 2 familial AD mutations (KM670/671NL and V717F), under the control of the
hamster PrP promoter.²⁵ P301S (PS19, that overexpress the human tau gene harboring the P301S

12

• Page 9, Lines 205-206. What was the correlation between change in succinylation
and change in protein abundance? Please illustrate/state

We have modified this section in the manuscript lines 250-264.

between AD and control subjects (Figures 3a, b). Ten succinylated peptides were increased while
succinylation of 19 peptides declined in AD. Proteomic analysis of 20 samples in two cohorts (Figure 1c)
showed that of the 4,678 identified proteins, 4,442 common proteins were quantifiable in both AD and
controls (Figure 1d and Extended Data Figure 2a, b). A comparison of the succinylome with the
proteome demonstrated little AD-related changes in protein levels of those succinylated proteins, and
therefore the succinylation variations are most likely independent from the changes of the corresponding
protein abundance (Figure 3c). The proteomic analysis showed that 81 proteins changed significantly
(two-tailed Student's *t*-test, $p < 0.05$ and $|\log_2FC| > 0.25$). Eight proteins decreased in brains from AD
patients while 73 protein levels were increased (Extended Data Figure 2a). In a recent large-scale
proteomic scan, the protein abundance of PDHA, PDHB, and DLD were all decreased in AD, which is
consistent with our finding, representing a decreased abundance of proteins in impaired mitochondrial

8

261 states¹⁷. Thus, changes in protein levels and succinylation may be important in AD. Relatively small fold
262 changes found between control and AD brain samples, were probably due to a well-known ratio
263 compression caused by the co-isolation of isobaric-labeled background ions in MS2-based TMT
264 quantitative proteomics.

• Has acetylation been observed at A β K16?

No one has reported this. We did not examine the acetylation since we focused on
 succinylation. We anticipate acetylation is probably similar to succinylation with relatively
 low occupancy rate. Therefore, an enrichment step using anti-acetylated Lysine antibody prior
 to LC-MS/MS analysis would be required to confidently identify acetylation sites.

• The authors should provide High-resolution images for Extended Data Figure 3a-b?

We agree. Figures 3a and 3b have been replaced with high resolution figures. have
 modified this. Please see Extended Data

• Please provide a representative image for cleaved A β 6-29 fragment precursor ion
 peaks in Extended Data Figure 3. The plot in Extended Figure 3f-g is confusing

**Table: MRM parameters with their retention time of targeted peptides and their**
 **fragments:**

Compounds	RT (min)	Precursor ion m/z; (z)	Compounds parameter		Fragment ion m/z; b/y ions
			DP (V)	CE (V)	
Peptide 1	5.64	675.826; (4)	50	30	876.406; (y9)
Peptide 2	5.71	700.830; (4)	50	30	815.884; (b13 ²⁺)
Fragment 1	1.2	668.805; (2)	50	40	788.321; (b7)
Fragment 2	6.21	691.851; (2)	50	30	876.406; (y9)
Fragment 3	3.87	479.545; (3)	50	25	560.210; (b5)

**Peptide 1: HDSGYEVHHQKLVFFAEDVGSNKG**

**Peptide 2: HDSGYEVHHQK(100.016)LVFFAEDVGSNKG**

**Fragment 1: HDSGYEVHHQK**

**Fragment 2: LVFFAEDVGSNKG**

**Fragment 3: HDSGYEVHHQK(100.016)**

**RT: retention time, DP: declustering potential, CE: Collision energy**

**Here, we performed both A β 6-29 and cleavage fragments measurement. We analyzed**
 **changes in these peptides incubated with rhADAM10 samples relative to the control**

peptides (without rhADAM10). For the fragments measurement, we set the results of
fragments from the 24 hrs incubation with rhADAM10 as 100%.

The figure was also changed to make a new and more clear representation figure as Fig 3g
where only cleaved A β 6-29 fragment precursor ion changes were shown with incubation
time.

We modified the legend to figure 3 in extended data lines 25-26

f. Multiple Reaction Monitoring (MRM) parameters used in assay for quantitation with their retention time of
targeted peptides and their fragments.

• What is “Percentage Change (%)” quantified in Figure 5f?

We modified Figure 5F to clarify the percentage change

The percent is simply the ratio with and without succinyl-lysine

• Page 13, Line 300: effects of what?

We have modified this. Please see Line 363-366 in the manuscript.

Previous studies indicate the removal of residue K311 in PHF6 abrogates fibril formation⁴⁵, but the
structural and functional implications of K311 succinylation are unknown. Thus, exploring the influence

of tau succinylation on K311 may be important as we seek to develop a comprehensive understanding of
its biological functions.

• Page 13, Line 302: “IF staining to compare the presence or abeta of succinylation

with that of tau” – What does this mean

We have modified this. Please see **Lines 363-376** in the manuscript.

**To characterize tau** succinylation in a transgenic mouse model of tangle formation, we used
immunofluorescence staining to compare the presence of lysine succinylation within tau oligomers (T-22)
⁴⁶ and phospho-tau (AT8) in the hippocampus from wild type and TgP301S mice. No phosphorylated tau
and few tau oligomers were present in the brain of wild type mice (**Figure 6c, d and Extended Data**
**Figure 4a, b**). The immunofluorescence signal of tau oligomers and phospho-tau was significantly
augmented in the hippocampal region of TgP301S mice⁴⁷. A parallel increase in lysine succinylation and
oligomeric tau T-22 (green) and phospho-tau (green) was observed in tau mice compared to wild type
animals (**Figure 6c, d and Extended Data Figure 4a, b**). Thus, tau succinylation associates with tau
aggregates in a transgenic mouse model of tauopathy.

• Page 13, Line 304: “but in 4-month-old TgP301S mice”.

o What occurred in 4-month-old TgP301S mice?

We have modified this. Please see **Lines 363-376** in the manuscript.

**To characterize tau** succinylation in a transgenic mouse model of tangle formation, we used
immunofluorescence staining to compare the presence of lysine succinylation within tau oligomers (T-22)
⁴⁶ and phospho-tau (AT8) in the hippocampus from wild type and TgP301S mice. No phosphorylated tau
and few tau oligomers were present in the brain of wild type mice (**Figure 6c, d and Extended Data**
**Figure 4a, b**). The immunofluorescence signal of tau oligomers and phospho-tau was significantly
augmented in the hippocampal region of TgP301S mice⁴⁷. A parallel increase in lysine succinylation and
oligomeric tau T-22 (green) and phospho-tau (green) was observed in tau mice compared to wild type
animals (**Figure 6c, d and Extended Data Figure 4a, b**). Thus, tau succinylation associates with tau
aggregates in a transgenic mouse model of tauopathy.

• Figure 6d: “d” is covered up by panel c

Corrected

• Please label Fig.6f-h with the peptide at hand (f=PHF6, g=S-PHF6, h=9:1 mix)

We have modified the legends Figure 6f-6h in the “figures and tables” (**lines 148-**
**150**)

**f-h.** Negative stain electron microscopy of *in vitro* polymerized PHFs after 24 hrs incubation. f: 50
μ M PHF6; g: 50 μ M S-PHF6; h: 50 μ M mixture (PHF6:S-PHF6=9:1). White arrows denote paired
helical filaments. Scale bar is 100 nm.

• Page 17, Line 394: “bot amyloidosis”?

We have modified this. Please see Line **461-463** in the manuscript.

Overall, our data represent the first report of the human brain succinylome and its implications for
mitochondrial function as well as for the molecular pathogenesis of amyloidosis and tauopathy, two of the
cardinal features of AD. We provide a rich resource for functional analyses of lysine succinylation, and

• Page 8 Line 179: “Since no specific motifs for lysine succinylation in human cells
have been reported” – They have been reported in HeLa cells in Weinert et al., Cell
Reports (2013) <http://dx.doi.org/10.1016/j.celrep.2013.07.024>. Since they have
reported previously, the authors should compare Motifs in AD brain to theirs, which
upon first glance, don’t exactly match.

We modified this **Lines 223-226** in the manuscript.

223 Since no specific motifs for lysine succinylation in human tissues have been reported, a succinylation
224 motif analysis of all 1908 succinylated peptides using Motif-X was performed. A total of five conserved
motifs were identified (**Figure 2d**) in which non-polar, aliphatic residues including alanine, valine and

7

isoleucine surround the acceptor succinylated lysines. The succinylated lysine site analysis indicated a

• Page 16, Line 375: fix spelling errors

We removed this sentence based on another reviewer's comments.

**Reviewer #2 (Remarks to the Author)**

The manuscript “Succinylation Links Metabolic Reductions to Amyloid and Tau
Pathology” compares succinylation in AD versus controls and surprisingly identifies
AB and tau as targets that succinylated exclusively in AD. As a potential rationale for
why these proteins may be succinylated in AD, the paper shows that mitochondrial
dysfunction in cells leads to escape of proteins from the mitochondria that may be
functioning in the succinylation of pathological AB and tau. In addition, the authors
perform a series of biochemical experiments which suggest a way that succinylation
may facilitate pathological AB and tau.

In the end, I think they can say that this paper demonstrates that a new protein
modification is found on APP and tau that may correlate with AD status. In addition,
some nice biochemical experiments raise a potential way in which this modification

can potentially influence the aggregation of these proteins. There is no evidence that it
is actually functionally doing so in vivo, nor is there any evidence that is correlates
with progression in humans, and the correlation in mice is the weakest part of the
paper. It remains possible that the modification is simply a consequence of
mitochondrial dysfunction in AD patients and the biochemistry not actually relevant
to what is functionally occurring in vivo- perhaps because the succinylation moiety is
rather large.

We think the large size of the succinylation suggests that it is biologically important.
We have shown that succinylation alters the activity of TCA cycle enzymes, APP
cleavage and tau function.

Nevertheless, even in this case, succinylation could serve as a marker, so it is still
potentially relevant. Also, there is sufficient data to warrant following up the work.
Thus, overall the findings are interesting.

However, the level of over-interpretation and over-blown claims are reckless and
unwarranted, so the text needs major revisions. There are also some experimental
concerns.

We have corrected all of the over-interpretation and over-blown claims.

**Major comments:**

The introduction is really short for an unknown topic and the Nature Communication
format. A lot more needs to be added for the reader to understand succinylation,
mitochondrial dysfunction, and AD pathology.

We redid the introduction according to reviewer's request **Lines 153-187** in the
manuscript.

**Introduction**

Misfolded deposits of the amyloid beta peptide (A β)^{1,2} and the microtubule-associated protein tau
(MAPT)³ are pivotal pathological features in Alzheimer's disease (AD), wherein reduced brain regional
glucose metabolism and synaptic density are correlated with the development of clinical cognitive
dysfunction⁴. Preclinical research studies show that reduced glucose metabolism exacerbates learning
and memory deficits concurrent with the accumulation of A β oligomers and plaques⁵, and misfolded
hyperphosphorylated tau^{6,7}. However, the interrelationship(s) linking these keys but apparently disparate
pathological processes remains unknown. While pro-amyloidogenic and/or immune-inflammatory genetic
factors have played prominent roles in advancing our understanding of AD, more recent formulations
have expanded the scope of molecular underpinnings of the disease^{8,9}. Sims and colleagues coined the
term "multiplex hypothesis of AD" to highlight the increasingly recognized shortcomings of the "amyloid
hypothesis of AD"⁹.

Post-translational modifications (PTMs) of proteins provide an efficient and rapid biological regulatory
mechanism that links metabolism to protein and cell functions. PTMs contribute to the functional
diversity of proteomes without the formation of new proteins or a change in their abundance by covalent
addition of functional groups that can alter protein charge, structure, and their interactions. Protein PTMs
play a central role in the pathology of neurological diseases. The function of tau can be altered via its
phosphorylation¹⁰, acetylation¹¹, methylation¹² and O-GlcNAcylation¹³. Protein succinylation of lysine
residues is a relatively novel PTM and changes the charge from positive to negative. The interactions of
lysine succinylation and acetylation play an important role in metabolic pathways¹⁴. However,
succinylation is poorly studied in the nervous system; our previous work demonstrated that lysine
succinylation functionally modifies enzymes of energy metabolism¹⁵.

There is an increasing interest in defining the precise metabolic pathways involved in the pathogenesis of
AD^{9,16-19}. A significant correlation between reduced brain regional glucose metabolism and decreased α -
ketoglutarate dehydrogenase complex (KGDHC)^{20,21} has been described in AD. Inhibition of KGDHC
activity leads to a wide-spread reduction in regional brain post-translational lysine succinylation, for
which the succinyl donor is presumably succinyl-CoA, both in yeast²² and cultured neurons^{15,23,24}. Studies
of organisms deficient in NAD⁺-dependent desuccinylase sirtuin 5 (SIRT5)²⁵ provide evidence of the
regulatory importance of succinylation in metabolic processes²⁶⁻³⁰. However, the role of succinylation in
metabolic pathways of the human nervous system or in neurodegenerative diseases is unknown. Our
report represents the first investigation of the human brain succinylome and its changes in AD. The
results suggest that succinylation may link AD-related metabolic deficits to structural, functional, and
pathological alterations involving APP and tau.

In Figure 3C, demonstrating that the change in succinylation is not just due to changes
in protein levels is a critical point. The correlation that is shown, though weak, is a bit
troubling. There is a new paper (Johnson et al, Nature Medicine 2020) focusing on
proteomics in neurodegenerative diseases. I think it would be important to compare
the changes in succinylated proteins to the changes in proteins presented in an
independent paper, such as this one to make this point more convincing.

**The goal of our proteomics dataset is to serve as benchmark demonstrating if in**
**very same 10 controls and 10 AD the abundance change of succinylated**
**peptides/sites found in AD is due to change of protein abundance or not.**

**We took 22 significantly changed succinylated proteins from Johnson's paper**
**and listed as below. Total protein levels vary by less than 12% (|Log₂(FC)|<0.12)**

in Johnson's paper, and we think this also support our view that changes in succinylation levels are not based on changes in protein levels.

Our paper					Johnson's paper			
UniProtKB ID	Gene name	Modifications	Succinylome (log ₂ FC)	Proteome (log ₂ FC)	Protein		Significance (Tukey P values)	Volcanoes log ₂ (FC)
					Gene Name	ID	Control-AD	Control-AD
P6995	HBA1	1xSuccinyl [K4]	1.127	0.441	HBA2	P69905	0.075	-0.119
		1xSuccinyl [K5]	0.978					
P68871	HBB	1xSuccinyl [K]	1.013	0.460	HBB	P68871	0.208	-0.087
		1xSuccinyl [K9]	0.933					
P338	ALDH4A1	1xSuccinyl [K19]	0.783	0.010	ALDH4A1	P30038	0.963	-0.007
Q8N465	D2HGDH	1xSuccinyl [K6]	0.670	-0.044	D2HGDH	Q8N465	0.011	0.298
P189	HSPD1	1xSuccinyl [K3]	0.519	-0.075	HSPD1	P10809	0.014	0.038
P49419	ALDH7A1	1xSuccinyl [K2]	0.496	0.043	ALDH7A1	P49419-2	0.306	0.038
Q9NVH6	TMLHE	1xSuccinyl [K10]	0.492	0.161	TMLHE	Q9NVH6-8	0.909	0.037
		1xSuccinyl [K8]	0.391					
Q9HR4	HDHD2	1xSuccinyl [K7]	-0.225	0.054	HDHD2	Q9H0R4-2	-	-
P24539	ATP5F1	1xSuccinyl [K8]	-0.315	-0.144	ALDH5A1	P51649	0.086	0.037
		1xSuccinyl [K11]	-0.469					
P3193	UQCRC1	2xCarbamidomethyl [C3;C11];1xSuccinyl [K5]	-0.323	-0.079	UQCRC1	P31930	0.000	0.093
P5165	PCCA	1xSuccinyl [K6]	-0.343	-0.002	PCCA	P05165	0.452	-0.028
P51649	ALDH5A1	1xAcetyl [K];1xSuccinyl [K]	-0.420	-0.090	ATP5F1	P24539	0.178	0.034
P8559	PDHA1	1xSuccinyl [K9]	0.439	-0.061	PDHA1	P08559	0.000	0.114
		1xSuccinyl [K1]	-0.582					
		1xOxidation [M1];1xSuccinyl [K4]	-0.691					
O75947	ATP5H	1xSuccinyl [K]	-0.490	-0.120	ATP5H	O75947	0.012	0.058
	ATP5H	1xSuccinyl [K2]	-0.516					
	ATP5H	1xAcetyl [K5];1xSuccinyl [K]	-1.334					
P5589	OXCT1	1xSuccinyl [K3]	-0.501	-0.069	OXCT1	P55809	0.056	0.038
		1xSuccinyl [K5]	-0.507					
P21926	CD9	1xSuccinyl [K7]	-0.503	-0.101	CD9	P21926	0.858	-0.031
P55	GOT2	1xAcetyl [K];1xCarbamidomethyl [C8];1xSuccinyl [K]	-0.627	-0.105	GOT2	P00505	0.996	-0.004
P621	PLP1	2xCarbamidomethyl [C15;C23];1xSuccinyl [K13]	-0.719	-0.152	PLP1	P60201	0.203	0.097
P56378	MP68	1xSuccinyl [K7]	-0.902	NA		P56378	-	-

Q9Y6M9	NDUFB9	1xSuccinyl [K3]	-1.026	-0.110	NDUFB9	Q9Y6M9	0.000	0.114
--------	--------	-----------------	--------	--------	--------	--------	-------	-------

We inserted the following into the text (Manuscript lines 258-264)

patients while 73 protein levels were increased (Extended Data Figure 2a). In a recent large-scale
 proteomic scan, the protein abundance of PDHA, PDHB, and DLD were all decreased in AD, which is
 consistent with our finding, representing a decreased abundance of proteins in impaired mitochondrial

8

states¹⁷. Thus, changes in protein levels and succinylation may be important in AD. Relatively small fold
 changes found between control and AD brain samples, were probably due to a well-known ratio
 compression caused by the co-isolation of isobaric-labeled background ions in MS2-based TMT
 quantitative proteomics.

Figure 4A, without a control for protein loading, the overall change in levels of succinylation are meaningless. This is somewhat mitigated by the B-actin control for the individual proteins in B. However, is the B-actin from 4B also being used to normalize 4C? The B-actin should be shown on the same blot in C. Also, there probably should be controls showing that the fractions have been sorted intact.

Figure 4A. The low cellular abundance of succinylation necessitates that we pull down the succinylated proteins with the anti-succinylation antibody. The IP only pulls down the succinylated proteins so we lose much of the beta actin which is not succinylated. No data suggests that the rotenone treatment will not affect the succinylation of beta-actin, so we think that beta-actin cannot be used as a loading control. In the Figure 4b we ran the two gels at the same time, so the loading samples were same, and we used the beta-actin from 4b to normalize 4c. We have repeated the whole assay and present the new results in the Figure 4c. See lines 65-68 in Figures and Tables.

c. The effects of rotenone (100 nM, 5 μM/20 min) on the distribution of PDHC protein between
mitochondria and non-mitochondrial fractions. The data from three different replicate experiments
were expressed as the mean with error bars from SEM (n = 3, **: $p < 0.01$, *: $p < 0.05$, two-way
ANOVA followed by Tukey's multiple comparisons test).

In Figure 4D, it looks like there is less colocalization in the Rotenone treatment. Are the images reversed? In either case, the resolution of the images is too low to comment on the localization. Also, why was only 100nm Rotenone shown? What about 5uM? Overall, I probably buy the interpretation, but the data could be cleaned up.

Figure 4d is correct. But we revised it and inserted magnified regions on the right.
Rotenone induces release of DLST into cytoplasm. In the control conditions, DLST
(magenta) was concentrated inside mitochondria defined by COX-IV labeling (green).
So, co-localization should maintain at the maximum level. After 1h of 100 nM
Rotenone treatment, additional DLST labeling was out of the mitochondria and spread
into the cytoplasm.

Lines 69-73 in Figures and tables legend

69 d. Rotenone induces release of DLST into cytoplasm. In the control conditions, DLST (magenta)
was concentrated inside mitochondria defined by COX-IV labeling (green). After 1h of 100 nM
Rotenone treatment, additional DLST labeling was found in the cytoplasm. Inserts on the right are
magnified regions. Magenta: DLST; Green: CoxIV; Error bars represent SEM deviation from the
mean (n = 98 fields from 19 dishes, ***: $p < 0.001$, Tukey's multiple comparisons test).

In Figure 5C and 5D, it would be nice to have an unaffected staining control to show that the change succinylation is specific.

We apologize, but we are not sure that we have understood the question. Wild type mice are unaffected. Succinylation is a part of normal brain metabolism in the same way as acetylation or phosphorylation. The succinyloomics data show that hundreds of proteins are succinylated in normals. These results are the first visualization in the

brain of AD and tau mice.

We expect to see succinylation not only in A β plaques or tangles, but also other
proteins. Indeed, the hypersuccinylation in AD brains was in APP and tau. We have
now discussed it in lines 442-446 in the manuscript

We show that transgenic mouse strains of either tauopathy or amyloidosis phenotype including plaques
exhibit widespread increases in lysine succinylation, which is not exclusive to tau and APP, but parallels
the formation of pathological species. While proteins in addition to tau or APP maybe altered, APP and
tau are only succinylated in brains from AD patients, which suggests that increased tau and APP
succinylation may play a role in the development of AD pathology. Thus, the modification of metabolism

More importantly, in WT there is a dramatic decrease in succinylation between 4
847 months and 10 months. Why is this? Could this be due to mitochondrial changes in
normal aging. This should probably be commented on in the discussion.

As discussed in detail above, the goal of this paper is to look at the association of
succinylation to tau and APP. The effects of aging and the maturation of the pathology
are beyond this paper and the ten-month data has been omitted.

However, for the review we offer the following speculative answer. This is a hard
question to resolve because this is the first study investigating succinylation in the
brain of APP or tau mice. Succinylation is a post-translational modification and
several factors can regulate the balance between succinylation and desuccinylation.
Our findings show that α -ketoglutarate dehydrogenase (KGDHC) is a major
succinylase in neurons. We have shown that brain KGDHC is not altered with age out
to 30 months (Freeman, Nielsen et al. 1987) suggesting the change is not a reduction
in succinylation. Whether aging may alter KGDHC migration to the cytosol has never
been studied. The desuccinylases in brain remain unknown. A prominent paper by two
of our co-authors have shown that sirtuin 5 (SIRT5) plays a central role in modulating
heart metabolism and function (Sadhukhan, Liu et al. 2016).

SIRT5 is localized in the mitochondria and shows a weak deacetylase activity but
a potent desuccinylase activity on lysine residues both *in vitro* and *in vivo* (Park, J
2013 Mol. Cell; Du et al., 2011; Peng et al., 2011). The catalytic reaction involves the
removal of a succinyl group from the lysine side chain of protein substrates, a process
that consumes NAD⁺ as a co-substrate and generates nicotinamide (NAM) and 2'-O-
succinyl-ADP-ribose (Rardin MJ 2013 Cell Metabol). SIRT5 KO mouse embryonic
fibroblasts display an increase in lysine succinylation but not acetylation (Du et al.
2011 Science). We have used SIRT5 to desuccinylate enzymes such as the pyruvate
dehydrogenase complex. We have also looked at succinylation in SIRT5 KO mice,
which show a significant increase in succinylation levels in the liver while trivial
changes were found in the brain. Liver succinylation, but not that in brain responds to
fasting. Furthermore, the data from the AD samples suggest that different
desuccinylases are likely important in the cytosol and mitochondria. We believe that
our current results justify further studies on the regulation of succinylation in the brain
during aging and in neurodegenerative diseases.

However, no studies reveal the relationship between aging and succinylation.
Since aging is one of the most important risk factors for the development of AD, we
think a separate deeper research need to investigate the age-dependent decrease in
succinylation in future studies. We believe that 4-month-old mice clearly show an
association between increased succinylation and disease progression (pathology).

The large increase in succinylation in 4-month-old Tg19959 mice agrees with our
hypothesis: abnormal mitochondrial function promotes the release of KGDHC, which
in turn, increases succinylation.

In the Tg mouse, there a lot more succinylation. However, there is also a decrease in
succinylation from 4 to 10 months that is similar to WT, despite the fact that AB is
definitely increasing in the Tg mice over this time period. Thus the two do not seem to
be particularly well correlated. Also, is the level of succinylation increase in the Tg
mice prior to appearance of AB? This should be checked because they already observe
a dramatic increase in succinylation at 4 months when AB is first forming. Thus, it is
possible that succinylation is changing in the Tg model well before this. Overall,
though it is clear that succinylation is responding to the Tg, it is not at all clear that it
correlates with the build up of AB. The exact same thing is true for tau in Figure 6.

The results clearly demonstrate that succinylation is clearly responding to the
transgenic in both mouse models. This is consistent and suggest that increased
succinylation at early stages of pathology development may be involved in the
succinylation of multiple proteins. Howearly this occurs is of great interest to us but
beyond the scope of this manuscript.

We used a pan succinylation antibody. We know a few hundred proteins are
succinylated with multiple largely unknown functions. Our data suggest that
succinylation could be involved in early stages of plaque or tangle formation to
promote overproduction of A β and inability of tau to bind to microtubules.
We agree with the reviewer that it would be great to know the role of succinylation
from development to old age, as well as well as pathology. Interestingly, the activity
of KGDHC peaks at about 30 days post-natal, when the cortex reaches final
maturation. We hope that this paper will encourage examination of succinylation
under all these conditions, but we feel it is beyond the scope of this paper.

The fact that the same phenomenon is true in both AB and Tau models is perhaps even
more disconcerting, because it suggests that the phenomenon is not specific. That is to
say, the overall increase in succinylation seems to occur regardless of the pathological
insult, which does not cause mitochondrial dysfunction exactly the same way in the
two models, and the timing of that pathological insult, which is not exactly the same
in the two mouse models. This should at least be discussed. In particular, it would be
nice to know how the changes in succinylation relate to the changes in mitochondrial
dysfunction in the two Tg models in their hands. Overall, the data supports a change
in succinylation that occurs in the two pathological models, but does not support the
conclusion that the change in succinylation truly correlates with pathology. At a

minimum, this should be clearly noted in the results and discussion.

The reviewer clearly states an important issue that speaks to the essence of the
whole paper. Since the extensive data required to answer is not available, we have
now discussed it in the revised version, as suggested by the Reviewer. As indicated by
the reviewer, the data shows and increase in succinylation in two models of pathology.

One question is whether precise changes in mitochondria are required to alter
succinylation of tau and APP. The precise changes in mitochondria from AD patients
or in animal models of pathology are unknown, so it hard to exactly model the precise
mitochondrial changes. The data on relation of the mitochondrial changes to
succinylation in brain is limited to our published studies, which show that
mitochondrial succinylation decreases rapidly in response to many altered metabolic
states and increases in response to others. In addition, the relation to cytosolic
succinylation has never been explored. The results in this manuscript suggest new
experiments to test the relation of select metabolic insults to the release of KGDHC
from the mitochondria and succinylation of specific cytosolic proteins.

The fascinating question of specificity may not be only related to the kind of
inhibition but to the severity and duration of the inhibition. The current studies
provide justification for exploring cytosolic succinylation in response to a variety of
insults. Future experiments could also determine which metabolic insults lead to
release of KGDHC and the succinylation of specific cytosolic proteins. The coupling
between succinylation due to unique modifications of the mitochondria or the severity
of the sick mitochondria must be different in diverse regions of the brain or maybe,
the sensitivity is different.

We have modified the discussion by adding the following paragraph Please see Lines
452-459 in the manuscript.

Lysines are highly modified residues. Understanding the relationship of succinylation to the other PTM is
critical to a complete understanding of its role in AD pathology. A direct comparison is practically
difficult because each PTM requires a different enrichment strategy. Lysine311 has been associated with
ubiquitination, dimethylation, and acetylation in transgenic mice⁵⁹, but not human brains. Our
experiments raise the question of whether precise changes in mitochondria are required to alter
modification of specific proteins. Succinylation appears directly linked to KGDHC and mitochondria.
Whether other post-translational modifications are also linked to mitochondrial dysfunction remains to be
determined.

460

There is a slight concern that the peptide used for the tubulin polymerization assay is
succinylated throughout, when they only detected succinylation at K311 in AD.

Nevertheless, the loss of Tau polymerization function is impressive. This should be
more clearly stated and used to qualify the interpretation. This is particularly true
because the succinylation moiety is quite large. The authors should definitely discuss
how such a large modification could affect proteins and the biochemical assays that
they perform on them. The decrease in Tau-tubulin interactions is more convincing,
particularly because they also performed this assay with Tau only succinylated at

K311.

Performing the site-specific succinylation of the lysine 311 in vitro is very
difficult, in the present case of the K19 peptide has more than one lysine. It is
also quite difficult to synthesize a 99-aa length peptide by using peptide
synthesis techniques. Finally, we decided that we first use a pan-succinylation
on K19 peptide and tested its ability on tubulin polymerization assay. Then
we tested if the succinylation of K311 is sufficient to specifically decrease
tau-tubulin interactions by 1H saturation transfer difference (STD) NMR.
We added the following comment to the discussion in the manuscript **lines**

425-428

dysfunction⁵⁴⁻⁵⁷, thereby contributing to progression of amyloidosis and tauopathy. It is not perhaps
surprising that succinylation would have such an effect since it increases the size of the lysine side chain
considerable and could lead to steric clashes, as well as reversing the charge of the side chain (Extended
Data Fig. 4j).

Overall, it is possible that succinylation is simply a consequence of mitochondrial
dysfunction, and not necessarily functional in AD. To mitigate this, I think it might be
good to provide some additional negative controls if possible. ie do AB and tau
accumulate any other post-translational modifications that might just be due to
disruption of mitochondria or the abnormal appearance of the pathological versions of
these proteins in the cytosol.

As indicated, the data suggests that the succinylation is a consequence of
mitochondrial dysfunction. We show succinylation is altered in AD at very specific
sites. We show that succinylation causes pathologically important changes in tau and
APP. How the succinylation interacts with other modification is critical but beyond
this study. Our studies show that the link of mitochondria and acetylation is very
different than the link mitochondria and acetylation. We added the following to the

manuscript (Lines 455-459)

ubiquitination, dimethylation, and acetylation in transgenic mice⁵⁹, but not human brains. Our
experiments raise the question of whether precise changes in mitochondria are required to alter
modification of specific proteins. Succinylation appears directly linked to KGDHC and mitochondria.
Whether other post-translational modifications are also linked to mitochondrial dysfunction remains to be
determined.

460

There is a lot of wild speculation:

Example: The last sentence of the abstract is wildly overexaggerated- While there is a
possibility that succinylation could contribute to pathologies in AD, the data presented

in the manuscript certainly are not by themselves enough to even raise the possibility
that succinylation must be addressed therapeutically for meaningful clinical benefit

Example: The last sentence of the introduction is overstated and unnecessary

We have completely re-written the abstract (manuscript lines 118-129) and
introduction (manuscript lines 153-187). We avoided wild speculations.

Example: line 311- this reflected a potential existence of succinylation-
phosphorylation switch as in the case of acetylation- the paper provides absolutely no
evidence for this. This could be speculated about in the discussion, but is completely
inappropriate for the results section.

We modified the manuscript in two places

Lines 423-425

423 altered cerebral metabolic function in AD as the disease progresses. Other PTMs, such as ubiquitination,
424 acetylation and phosphorylation, have been recently shown to affect amyloid degradation^{54,55} and tau
425 dysfunction^{54,57}, thereby contributing to progression of amyloidosis and tauopathy. It is not perhaps

Lines 461-466

Overall, our data represent the first report of the human brain succinylome and its implications for
mitochondrial function as well as for the molecular pathogenesis of amyloidosis and tauopathy, two of the
cardinal features of AD. We provide a rich resource for functional analyses of lysine succinylation, and
facilitate the dissection of metabolic networks in AD. The current studies also lay the foundation for
future investigation into the crosstalk between different PTMs, including acetylation, phosphorylation,
ubiquitination and succinylation associated with AD and tau pathology. The discovery that succinylation

Example: line 283- taken together the accumulated data strongly suggest that
succinylation of K678 might lead to an early-onset enhanced generation,
oligomerization and plaque biogenesis, consistent with the effects of known genetic
disease mutations at this site. While the data suggest a potential was for succinylation
to affect Aβ cleavage, there is no functional evidence that it does so.

We have modified this. Please see Line 347-350 in the manuscript.

at t = 24 or 48 hrs (Figure 5g). These data demonstrate that succinylation of K612 of APP is a key
molecular event that promotes Aβ oligomerization. Taken together, our findings suggest that
succinylation of K678 might lead to early-onset and/or enhanced generation, oligomerization and plaque
deposition of Aβ, consistent with the effects of known genetic disease mutations at this site^{37,42}.

Example line 369: Notably, these results demonstrate for the first time that succinylation is the key link between the signature metabolic reductions and amyloid

plaques and neurofibrillary tangles in AD.- Again, although this is possible, there is
absolutely no evidence for this in the manuscript

We have modified this. Please see Line 419-425 in the manuscript.

Our study provides a system level view of the human brain succinylome in metabolic process, particularly
in mitochondria, and reveals a dramatic alteration on succinylation in AD. Our results demonstrate for the
first time that succinylation is a key link between the signature metabolic reductions and A β plaques as
well as NFTs in AD. We show that changes in protein succinylation, as a molecular signal, correlate with
altered cerebral metabolic function in AD as the disease progresses. Other PTMs, such as ubiquitination,
acetylation and phosphorylation, have been recently shown to affect amyloid degradation^{54,55} and tau
dysfunction^{54,57}, thereby contributing to progression of amyloidosis and tauopathy. It is not perhaps

Example line 371: The current results reveal that varied in protein succinylation, as a
molecular signal, correlates with altered cerebral metabolic function in AD as the
disease progresses.- While there may be a small amount of evidence of this in the
mice data (if you compare to previous analysis of the mice strains employed and you
ignore the fact that the mouse data don't truly correlate), they do not show this on
their own and they certainly did not examine succinylation across the progression of
the disease in human cases, so this statement is not warranted.

This phrase has been deleted in the red-write of the section.

Minor comments:

I really don't understand the math in figure 1, and there is no description of what the
29 proteins in B are? Are they differential between AD vs control, as in D? And what
percentages are up vs down? An effort should be made to make the numbers more
clear.

We provide Fig.1 as a schematic workflow for a better understanding of the
data analysis. Since we did independent cohorts, we would like to explain that
we have high reproducibility in two cohorts (both succinylome and proteome).
29 proteins in B are those succinylated proteins having the sites that have $\log_2 |\text{fold change}| > 0.25$
and p value < 0.05 found in succinylome analysis of human brain. Yes, they are differential between
AD vs control. Out of 29 succinylated proteins, the abundance of 12 succinylated sites of the
proteins is increased while the abundance of the rest 17 sites is decreased. The fold change of
the succinylome and proteome were provided in Supplementary Table 4 and
6, also they have been visualized in Fig 3a and Extended Data Figure 2.

The antibody used in extended data figure 3B needs to be clearly shown in the figure.

The antibody names are on the figure, so we are not sure of the question. We did
improve the resolution.

All antibodies are in the methods section.

In extended figure 3F and G, why is the full length protein so different when the
production of the cleavage products remains the same? Perhaps I don't quite
understand the assay, but this should be clarified.

We performed both A β 6-29 and cleavage fragments measurement. The cleavage
products along with remaining A β 6-29 were quantitatively determined by LC-MS/MS
analysis. Specifically, an MRM (multiple reaction monitoring) method was used in
the mass spectrometric assay to measure the amount of normal and succinylated
peptides of A β 6-29 changes with time after incubated with rhADAM10. The extended
Fig 3g and 3h was changed into a new and clear Fig 3g where the changes with time
was shown only for the peptides, not for the fragments produced per reviewer's
suggestion.

The label of the graph in 5F needs to be clarified as percentage change from 0 hrs.

During the 0 hrs, the intensity will be 0, and it can't be the denominator.

Line 309 refers to succinylated tau. However, it should refer to succinylation in the
Tau Tg mice. There is a big difference.

This phrase has been deleted.

The Y-axis in Figure 4B says evel instead of level

We corrected the figure. We also modified the legend to Figure 4b in Figures and
Tables line 61-64.

61 b. The effects of rotenone (100 nM, 5 μ M/20 min) on the distribution of KGDHC protein between
62 mitochondria and non-mitochondrial fractions. The data from three different replicate experiments
63 were expressed as the mean with error bars from SEM (n = 3, ****: $p < 0.0001$, **: $p < 0.01$, *: p
64 < 0.05, two-way ANOVA followed by Tukey's multiple comparisons test).

The Tg19959 and P301S mouse models need to be defined for the reader.

We added more detail to the methods section lines 498-506 The references were also
included in the main text.

All the experiments were carried out in four and ten-month-old transgenic mouse models of AD.
Tg19959 mice (that overexpress a double mutant form of the human amyloid precursor protein) were
obtained from Dr. George Carlson (McLaughlin Research Institute, Great Falls, MT, USA). Tg19959
mice were constructed by injecting FVB X 129S6 F1 embryos with a cosmid insert containing
human APP₆₉₅ with 2 familial AD mutations (KM670/671NL and V717F), under the control of the
hamster PrP promoter.²⁵ P301S (PS19, that overexpress the human tau gene harboring the P301S

12

mutation) transgenic mice were purchased from The Jackson Laboratory (Bar Harbor, ME, USA). The
transgenic mice used in this study express the human pathogenic mutation P301L of tau together with the
longest human brain tau isoform (htau40) under control of the neuron-specific mThy1.2 promoter.²⁶

Line 194 should say from rather than form

We have modified Lines 235-238 of the manuscript.

Completion of the human brain succinylome and global proteome analyses allowed direct comparison
between brains from controls and AD patients. Without enrichment of succinylated peptide in global
proteome data, the number of succinylated peptides identified is 0.13% total peptides for cohort 1 and
0.28% for cohort 2. The notable difference in ratio of succinylated peptides over total peptides between

PDHA1 is mentioned with no context

We modified the text lines 205-208

205 73% (229/314) of the succinylated proteins were mitochondrial (**Figure 2b**). The pyruvate dehydrogenase
206 complex E1 component subunit alpha (PDHA1), which links glycolysis to the TCA cycle, was
207 significantly succinylated. The eight enzymes of the TCA cycle located within the mitochondrial matrix
208 and their multiple subunits were also extensively succinylated. Furthermore, succinylation of proteins was

Line 257: paralleled should be parallel

We have modified the text on lines 306-310

which carries the human APP with two mutations (Figure 5c and Extended Data Figure 3a). Double
immunofluorescence staining with antibodies to pan-lysine-succinylation and to A β oligomers (NU-4)³⁵
or to A β plaque (β -Amyloid, D3D2N) revealed an early increase in lysine succinylation that appeared to
parallel oligomer formation and subsequent plaque deposition in the hippocampus. These findings suggest
that the APP succinylation might be involved in A β oligomerization and plaque formation *in vivo*.

The logic in line 266-267 needs to be better spelled out for the reader with regard to
the competition of the two enzymes. Likewise in line 271, ADAM10 needs to be
introduced as a secretase for the reader.

A: We have modified Lines 314-330 in the manuscript

The generation of the A β is a highly regulated process by the secretases. β -secretase initiates the
amyloidogenic pathway, while α -secretase is part of the non-amyloidogenic pathway bisection the A β
domain and thereby inhibiting the formation of A β . In subsequent experiments, we tested the relationship
between succinylation and APP processing by the secretase enzymes. K612-L613 is the APP α -secretase
bond, and missense mutation at K612N produces early onset AD³⁷. Furthermore, global proteomics
showed an increase in β -secretase (BACE1) abundance of 31% in AD brains compared to controls
(Supplementary Data Table 6), while no changes occurred for either α -secretase or the sirtuins (SIRT)
family (Extended Data Figure 2c). Seyfried et al., quantified a total of 2,745 proteins in two regions
(dorsolateral prefrontal cortex (FC, Brodmann Area 9) and precuneus (PC, Brodmann Area 7)
were quantified. The number that overlapped was about 2,332 proteins (85.3%) compared with
our data (4442 proteins from 10 controls and 10 AD, Brodmann area 44/45).

The four ADAM family members identified in that paper were also identified in our proteome.
The protein level of ADAM 10, 22, 23 did not change in that paper nor our data, while ADAM11
showed a similar decrease in the two cases. SIRT2 and SIRT5 levels did not vary³⁸. Further,
protein levels of SIRTs do not necessarily reflect activity, which are often regulated by substrates
and post-translational modifications.

The S and C labels in 5C need to be defined in the main figure.

We presume the reviewer means the label in Figure 5e. We have modified the text in
Figures and Legends lines 100-108

e. Succinylation blocks α -cleavage. Peptides were incubated for 24 h with or without rhADAM10.

Peak area ratio values were calculated and are shown relative to corresponding controls without

rhADAM10. Each sample was run in triplicate and data were expressed as the mean with SEM

(**: $p < 0.01$, two-way ANOVA followed by Bonferroni's multiple comparisons test; except for

one sample from the group of succinylated peptide without rhADAM10 was damaged).

f. Western blot analysis of succinylated and control A β ₄₂ from aggregation assay showed that the

succinylation generates more oligomerized A β even after a long incubation. The data from two

different replicate experiments were expressed as the mean with error bars from SEM (****: $p <$

0.0001, **: $p < 0.01$, two-way ANOVA followed by Bonferroni's multiple comparisons test).

**Line 302: abeta should be absence?**

We modified the manuscript **Lines 368-370**

To characterize tau succinylation in a transgenic mouse model of tangle formation, we used
immunofluorescence staining to compare the presence of lysine succinylation within tau oligomers (T-22)
⁴⁶ and phospho-tau (AT8) in the hippocampus from wild type and TgP301S mice. No phosphorylated tau

**Line 371: varied should be variation**

We modified the manuscript lines **Lines 419-422**

Our study provides a system level view of the human brain succinylome in metabolic process, particularly
in mitochondria, and reveals a dramatic alteration on succinylation in AD. Our results demonstrate for the
first time that succinylation is a key link between the signature metabolic reductions and A β plaques as
well as NFTs in AD. We show that changes in protein succinylation, as a molecular signal, correlate with

**Line 375: involvon should be involved in?**

We removed this sentence based on another reviewer's comments.

**Line 394: bot should be both?**

We have modified the text lines **430-432**.

The mechanisms and control of both non-enzymatic and enzymatic succinylation by cellular
succinyltransferases and desuccinylases are unknown. Our data clearly demonstrate that impairing
mitochondrial function decreases mitochondrial succinylation and promotes succinylation of specific non-

**Line 385: The decline in succinylation of mitochondrial proteins suggests that**
**activation of descuccinylases- The alternative, that there could be a failure to maintain**
**succinylation levels, should be mentioned.**

We modified **the text lines 427-437**

The mechanisms and control of both non-enzymatic and enzymatic succinylation by cellular
succinyltransferases and desuccinylases are unknown. Our data clearly demonstrate that impairing
mitochondrial function decreases mitochondrial succinylation and promotes succinylation of specific non-
mitochondrial proteins by altering the distribution of succinyltransferases from the mitochondria to
cytosol. Precedent for this concept is provided by results showing that translocation of the DLST subunit
of KGDHC to the nucleus increases histone succinylation³⁵. Rotenone induces translocation of PDHC

13

from mitochondria to other cellular compartments³⁸. The decline in succinylation of mitochondrial
proteins, appears due to a failure in maintaining succinylation levels, and may suggest that activation of
desuccinylases (e.g., Sirtuins) or general increases in NAD⁺ should be reconsidered. The large increase
in succinylation in 4-month-old Tg19959 mice agrees with our hypothesis, in which abnormal
mitochondrial function in AD promotes the release of KGDHC and subsequent increases succinylation.

**The manuscript should also be edited for grammar.**

Examination of the marked copy shows that we carefully changed the manuscript
including changes in grammar.

**Reviewer #3 (Remarks to the Author)**

The authors have investigated the potential role of succinylation and Amyloid and Tau
pathology using brain tissue from AD cases and controls. They analysed brain tissue
cell lysate proteomes using 10 plex TMT. They also analyzed the same 10 controls
and 10 AD patients' brain samples Succinylome using Cell Signaling Tech IP-MS kit
and ran LCMS of the PTM enriched tryptic peptides.

**Comments:**

**1- In addition to bioinformatics analysis of succinylome IP-MS data it would be**
**useful to analyze and show the biological significance of whole tissue lysate 10- plex**
**TMT data as well. It would be useful to cover the global proteome analysis done**
**which may be relevant to disease pathogenesis in addition to the succinylome targeted**
**concept.**

**We agree that the whole proteome analysis relevant to disease progression is**
**important to find out some potential biomarkers in AD. It would be too much to add**
**to this paper since the primary focus of this manuscript is the succinylation. We had**
**added the global proteome data to highlight the changes in succinylation**
**peptides/sites of the protein are unrelated to any changes in protein abundance in AD**

compared with healthy samples.
We have added references to recent papers that report the complete proteome.
The whole proteome is posted online if the reviewer wants to see but it would change
the focus from this paper. We have added several mentions of the Johnson et al paper
and the focus of that paper is proteomics.

2- It is important to pinpoint sites of protein succinylation. Succinylome localization
shown in Figure 1a is not clear.

In Fig 1a, we only present a schematic workflow for succinylome studies where it is
shown that succinylation on lysine residues were immune- enriched prior to nano
scale LC-MS/MS analysis.

3- The authors considered impaired mitochondrial function resulted in succinylome
localization to be pushed out of mitochondria to cytosol by leakage (Figure 4, Line
244-247), however, whole tissue lysate mass spec succinylome data suggested an
overall decrease in AD. These findings need to be reconciled

Please remember the majority of succinylated proteins in AD brains are in the
mitochondria. Mitochondrial damage results in impaired enzyme activity at the total
succinylation level. So, our succinylome supports this idea. The leakage of
mitochondrial enzymes, this will result in unexpected modifications on some proteins
(APP or Tau) that have no or less access to these succinyl transferases.

4- The K687 site in the middle of APP is the interaction site of α -secretase and
the cleavage was inhibited when the K was succinylated in vitro (Figure 5e).

However, it cannot be assumed that succinylated A β has more aggregation property
since the comparison shown in Figure 5f did not show a significant difference. The
AD patient succinylome mass spec identified succinylated K687 peptide (S Table 5).
It is not clear whether the succinylome IP conditions favored solubilizing Abeta
plaque and protofibril oligomers.

We think Figure 5f shows a significant difference. Compared with the intensity, we
found succinylation generates more oligomerized A β . During the MS protocol, we
broke the protein complex molecule into peptides, and then we did the IP. So, during
the IP processing, there is no big complex (no plaque nor tangle), so it is unlikely
there will be such a preference. The anti-Succ-K antibody recognizes the exposed
succinylated lysine. As described in the methods, all of the exposed succinyl-lysines is
expected to be effectively pull down by the antibody in our enrichment steps.

However, if this antibody pull down would favor certain sequence dependent species
of AB or tau is unknown. Since we really compare the same succinylated

peptides/sites among all samples, the possible bias introduced by variable peptide
sequence would not affect our quantitative results between AD and health cases.

**5- APOE4 mutation is a risk factor of AD. Does the proteomics data reveal mutation**
**in the 20 patient brains analyzed?**

No, the global proteomic data cannot reveal the mutation sites of APOE4. In shotgun-
based mass spectrometry analysis, we were only able to identify proteins by a series
of tryptic peptides. Normally, these detected tryptic peptides cannot cover the whole
protein sequence. When the software searches acquired spectra against a database
containing a particular protein, it can only match the exact same sequences of the
tryptic peptide of the native protein in the database. If mutations occur at certain
sites, the database searching software cannot recognize these peptides unless the
mutated protein sequence was added to the database prior to database search. We have
verified the APP data. The missing succinylation of APP in only one case was not due
to a mutation.

Freeman, G. B., P. E. Nielsen and G. E. Gibson (1987). "Effect of age on behavioral and enzymatic
changes during thiamin deficiency." Neurobiology of Aging **8**(5): 429-434.

Kaden, D., A. Harmeier, C. Weise, L. M. Munter, V. Althoff, B. R. Rost, P. W. Hildebrand, D. Schmitz,
1240 M. Schaefer, R. Lurz, S. Skodda, R. Yamamoto, S. Arlt, U. Finckh and G. Multhaup (2012). "Novel
APP/A β mutation K16N produces highly toxic heteromeric A β oligomers." EMBO Molecular Medicine
**4**(7): 647-659.

Morris, M., G. M. Knudsen, S. Maeda, J. C. Trinidad, A. Ioanoviciu, A. L. Burlingame and L. Mucke
(2015). "Tau post-translational modifications in wild-type and human amyloid precursor protein
transgenic mice." Nature Neuroscience **18**(8): 1183-1189.

Sadhukhan, S., X. Liu, D. Ryu, O. D. Nelson, J. A. Stupinski, Z. Li, W. Chen, S. Zhang, R. S. Weiss, J.
1247 W. Locasale, J. Auwerx and H. Lin (2016). "Metabolomics-assisted proteomics identifies succinylation
and SIRT5 as important regulators of cardiac function." Proceedings of the National Academy of
Sciences **113**(16): 4320.

Seyfried, N. T., E. B. Dammer, V. Swarup, D. Nandakumar, D. M. Duong, L. Yin, Q. Deng, T. Nguyen,
C. M. Hales, T. Wingo, J. Glass, M. Gearing, M. Thambisetty, J. C. Troncoso, D. H. Geschwind, J. J. Lah
and A. I. Levey (2017). "A Multi-network Approach Identifies Protein-Specific Co-expression in
Asymptomatic and Symptomatic Alzheimer's Disease." Cell Systems **4**(1): 60-72.e64.

Tracy, T., K. C. Claiborn and L. Gan (2019). Regulation of Tau Homeostasis and Toxicity by Acetylation.
Tau Biology. A. Takashima, B. Wolozin and L. Buee. Singapore, Springer Singapore: 47-55.

REVIEWER COMMENTS

Reviewer #1 (Remarks to the Author):

The authors have done an excellent job addressing the previous critiques.

Reviewer #2 (Remarks to the Author):

Overall, the manuscript is improved. However, the following points still need to be addressed:

In 5C, the authors need to perform dual IF with another protein that is unchanged in the same image that they are staining for succinylation, so that we can be sure that the changes are not just due the one sample staining a bit better, and therefore being brighter. The other Ab that is used currently is for AB oligomers is nice for co-localization, but is uninformative for comparing levels of succinylation. This is standard control when trying to quantify changes in levels from staining.

I think that removing the 10 month time point from figure 5 is not appropriate, as it risks just hiding data that potentially does not fit their model. I think the authors need to put the 10 month time point back and add a potential explanation. Personally, I don't find the argument that they provide in the rebuttal to be that compelling, but that is for the reader to judge. Removing the data completely does not provide the reader the opportunity to do so, and is therefore inappropriate.

Along these lines, the authors response to the critique that the changes in succinylation in the transgenic mouse model don't correlate very well with the build up of AB is insufficient. While it is clearly true that succinylation is responding to the transgene, the fact that succinylation doesn't correlate well with the build up of AB provides a major caveat to the interpretation. If the authors do not wish to perform additional experiments to address why this might not be the case, they need to at least acknowledge the caveat and discuss it. As is, the authors seem to be saying that because the transgene causes a change in succinylation, it doesn't matter that that change does not seem to correlate well with the build up of pathological AB. When actually, this disconnect raises the clear possibility that AB build up and succinylation are not directly related. This doesn't mean that their results are not potentially interesting or important, but it does need to be acknowledged and discussed so that the reader is not left with an incorrect view of the data.

Similarly, the discussion that has been added to supposedly address the review point about the changes occurring in models of both AB and Tau is also not sufficient. The authors need to specifically discuss why succinylation may be changing in both AB and Tau models, despite the models being different. The new discussion that was added does not address this directly at all, and so only serves to further muddy the interpretation.

The above points are especially important, considering the revised abstract that states "Our results reveal a tight relationship linking lysine succinylation status and AD-associated proteinopathies" As discussed above, this relationship is not necessarily tight, so this seems inappropriate. It is fine for the authors to state that they find a relationship in both human cases and mouse models. This certainly raises the possibility that there is a mechanistic relationship, which is interesting. However, there are major caveats to this relationship, and the actual mechanistic relationship is certainly unclear from their data. So, I don't understand why the authors keep insisting on such language.

I think the authors need to add a sentence to the results clearly stating that the peptide used for the tubulin polymerization assay is succinylated throughout, so that it is very clear for the reader.

The authors also continue to have language that is too strong for their data. For example, line 347 should say something like "may be a key molecular event that contributes" rather than "is a key molecular event that promotes." In line 421 it could say something like "our results raise the possibility that succinylation may provide a link" rather than our results "demonstrate that succinylation is a key

link." The data are compelling. There is no need to over sell it.

Reviewer #3 (Remarks to the Author):

The authors investigated lysine succinylation changes in the brain associated with Alzheimer's disease comparing lysine succinylomes and proteomes from AD and control brains. They found in AD brains, succinylation declined for multiple mitochondrial proteins, and increased for a smaller number of cytosolic proteins, among which the amyloid precursor (APP) and tau exhibited the largest increases. In transgenic mice models of AD, they also found elevated succinylation of soluble and insoluble APP and tau. They examined the effect of succinylation and observed disrupted normal secretase processing of APP and A β accumulation. Succinylation of tau also promoted its aggregation and impaired microtubule assembly.

PTMs have been investigated in connection with neurodegenerative diseases. Tau proteins are known to undergo PTMs While succinylation is a well known modification that causes significant changes in proteins the studies presented by the authors are novel and expand our knowledge of the pathology in AD.

The authors have largely addressed prior reviewer concerns related to mass spectrometry. A concern about the prior submission was the exclusive focus on succinylation without comparison to other PTMS. The authors response seems to acknowledge this issue by expanding the introduction to cover other PTMs in AD as background but no additional data is provided. The response of the authors to other issues such as independent validation of data being beyond the scope of the paper is understandable.

Overall the quality of the revised paper is much improved and the findings are of interest to the field

We appreciate the insightful consideration and valuable comments of reviewer #2. We have addressed each of the constructive comments on an item-by-item basis, and provide responses addressing each of them (we insert the line numbers of the marked documents). The changes are highlighted in yellow in the marked manuscript and methods section and our answers are inserted in blue immediately after each reviewer's question.

REVIEWER COMMENTS

Reviewer #1 (Remarks to the Author):

The authors have done an excellent job addressing the previous critiques.

Reviewer #2 (Remarks to the Author):

Overall, the manuscript is improved. However, the following points still need to be addressed:

In 5C, the authors need to perform dual IF with another protein that is unchanged in the same image that they are staining for succinylation, so that we can be sure that the changes are not just due the one sample staining a bit better, and therefore being brighter. The other Ab that is used currently is for AB oligomers is nice for co-localization, but is uninformative for comparing levels of succinylation. This is standard control when trying to quantify changes in levels from staining.

Methods lines 411-415.

We thank you the reviewer for raising this point. All sections were stained at the same time under the same conditions (solutions, washing, temperature, etc.) and analyzed under identical experimental settings. Our results are expressed as the mean with SEM representative of the average of ~900-1000 pyramidal neurons comprised in 3-4 different brain sections per animal (n = 4 mice per each group).

Manuscript lines 379-381.

Similar to what we observed found in A β deposits, 10-month-old wild-type and transgenic tau mice displayed a significant reduction in the levels of succinyl-lysine in comparison to 4-month-old mice, thereby leading to an attenuated colocalization between succinylation epitopes and tauopathy epitopes.

Manuscript lines 468-490.

While proteins in addition to tau or APP are succinylated, APP and tau succinylation status increase in brains from AD patients, which suggests

that increased tau and APP succinylation may play a role in the development of AD pathology. Intriguingly, lysine succinylation levels decrease in 10-month-old mice over 4-month mice, while both amyloid aggregation and tauopathy continued to increase. This may reflect either de-succinylation processes, or sequestration of succinylated sites away from labeling antibodies. Notably, both K16 in A β and K311 in tau are buried in the structured core of their respective aggregated forms(Goedert *et al.* 2021),(Zhang *et al.* 2019). The decrease in the association between succinylation and pathology at 10 months may be due to results at least in part incorporation of succinylated sites inside aggregated species, preventing detection by immunohistochemistry. However, based on the current data, it is not possible to rule out alternative explanations, including potential changes in metabolism leading to de-succinylation reactions that may be related, or unrelated, to the progression of pathology and disease. Importantly, some precedent is provided by reports in which tau acetylation at residue K280 also peaks and decays during the course of tangle formation and cell death, leading to the suggestion that this epitope is either masked in paired helical filaments (PHFs) or else is subjected to deacetylation in later stages of aggregate maturation(Irwin *et al.* 2012). An adequate explanation requires a complete accounting of which proteins are involved (i.e., a complete mouse brain succinylome at multiple ages) and knowing which proteins are incorporated into deposits in both humans and mouse models. The mitochondrial succinylome in human brain tissues was significantly reduced in AD while succinylation of APP and tau was increased. Despite these remaining questions, our results suggest that the modification of metabolism in disease may lead to critical succinyl-mediated modifications of extramitochondrial proteins including APP and tau leading to aggregation and deposition. Preventing APP and tau succinylation and/or increasing mitochondrial succinylation may provide novel therapeutic targets for the prevention and/or treatment of AD and associated pathologies(Yang & Gibson 2019).

I think that removing the 10 month time point from figure 5 is not appropriate, as it risks just hiding data that potentially does not fit their model. I think the authors need to put the 10 month time point back and add a potential explanation. Personally, I don't find the argument that they provide in the rebuttal to be that compelling, but that is for the reader to judge. Removing the data completely does not provide the reader the opportunity to do so, and is therefore inappropriate.

We thank the reviewer for the suggestion. We have now restored the 10-month age data. While we cannot provide a conclusive explanation for the

decrease in succinylation immunoreactivity at 10 months, we propose that succinylated sites become masked in fibrillar species, pointing out that both the A β and tau succinylation sites (that we have discovered) are buried in the core of the respective fibrillar forms of these proteins. At the same time, we acknowledge that our data do not state directly to this possibility and that it is also possible that the decrease may be due to a de-succinylation process, which can occur as a consequence of metabolic alterations related to pathology or disease. Sorting out these different possibilities would require a way to assess the presence of inaccessible succinylation sites at different time points in the mouse brains. This could potentially be addressed by further mass spectrometry experiments but would be a major effort that is beyond the scope of the current study.

Manuscript lines 312-319. Figure 5d and Extended Data Figure 3b.

However, the immunoreactivity of lysine succinylation was significantly decreased in 10-month-old wild-type and transgenic mice relative to 4-month-old mice, which results in a reduced colocalization between lysine succinylation and A β plaque accumulation (**Figure 5d** and **Extended Data Figure 3b**). This could result either from a decrease in lysine succinylations or from their sequestration into a context (e.g., perhaps in the form of A β plaques) that prevents Succinyl-K antibody from access to possibly buried succinylation sites. These findings suggest that APP succinylation might be involved in early A β aggregation events *in vivo*, while its role and mechanism in later events leading to subsequent plaque development remain to be further explored.

Along these lines, the authors response to the critique that the changes in succinylation in the transgenic mouse model don't correlate very well with the build up of AB is insufficient. While it is clearly true that succinylation is responding to the transgene, the fact that succinylation doesn't correlate well with the build up of AB provides a major caveat to the interpretation. As is, the authors seem to be saying that because the transgene causes a change in succinylation, it doesn't matter that that change does not seem to correlate well with the build up of pathological AB. When actually, this disconnect raises the clear possibility that AB build up and succinylation are not directly related. This doesn't mean that their results are not potentially interesting or important, but it does need to be acknowledged and discussed so that the reader is not left with an incorrect view of the data.

Manuscript lines 312-319. Figure 5d and Extended Data Figure 3b.

However, the immunoreactivity of lysine succinylation was significantly decreased in 10-month-old wild-type and transgenic mice relative to 4-month-old mice, which results in a reduced colocalization between lysine

succinylation and A β plaque accumulation (**Figure 5d** and **Extended Data Figure 3b**). This could result either from a decrease in lysine succinylations or from their sequestration into a context (e.g., perhaps in the form of A β plaques) that prevents Succi-K antibody from access to possibly buried succinylation sites. These findings suggest that APP succinylation might be involved in early A β aggregation events *in vivo*, while its role and mechanism in later events leading to subsequent plaque development remain to be further explored.

Manuscript lines 379-381. Figure 6c,d and extended data 4a,b.

Similar to what we observed found in A β deposits, 10-month-old wild-type and transgenic tau mice displayed a significant reduction in the levels of succinyl-lysine in comparison to 4-month-old mice, thereby leading to an attenuated colocalization between succinylation epitopes and tauopathy epitopes.

Similarly, the discussion that has been added to supposedly address the review point about the changes occurring in models of both AB and Tau is also not sufficient. The authors need to specifically discuss why succinylation may changing in both AB and Tau models, despite the models being different.

We agree with the reviewer that this should be discussed more thoroughly. Although we do not know the exact answer, it is tempting to speculate that this is a consequence of the complex interplay between metabolic/mitochondrial dysfunction and pathological aggregation. We agree that data mapping the progression of mitochondrial dysfunction to changes in succinylation in the two mouse models would help to establish the relationship between these events and protein aggregation, as well as to reveal any commonalities between the two mouse models. This represents an important set of experiments that would not be practicable to incorporate into the current study.

Manuscript lines 448-459

We showed that transgenic mouse strains of either tauopathy or amyloidosis phenotype, exhibit widespread increases in lysine succinylation at 4 months of age, which is not exclusive to tau and APP but parallels the early appearance of these proteinopathies. This suggests that each transgene is altering common processes (e.g., mitochondria/metabolism) in addition to tau or APP processing. Metabolism is altered even in embryonic cultures of mouse models of AD(Trushina *et al.* 2012). The data in Figure 4 demonstrate that disrupted mitochondrial function increases succinyl transferase in the cytosol. Indeed, the widespread succinylation in both models provides further evidence of that possibility. Interestingly, a pharmacological increase in vitamin B1 (a key vitamin in metabolism) significantly reduces A β burden(Pan *et al.* 2010) and tauopathy(Tapias *et al.*

2018) in mice and also showed encouraging results in AD patients(Gibson *et al.* 2020), suggesting these fundamental processes are critical even in mice genetically engineered to create the pathologies. A more precise interpretation requires knowing which proteins are succinylated since the human brain succinylome probably involves hundreds of succinylated proteins.

The above points are especially important, considering the revised abstract that states It is fine for the authors to state that they find a relationship in both human cases and mouse models. This certainly raises the possibility that there is a mechanistic relationship, which is interesting. However, there are major caveats to this relationship, and the actual mechanistic relationship is certainly unclear from their data. So, I don't understand why the authors keep insisting on such language.

As requested, we have changed the abstract to improve our findings, indicating that our results suggest a potential link (rather than establish a tight relationship) between succinylation and AD/proteinopathy.

Manuscript lines 129-131.

Our results suggest the potential existence of a link between lysine succinylation and AD-associated proteinopathies and that aberrant succinylation may be involved in the initiation and/or progression of AD.

I think the authors need to add a sentence to the results clearly stating that the peptide used for the tubulin polymerization assay is succinylated throughout, so that it is very clear for the reader.

This has been amended in the results section.

Manuscript 400-405.

To understand the role of succinylation in tau function, tubulin polymerization was assessed using the tau K19 peptide, a 99-residue 3-repeat tau microtubule-binding domain (MBD) fragment (MQ244-E372), and succinylated K19 (**Extended Data Figure 4d-f**). Native tau K19 promoted tubulin assembly as determined by increased light scattering at 350 nm, as previously reported (Cohen *et al.* 2011; Lu *et al.* 1999). Nevertheless, succinyl-CoA treated K19, which is succinylated at multiple lysine residues including Lys311, showed a complete suppression of tubulin assembly activity (**Figure 6k**).

The authors also continue to have language that is too strong for their data. For example, line 347 should say something like "may be a key molecular event that contributes" rather than "is a a key molecular even that promotes."

In line 421 it could say something like “our results raise the possibility that succinylation may provide a link” rather than our results “demonstrate that succinylation is a key link.” The data are compelling. There is no need to over sell it.

As requested, we have changed the abstract to improve our findings, indicating that our results suggest a potential link (rather than establish a tight relationship) between succinylation and AD/proteinopathy.

Manuscript lines 129-131.

Our results suggest the potential existence of a link between lysine succinylation and AD-associated proteinopathies and that aberrant succinylation may be involved in the initiation and/or progression of AD.

Reviewer #3 (Remarks to the Author):

The authors investigated lysine succinylation changes in the brain associated with Alzheimer’s disease comparing lysine succinylomes and proteomes from AD and control brains. They found in AD brains, succinylation declined for multiple mitochondrial proteins, and increased for a smaller number of cytosolic proteins, among which the amyloid precursor (APP) and tau exhibited the largest increases. In transgenic mice models of AD, they also found elevated succinylation of soluble and insoluble APP and tau. They examined the effect of succinylation and observed disrupted normal secretase processing of APP and A β accumulation. Succinylation of tau also promoted its aggregation and impaired microtubule assembly.

PTMs have been investigated in connection with neurodegenerative diseases. Tau proteins are known to undergo PTMs While succinylation is a well known modification that causes significant changes in proteins the studies presented by the authors are novel and expand our knowledge of the pathology in AD.

The authors have largely addressed prior reviewer concerns related to mass spectrometry. A concern about the prior submission was the exclusive focus on succinylation without comparison to other PTMS. The authors response seems to acknowledge this issue by expanding the introduction to cover other PTMs in AD as background but no additional data is provided. The response of the authors to other issues such as independent validation of data being beyond the scope of the paper is understandable.

Overall the quality of the revised paper is much improved and the findings are of interest to the field

Cohen, T. J., Guo, J. L., Hurtado, D. E., Kwong, L. K., Mills, I. P., Trojanowski, J. Q. and Lee, V. M. (2011) The acetylation of tau inhibits its function and promotes pathological tau aggregation. *Nature communications* **2**, 252.

- Gibson, G. E., Luchsinger, J. A., Cirio, R. et al. (2020) Benfotiamine and Cognitive Decline in Alzheimer's Disease: Results of a Randomized Placebo-Controlled Phase IIa Clinical Trial. *Journal of Alzheimer's Disease* **78**, 989-1010.
- Goedert, M., Spillantini, M. G., Falcon, B., Zhang, W., Newell, K. L., Hasegawa, M., Scheres, S. H. W. and Ghetti, B. (2021) Tau Protein and Frontotemporal Dementias. In: *Frontotemporal Dementias : Emerging Milestones of the 21st Century*, (B. Ghetti, E. Buratti, B. Boeve and R. Rademakers eds.), pp. 177-199. Springer International Publishing, Cham.
- Irwin, D. J., Cohen, T. J., Grossman, M., Arnold, S. E., Xie, S. X., Lee, V. M. Y. and Trojanowski, J. Q. (2012) Acetylated tau, a novel pathological signature in Alzheimer's disease and other tauopathies. *Brain* **135**, 807-818.
- Lu, P.-J., Wulf, G., Zhou, X. Z., Davies, P. and Lu, K. P. (1999) The prolyl isomerase Pin1 restores the function of Alzheimer-associated phosphorylated tau protein. *Nature* **399**, 784.
- Pan, X., Gong, N., Zhao, J. et al. (2010) Powerful beneficial effects of benfotiamine on cognitive impairment and β -amyloid deposition in amyloid precursor protein/presenilin-1 transgenic mice. *Brain* **133**, 1342-1351.
- Tapias, V., Jainuddin, S., Ahuja, M. et al. (2018) Benfotiamine treatment activates the Nrf2/ARE pathway and is neuroprotective in a transgenic mouse model of tauopathy. *Human Molecular Genetics* **27**, 2874-2892.
- Trushina, E., Nemutlu, E., Zhang, S. et al. (2012) Defects in Mitochondrial Dynamics and Metabolomic Signatures of Evolving Energetic Stress in Mouse Models of Familial Alzheimer's Disease. *PLOS ONE* **7**, e32737.
- Yang, Y. and Gibson, G. E. (2019) Succinylation Links Metabolism to Protein Functions. *Neurochemical Research* **44**, 2346-2359.
- Zhang, W., Falcon, B., Murzin, A. G., Fan, J., Crowther, R. A., Goedert, M. and Scheres, S. H. W. (2019) Heparin-induced tau filaments are polymorphic and differ from those in Alzheimer's and Pick's diseases. *eLife* **8**, e43584.

REVIEWER COMMENTS

Reviewer #2 (Remarks to the Author):

Overall, the manuscript is substantially improved and almost all of my comments have been addressed. However, in figure 5, the authors still have not provided dual IF with a control antibody that is unchanged. While I understand that performing quantification of imaging on multiple sections and multiple mice somewhat mitigates the concern, it seems like it is really not that difficult to perform this standard control. The authors should have some tissue remaining from the original experiments to do this quickly. Thus, this control should be provided or a compelling reason stated why this control cannot be provided.

Reviewer #2 (Remarks to the Author):

Overall, the manuscript is substantially improved and almost all of my comments have been addressed. However, in figure 5, the authors still have not provided dual IF with a control antibody that is unchanged. While I understand that performing quantification of imaging on multiple sections and multiple mice somewhat mitigates the concern, it seems like it is really not that difficult to perform this standard control. The authors should have some tissue remaining from the original experiments to do this quickly. Thus, this control should be provided or a compelling reason stated why this control cannot be provided.

We agree with the reviewer about the importance of the concern. We believe that our extended data, in particular Extended Data Figures 3 and 4, already answer the Reviewer's question. We have shown that the immunoreactivity of MAP2 (a neuronal marker, cyan) does not change during aging, either in WT or A β /tau TG mice. We have created a collage from the first column of panel a in those figures, which demonstrates that the decline in succinylation at 10 months is not an artifact.

We added the following comment to address this concern (Lines 471-476) of the marked manuscript.

This change is not likely to be a technical artifact. All sections were stained at the same time under the same conditions (solutions, washing, temperature, antibody preparation, etc.) and analyzed under identical experimental settings. In addition, perusal of the first column of panel in the Extended Data Figures 3 and 4 show that the immunoreactivity of MAP2 (a neuronal marker, cyan) does not change during aging, either in WT or A β /tau TG mice. The decline in succinylation may reflect either de-succinylation processes, or sequestration of succinylated sites away from labeling antibodies.

REVIEWER COMMENTS

Reviewer #2 (Remarks to the Author):

The authors have now addressed my main concerns.